# REX-THINKER: GROUNDED OBJECT REFERRING VIA CHAIN-OF-THOUGHT REASONING

**Qing Jiang**[1,2*]**, Xingyu Chen**[2,3,4*]**, Zhaoyang Zeng**[2]**, Junzhi Yu**[3]**, & Lei Zhang**[1,2†]
[1] South China University of Technology    [2] International Digital Economy Academy (IDEA)
[3] Peking University    [4] Zhongguancun Adacemy

## ABSTRACT

Object referring aims to detect all objects in an image that match a given language description. A robust model for this task must be grounded, providing explainable and visually faithful predictions. This demands two key properties: **1) Verifiability**, through interpretable reasoning linked to visual evidence; and **2) Trustworthiness**, by rejecting expressions without matching objects, thereby mitigating hallucination. Current methods, often treating referring as direct bounding box prediction, largely fail to provide such interpretability or effectively abstain. In this work, we propose Rex-Thinker, an MLLM framework that reformulates object referring as an explicit Chain-of-Thought (CoT) reasoning task. Recognizing that standalone detectors lack sufficient language comprehension and MLLMs struggle with precise localization, Rex-Thinker adopts a two stage approach. It first leverages an open-vocabulary detector for candidate object proposals, then employs an MLLM to perform step-by-step reasoning over these candidates to assess their match with the expression. To enable this, we construct HumanRef-CoT, a large-scale (90,824 samples) CoT-style referring dataset. Each reasoning trace follows a structured planning, action, and summarization format, facilitating the learning of decomposed, grounded reasoning. Rex-Thinker is trained in two stages: supervised fine-tuning for structured reasoning, followed by GRPO-based reinforcement learning for enhanced accuracy and generalization. Extensive experiments demonstrate that Rex-Thinker not only achieves state-of-the-art performance in precision and interpretability but also drastically improves the ability to reject hallucinated outputs on in-domain evaluation, while showing strong generalization in out-of-domain settings. Code is available at `https://github.com/IDEA-Research/Rex-Thinker`.

## 1 INTRODUCTION

Object Referring, also known as Referring Expression Comprehension (REC) (Qiao et al. (2020); Yu et al. (2016); Mao et al. (2016); Kazemzadeh et al. (2014); Zhang et al. (2019); Yu et al. (2018); Yang et al. (2019); Liao et al. (2020)), aims to predict the bounding boxes for objects in an image that match a given natural language description, which may refer to visual attributes, spatial relations, or interactions. This task has broad applications, however, compared to standard open-vocabulary object detection (Jiang et al. (2025a); Ren et al. (2024a;b); Liu et al. (2024); Yao et al. (2022a); Jiang et al. (2024a; 2023); Li et al. (2024a); Ren et al. (2024a); Li et al. (2022); Cheng et al. (2024); Minderer et al. (2022); Zareian et al. (2021); Wu et al. (2023)), REC is significantly more challenging, as it requires both fine-grained visual grounding and more complicated language understanding.

Benefiting from the strong language comprehension capabilities of large language models (LLMs), multimodal large language models (MLLMs) have demonstrated impressive performance on this task. There are mainly two paradigms: one treats bounding box coordinates as text tokens and predicts them directly (Chen et al. (2023); You et al. (2023); Zhang et al. (2024); Wang et al. (2023); Zhan et al. (2025; 2024); Bai et al. (2025); Wu et al. (2024); Chen et al. (2024); McKinzie et al. (2024)), while the other adopts a retrieval-based strategy (Ma et al. (2024); Jiang et al. (2024b;

---

\* Equal contributions, work done during internship or academic visit at IDEA. † Corresponding author.

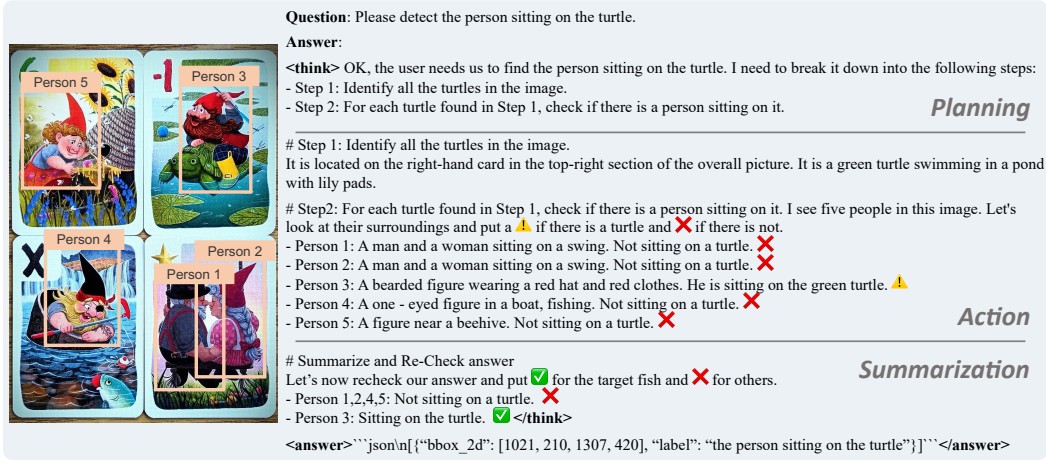

**Question**: Please detect the person sitting on the turtle.

**Answer**:

**<think>** OK, the user needs us to find the person sitting on the turtle. I need to break it down into the following steps:
- Step 1: Identify all the turtles in the image.
- Step 2: For each turtle found in Step 1, check if there is a person sitting on it.

*Planning*

# Step 1: Identify all the turtles in the image.
It is located on the right-hand card in the top-right section of the overall picture. It is a green turtle swimming in a pond with lily pads.

# Step2: For each turtle found in Step 1, check if there is a person sitting on it. I see five people in this image. Let's look at their surroundings and put a ⚠️ if there is a turtle and ❌ if there is not.
- Person 1: A man and a woman sitting on a swing. Not sitting on a turtle. ❌
- Person 2: A man and a woman sitting on a swing. Not sitting on a turtle. ❌
- Person 3: A bearded figure wearing a red hat and red clothes. He is sitting on the green turtle. ⚠️
- Person 4: A one - eyed figure in a boat, fishing. Not sitting on a turtle. ❌
- Person 5: A figure near a beehive. Not sitting on a turtle. ❌

*Action*

# Summarize and Re-Check answer
Let's now recheck our answer and put ✅ for the target fish and ❌ for others.
- Person 1,2,4,5: Not sitting on a turtle. ❌
- Person 3: Sitting on the turtle. ✅ **</think>**

*Summarization*

**<answer>**```json\n[{"bbox_2d": [1021, 210, 1307, 420], "label": "the person sitting on the turtle"}]```**</answer>**

Figure 1: An example of Rex-Thinker for object referring with CoT reasoning of planning (task decomposition), action (evaluating each candidate), and summarization (final decision). Each step is grounded in a specific hint box (as denoted in the left image), enabling interpretable predictions.

2025b)), where the model is given a set of candidate boxes and predicts the box indices that match the expression. Although both approaches have shown promising results, they remain fundamentally implicit, lacking interpretable reasoning steps that reveal how the model arrives at its final prediction. Furthermore, these models are prone to hallucination (Jiang et al. (2025b)), often producing outputs for objects that do not exist in the image, thereby limiting their reliability in real-world applications.

We argue that a robust referring system should be *grounded*, i.e., its predictions must be both explainable and tightly linked to visual evidence. This requires two essential properties: **1) Verifiable**, by providing an explicit reasoning process that allows its decisions to be examined and traced to specific image regions; and **2) Trustworthy**, by minimizing hallucinated outputs and learning to reject when no object in the image satisfies the given description. To meet these criteria, we draw inspiration from how humans naturally approach referring expressions. For example, when asked to locate "the person wearing a blue shirt", humans would typically first identify all people in the image, then examine each one to determine whether it matches the described attribute. This step-by-step approach reflects a grounded reasoning process, i.e., first localizing relevant object candidates, and then carefully verifying each one against the expression.

Motivated by this observation, we propose Rex-Thinker, an MLLM framework that performs object referring through explicit Chain-of-Thought (CoT) reasoning. Specifically, given an image and a referring expression, we first use an open-vocabulary object detector (Liu et al. (2024)) to extract all candidate object boxes corresponding to the referred category. These candidate boxes, along with the image and the expression, are then passed into the MLLM to form a two stage system. Rex-Thinker follows a structured CoT framework consisting of three key stages as shown in Figure 1: **1) Planning**, where the model decomposes the referring expression into subgoals; **2) Action**, where the model examines each candidate box to determine whether it satisfies its current subgoal; **3) Summarization**, where it aggregates the intermediate decisions to produce the final prediction. Following DeepSeek-R1 (Guo et al. (2025a)), we instruct the model to place its reasoning steps within a `<think>...</think>` block and to output the final prediction inside a `<answer>...</answer>` block. This structured reasoning process not only improves interpretability, but also enables transparent and verifiable predictions, as each reasoning step is grounded in a specific candidate region.

To support this CoT framework, we construct a CoT-style referring dataset named HumanRef-CoT, containing 90,824 samples generated by prompting GPT-4o (Hurst et al. (2024)) on the HumanRef (Jiang et al. (2025b)) dataset. Each example is annotated with a structured reasoning trace following the planning, action, and summarization paradigm, enabling explicit supervision for step-by-step reasoning. We train our model in two stages: a cold-start supervised fine-tuning phase to teach the model how to perform structured reasoning, followed by reinforcement learning (RL) based on Group Relative Policy Optimization (GRPO) (Shao et al. (2024)) to further improve accuracy and generalization. Experiments demonstrate that our CoT-based approach consistently outperforms direct coordinate prediction baselines. On the in-domain HumanRef benchmark, our model achieves state-of-the-art results with higher detection accuracy and significantly fewer hallucinated outputs, especially on rejection cases. In out-of-domain evaluations on RefCOCOg (Mao et al. (2016)), the

model trained only on HumanRef-CoT shows strong zero-shot generalization. Further fine-tuning with GRPO on RefCOCOg yields additional performance gains while preserving the model's ability to perform grounded CoT reasoning across arbitrary object categories. Our contributions lie in:

- We formulate the grounded object referring task as a **planning–action–summarization** problem, leveraging Chain-of-Thought reasoning to build a verifiable and trustworthy system.
- We introduce HumanRef-CoT, the first large-scale dataset for grounded object referring with step-by-step reasoning annotations, enabling the supervised training of model interpretability.
- We propose Rex-Thinker, a grounded object referring model trained via cold-start SFT and GRPO-based reinforcement learning. Rex-Thinker achieves SOTA performance on the Human-Ref benchmark and demonstrates strong generalization on out-of-domain scenes and objects.

## 2 RELATED WORK

**MLLM-based Object Referring Methods.** Recent progress in multimodal large language models (MLLMs) OpenAI (2023); Bai et al. (2025); Wu et al. (2024); Chen et al. (2024); Deitke et al. (2024); Agrawal et al. (2024); Wang et al. (2024); Li et al. (2024b; 2025); Liu et al. (2023); Chen et al. (2025a); Guo et al. (2025b); Yao et al. (2022b;a); Yang et al. (2023b) has led to strong performance in referring expression comprehension. Existing approaches typically follow two paradigms. One line of work treats bounding box coordinates as textual tokens Chen et al. (2021) and directly generates them during decoding Chen et al. (2023); You et al. (2023); Wang et al. (2023); Zhan et al. (2025); Zhang et al. (2024). The other line formulates the task as retrieval Jiang et al. (2024b); Ma et al. (2024); Jiang et al. (2025b), where a detector proposes candidate regions and the model selects the best-matching box indices based on the input expression. This decouples localization from semantic understanding and simplifies learning. While both paradigms achieve strong results on standard benchmarks such as RefCOCO/+/g Mao et al. (2016); Yu et al. (2016), they face key limitations: a lack of interpretability and an inability to abstain when no object in the image matches the expression Jiang et al. (2025b). To address this, we introduce a Chain-of-Thought reasoning framework that enables step-by-step evaluation over candidate boxes. This improves interpretability, reduces hallucinations, and grounds the model's predictions in the input image.

**Reasoning-based LLMs and MLLMs.** Recent work in large language models (Jaech et al. (2024); Guo et al. (2025a); Team et al. (2025); Muennighoff et al. (2025); Xiang et al. (2025); Xiong et al. (2025); Chu et al. (2025); OpenAI et al. (2025)) has demonstrated that reasoning ability can be significantly enhanced through Chain-of-Thought (CoT) training or reinforcement learning-based post-training. OpenAI o1 (Jaech et al. (2024)) model demonstrates that inference-time scaling can greatly enhance performance on complex tasks like math and coding. DeepSeek-R1 (Guo et al. (2025a)) introduces GRPO (Shao et al. (2024)) as a post-training method to improve reasoning without requiring costly critic models. In the multimodal domain, efforts such as LLaVA-CoT (Xu et al. (2024)) and LlamaV-o1 (Thawakar et al. (2025)) aim to enhance reasoning by constructing CoT-style data or employing multi-step curriculum learning, without relying on reinforcement learning. For referring expression comprehension task, reinforcement learning has been used in early work to model the grounding process as a sequential reasoning problem, such as through iterative shrinking Sun et al. (2021) or dynamic reasoning steps Zhang et al. (2023). More recently, inspired by DeepSeek-R1 (Guo et al. (2025a)), a growing number of works adopt GRPO-based post-training to endow MLLMs with reasoning capabilities. GRPO has been successfully applied to enhance multimodal reasoning across a wide range of domains, including mathematical problem solving (Yang et al. (2025); Peng et al. (2025); Zhang et al. (2025); Deng et al. (2025); Wei et al. (2025)), video understanding (Feng et al. (2025); Liao et al. (2025)), and perception tasks (Liu et al. (2025a;b); Ma et al. (2025); Shen et al. (2025); Yu et al. (2025)) such as object detection, segmentation, and referring expression comprehension. Following the DeepSeek-R1 paradigm, we first fine-tune Rex-Thinker on structured CoT data to teach the model how to perform grounded object reasoning. GRPO is then applied in a second stage to further improve accuracy and generalization.

## 3 CHAIN-OF-THOUGHT REASONING REFERRING DATA

High-quality supervision is critical for teaching the model to reason explicitly. To this end, we develop a data engine that generates structured referring annotations aligned with our Chain-of-Thought formulation. In this section, we introduce the design principles of our CoT reasoning

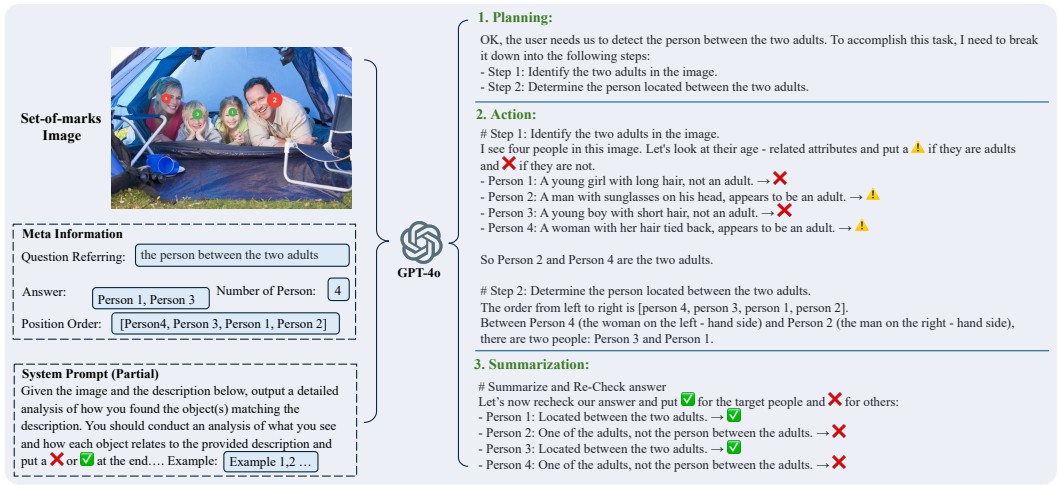

Figure 2: Overview of the proposed CoT reasoning referring data engine. We prompt GPT-4o to generate a three-step CoT reasoning process, including planning, action, and summarization.

structure and present the data construction pipeline that transforms existing REC annotations into step-by-step reasoning traces suitable for supervised training.

## 3.1 CoT Formulation

The core idea behind our CoT formulation for REC is to transform the task into a structured, grounded reasoning process over a set of candidate objects. Rather than directly predicting the referred object, the model evaluates each candidate in sequence, guided by input box hints that localize specific regions in the image. We decompose this CoT process into three key stages:

- **Planning:** The model analyzes the complexity of the referring expression and determines how many reasoning steps are needed. For simple expressions, it may plan a single step to directly match an attribute such as color or size. For more complex expressions, the model generates a multi-step plan, where each step focuses on resolving a specific sub-aspect.
- **Action:** Based on the reasoning plan, the model checks whether each candidate region, grounded via its input box hint, satisfies the current subgoal. This makes the reasoning clear and directly tied to specific regions in the image.
- **Summarization:** Finally, the model reviews the reasoning results across all steps and determines which objects best match the overall expression and outputs the final prediction.

This structured CoT process improves both interpretability and verifiability. Each candidate is evaluated corresponding to the input box hints, allowing every reasoning step to be explicitly grounded to a specific region of the image. This makes the model's decisions transparent and easy to trace. Additionally, breaking complex expressions into sub-tasks enables step-by-step reasoning, which enhances accuracy and reflects how humans typically process such tasks.

## 3.2 Data Engine Pipeline

Building on the structured CoT formulation, we develop a data engine that leverages GPT-4o (Hurst et al. (2024)) to generate high-quality CoT annotations tailored to the referring task.

**Data Acquisition** We construct our CoT dataset based on HumanRef (Jiang et al. (2025b)), a recently proposed dataset specifically designed for REC in human-centric scenarios. Unlike prior REC datasets such as RefCOCO/+/g (Mao et al. (2016); Yu et al. (2016)), HumanRef emphasizes multi-instance referring expressions, where a single expression may refer to multiple target persons. It also categorizes expressions into six distinct subsets: attribute, position, interaction, reasoning, celebrity recognition, and rejection. Since the HumanRef dataset provides all person boxes in an image, it can be directly used in our CoT annotation pipeline.

**GPT-4o Annotation** To generate high-quality CoT annotations, we employ in-context prompting with GPT-4o (Hurst et al. (2024)) as shown in Figure 2. Given an image and the bounding boxes of all persons within it, we apply the Set-of-Mark (Yang et al. (2023a)) strategy: each individual is

| Subset | GPT-4o Annotated Data | Removed Data | Removal Rate |
|---|---|---|---|
| Celebrity | 6,775 | 294 | 4.3% |
| Interaction | 5,875 | 72 | 1.2% |
| Position | 15,950 | 533 | 3.3% |
| Reasoning | 9,488 | 719 | 7.6% |
| Attribute | 37,648 | 1,184 | 3.1% |
| Rejection | 19,096 | 1,184 | 6.2% |

Table 1: Statistics of our two-stage automated filtering process on the generated HumanRef-CoT data. This process removes samples with logical inconsistencies or incorrect final predictions.

labeled with an indexed visual marker, where ground-truth targets are marked in green and others in red. This design grounds the answer and guides GPT-4o to reason along the correct path. The prompt includes three key components: 1) meta-information such as the referring question, the number of people, their left-to-right spatial order, and the correct answer; 2) a system prompt specifying the desired planning–action–summarization structure; and 3) several in-context examples written by humans to illustrate the expected reasoning format. In essence, we provide GPT-4o with both the referring expression and its ground-truth answer, and prompt it to generate step-by-step reasoning in our CoT format.

**Quality Control:** To ensure the quality and logical consistency of these annotations, we implement a rigorous two-stage filtering process that goes beyond simply matching the final answer. First, we enforce internal logical coherence by verifying that the intermediate conclusions in the "Action" phase are consistent with the final "Summarization" phase. Second, we ensure final accuracy by requiring the "Summarization" outcome to perfectly match the ground-truth labels. This automated check is performed by parsing the correct and incorrect emojis generated by the model at each step. This filtering process is crucial for data quality, and as shown in Table 1, it identifies and removes a significant number of initially generated samples due to inconsistencies, especially in complex cases like the "Reasoning" subset (7.6% removal rate).

To further validate the quality of the filtered data, we conducted a manual human evaluation study (full details in Appendix A.2.2). The study confirmed the high quality of the final dataset, finding zero logical or summarization errors in a 600-sample review, with only a minor factual error rate of 1.2% in intermediate reasoning steps. This confirms our dataset provides a reliable training signal.

In total, we construct a total of 90,824 high-quality CoT annotations based on the HumanRef dataset, which we refer to as HumanRef-CoT. This diverse and large-scale dataset serves as the foundation for both our initial cold-start SFT and GRPO-based post-training.

## 4 METHOD

### 4.1 RETRIEVAL-BASED OBJECT REFERRING

To leverage the CoT-style referring data, we present Rex-Thinker, a retrieval-based model that performs object referring through explicit Chain-of-Thought reasoning.

To support explicit Chain-of-Thought (CoT) reasoning, we reformulate referring expression comprehension as a retrieval-based task. As shown in Figure 3, rather than directly regressing bounding boxes, we first use an open-vocabulary detector (Liu et al. (2024)) to extract a set of candidate object boxes corresponding to the referred object category. These candidate boxes serve as *box hints* to guide both the reasoning path and final decision of the model. This retrieval-based formulation brings two key advantages. First, during the reasoning phase, the model evaluates each candidate region in the order they appear in the input box hints (e.g., "Person 1" corresponds to the first input box). This alignment ensures that each step in the CoT trace is explicitly grounded to a specific region in the image, making the reasoning process interpretable and visually verifiable. Second, during the prediction phase, the model can directly select from the input box hints when producing the final output, thereby easing the challenge of precise coordinate regression.

We build Rex-Thinker on top of Qwen2.5-VL-7B (Bai et al. (2025)), preserving its original architecture and using JSON-format bounding box coordinates as the final output. The model input includes

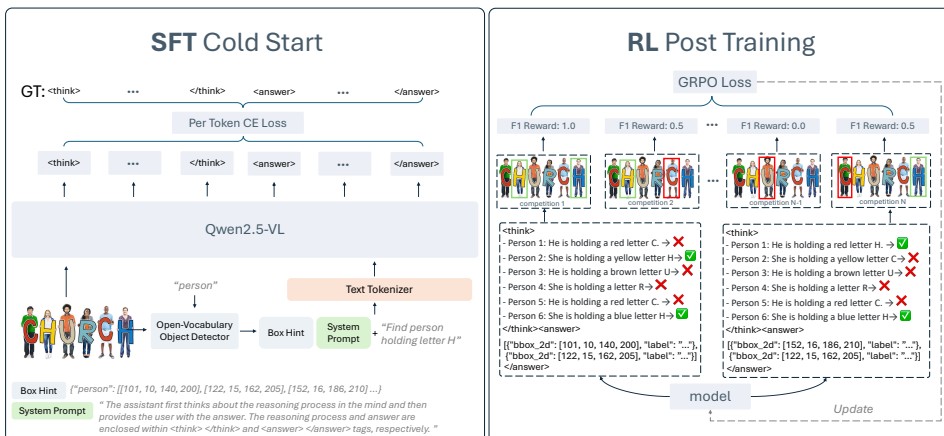

Figure 3: Overview of the Rex-Thinker architecture and our two-stage training methods

---

`<image>`. A conversation between User and Assistant. The user asks a question, and the Assistant solves it. The assistant first thinks about the reasoning process in the mind and then provides the user with the answer. The reasoning process and answer are enclosed within `<think> </think>` and `<answer> </answer>` tags, respectively, i.e., `<think>` reasoning process here `</think>` `<answer>` answer here `</answer>`. Hint: Object and its coordinates in this image: Box Hint. User: Locate Referring. Assistant:

---

Table 2: Prompt Template for Rex-Thinker. Box Hint and Referring will be replaced with the input candidate boxes and the referring expression, respectively.

the image, the box hint, the referring expression, and a system prompt that guides the reasoning process. The input prompt format in shown in Table 2.

## 4.2 TRAINING

Our training strategy is a two-stage process, consisting of supervised fine-tuning (SFT) for cold start and GRPO-based reinforcement learning (RL) for post-training. We adopt this methodology, inspired by DeepSeek-R1 (Guo et al. (2025a)), because it has proven highly effective for teaching models to follow complex, structured reasoning paths and for subsequently refining these behaviors through reward-guided optimization.

**SFT Cold Start** We begin by fine-tuning Rex-Thinker on the HumanRef-CoT dataset to instill the ability to perform structured reasoning following our defined planning, action, and summarization format. We apply cross-entropy loss at the token level to both the reasoning trace and the final answer, providing strong supervision across the entire generation process. This stage teaches the model how to reason step-by-step in a CoT manner and also how to utilize the provided box hints to guide its final predictions.

**GRPO Post Training** While SFT teaches the model to follow our grounded CoT format, its strict token-level supervision may constrain the model to explore alternative reasoning traces and generalize beyond the training data. To enhance generalization beyond the limitations of supervised learning, we employ GRPO-based (Shao et al. (2024)) reinforcement learning for post-training. GRPO optimizes model performance by 1) sampling multiple candidate responses for each question and, 2) selectively reinforcing responses that achieve higher task-level rewards.

In our setting, given an image and a referring expression $(I, x)$, the model generates a group of $G$ complete responses $o_1, o_2, \ldots, o_G$ from the current model $\pi_\theta$. Each response contains a full reasoning trace and a final predicted bounding box set. For each $o_i$, we compute a scalar reward $r_i$ (detailed in below), and normalize these rewards to estimate group-relative advantages:

$$A_i = (r_i - \text{mean}(r_1, \ldots, r_G))/\text{std}(r_1, \ldots, r_G). \tag{1}$$

Define the token-level advantage estimates $\hat{A}_{i,t} = A_i$, and the importance ratio at each decoding step as follows,

$$\rho_{i,t} = \frac{\pi_\theta(o_{i,t} \mid (I, x), x, o_{i,<t})}{\pi_{\theta_{\text{old}}}(o_{i,t} \mid (I, x), x, o_{i,<t})}, \tag{2}$$

where $\pi_{\theta_{\text{old}}}$ is the model before the current update. Then, the GRPO objective is given as follows,

$$\mathcal{J}_{\text{GRPO}}(\theta) = \frac{1}{G} \sum_{i=1}^{G} \frac{1}{|o_i|} \sum_{t=1}^{|o_i|} \left[ \min \left( \rho_{i,t} \hat{A}_{i,t}, \text{clip} \left( \rho_{i,t}, 1 - \epsilon, 1 + \epsilon \right) \hat{A}_{i,t} \right) - \beta \mathbb{D}_{\text{KL}} \left[ \pi_\theta \| \pi_{\text{ref}} \right] \right] \quad (3)$$

$$\mathbb{D}_{\text{KL}} \left[ \pi_\theta \| \pi_{\text{ref}} \right] = \frac{\pi_\theta(o_{i,t} \mid (I, x), x, o_{i,<t})}{\pi_{\text{ref}}(o_{i,t} \mid (I, x), x, o_{i,<t})} - \log \frac{\pi_\theta(o_{i,t} \mid (I, x), x, o_{i,<t})}{\pi_{\text{ref}}(o_{i,t} \mid (I, x), x, o_{i,<t})} - 1, \quad (4)$$

where $\epsilon$ is a hyperparameter controlling the clipping range, $\pi_{\text{ref}}$ is the model fixed after SFT stage, and $\beta$ is the KL penalty coefficient.

We argue that this formulation is suited to policy exploration in our reasoning-driven task. Given that the model is already capable of producing structured reasoning traces after SFT, GRPO allows it to freely explore different reasoning paths. In each iteration, the model generates diverse reasoning strategies that may lead to different predicted object sets. The reward function then guides the model to reinforce reasoning paths that yield accurate predictions.

*Accuracy Reward*: We use the F1 score to jointly evaluate the precision and recall of the model's predictions. Given a set of predicted boxes $\hat{B}$ and the ground-truth set $B^*$, since box hints are provided as input, we define a match only when a predicted box exactly overlaps with a ground-truth box (i.e., IoU = 1), which encourages the model to select final outputs directly from the box hints. Let $M = \hat{B} \cap B^*$ denote the set of matched box pairs under this criterion. We compute precision, recall, and the F1 reward as:

$$\text{Precision} = \frac{|M|}{|\hat{B}|}, \quad \text{Recall} = \frac{|M|}{|B^*|}, \quad r^{\text{F1}} = \frac{2 \cdot \text{Precision} \cdot \text{Recall}}{\text{Precision} + \text{Recall}}. \quad (5)$$

*Format Reward*: To encourage interpretable and well-structured output, we define a format reward $r^{\text{fmt}}$ that equals 1 if the output follows the required structure: the reasoning must be enclosed in <think>...</think> and the final result in <answer>...</answer>, and 0 otherwise.

The total reward is a weighted combination of the accuracy and format rewards, i.e., $r_i = \lambda \cdot r_i^{\text{F1}} + (1 - \lambda) \cdot r_i^{\text{fmt}}$, where $\lambda = 0.9$ to emphasize correct detection while still enforcing output structure.

## 5 EXPERIMENTS

In this section, we evaluate the effectiveness of our CoT-based reasoning approach for object referring. We first introduce the experimental setup, then present in-domain results on the HumanRef benchmark, followed by out-of-domain evaluation on the RefCOCOg benchmark. Lastly, we conduct ablation studies to analyze key design choices.

### 5.1 EXPERIMENTAL SETUP

**Model Setting.** We use Qwen2.5-VL-7B as our base model. Qwen2.5-VL outputs absolute bounding box coordinates rather than quantized tokens, which provides better localization accuracy for detection tasks. We adopt this native decoding format for final bounding box predictions.

**SFT Training.** We fine-tune the model on the full HumanRef-CoT dataset using supervised learning. We use a learning rate of 2e-5, weight decay of 0.01, and cosine decay scheduling. The maximum generation length is set to 2048 tokens. During SFT, the vision encoder and MLP projector are frozen, and we update only the LLM parameters. For each training instance, we use all person bounding boxes in the image as box hints.

**GRPO Training.** After SFT, we apply GRPO for reward-driven post-training. We continue training on HumanRef-CoT, but randomly shuffle the box hint order in each training data to create novel input configurations. This leads the model to explore different reasoning paths than those seen during SFT. During this phase, we train only the LLM. We use a learning rate of 1e-6, 8 rollout samples per input, a batch size of 8, and gradient accumulation steps of 2. The KL penalty coefficient $\beta$ is set to 0.04, the sampling temperature to 1.0, and the output length remains 2048 tokens.

| Method | Attribute | | | Position | | | Interaction | | | Reasoning | | | Celebrity | | | Average | | | Rejection |
|---|---|---|---|---|---|---|---|---|---|---|---|---|---|---|---|---|---|---|---|
| | R | P | DF1 | R | P | DF1 | R | P | DF1 | R | P | DF1 | R | P | DF1 | R | P | DF1 | Score |
| DINOX (Ren et al. (2024a)) | 59.5 | 28.8 | 20.9 | 78.8 | 28.1 | 17.6 | 67.3 | 28.5 | 18.9 | 76.2 | 32.1 | 22.2 | 94.1 | 48.0 | 37.0 | 75.2 | 33.1 | 23.3 | 36.0 |
| InternVL-2.5-8B (Chen et al. (2025b)) | 23.5 | 39.0 | 27.1 | 23.0 | 28.0 | 24.3 | 27.8 | 40.1 | 31.3 | 17.5 | 22.8 | 18.9 | 57.4 | 59.3 | 58.0 | 29.8 | 37.8 | 31.9 | 54.9 |
| Ferret-7B (You et al. (2023)) | 27.9 | 44.4 | 30.4 | 30.2 | 36.2 | 29.8 | 30.8 | 41.8 | 31.2 | 19.7 | 33.7 | 22.8 | 63.2 | 60.0 | 57.5 | 34.4 | 43.2 | 34.3 | 2.0 |
| Groma-7B (Ma et al. (2024)) | 67.5 | 47.8 | 38.6 | 63.2 | 43.1 | 37.2 | 66.6 | 48.1 | 40.6 | 59.1 | 41.4 | 34.8 | 73.2 | 63.3 | 59.1 | 65.9 | 48.7 | 42.1 | 0.0 |
| ChatRex-7B (Jiang et al. (2024b)) | 44.3 | 78.0 | 51.8 | 48.0 | 66.7 | 52.5 | 49.6 | 74.8 | 56.5 | 36.6 | 65.1 | 42.8 | 73.7 | 76.5 | 74.2 | 50.4 | 72.2 | 55.6 | 0.0 |
| Qwen2.5-VL-7B (Bai et al. (2025)) | 49.1 | 71.3 | 54.4 | 50.2 | 61.7 | 52.8 | 48.2 | 66.3 | 53.2 | 34.6 | 61.2 | 40.3 | 80.3 | 81.9 | 80.1 | 52.5 | 68.5 | 56.2 | 7.1 |
| DeepSeek-VL2-small (Wu et al. (2024)) | 52.3 | 78.0 | 57.7 | 56.4 | 66.1 | 58.1 | 55.4 | 75.7 | 60.7 | 46.6 | 61.7 | 50.1 | 85.9 | 74.3 | 70.7 | 59.3 | 71.2 | 59.5 | 3.1 |
| Molmo-7B-D (Deitke et al. (2024)) | 82.7 | 86.4 | 76.3 | 78.0 | 80.6 | 72.4 | 69.9 | 77.7 | 66.1 | 72.1 | 80.4 | 65.5 | 85.9 | 87.5 | 82.9 | 77.7 | 82.5 | 72.6 | 68.6 |
| RexSeek-7B (Jiang et al. (2025b)) | 87.2 | 86.8 | 81.5 | 86.1 | 86.3 | 83.8 | 84.8 | 84.6 | 80.7 | 87.8 | 84.7 | 81.5 | 83.4 | 86.5 | 84.2 | 85.9 | 85.8 | 82.3 | 54.1 |
| Rex-Thinker-Plain | 83.0 | 88.7 | 81.4 | 82.5 | 83.9 | 81.3 | 80.1 | 85.6 | 80.2 | 80.5 | 82.2 | 77.3 | 86.7 | 88.7 | 86.8 | 82.6 | 85.8 | 81.4 | 53.5 |
| Rex-Thinker-CoT | 86.6 | 87.7 | 82.7 | 86.5 | 87.0 | 84.3 | 79.6 | 81.7 | 77.2 | 85.7 | 83.8 | 80.3 | 87.6 | 89.5 | 87.2 | 85.2 | 85.9 | 82.3 | 67.3 |
| Rex-Thinker-GRPO | 88.5 | 88.7 | 84.1 | 87.2 | 87.1 | 84.6 | 81.5 | 83.5 | 79.1 | 87.7 | 85.4 | 82.3 | 88.0 | 89.3 | 87.2 | 86.6 | 86.8 | 83.5 | 68.2 |

Table 3: In-domain evaluation results on the HumanRef benchmark. R, P, and DF1 represent Recall, Precision, and DensityF1. The **blod** and underline fonts indicate the best and second numbers.

**Evaluation Protocol.** For in-domain evaluation, we evaluate our model on the HumanRef benchmark, which consists of six subsets: attribute, position, interaction, reasoning, celebrity recognition, and rejection. Following (Jiang et al. (2025b)), we report Recall (R), Precision (P), and DensityF1 (DF1) scores averaged over IoU thresholds from 0.5 to 0.95. For the rejection subset, we report the rejection score, defined as the proportion of 1,000 images where the model correctly outputs no bounding box when the object described by the referring expression is not present in the image. For out-of-domain evaluation, we evaluate our model on the RefCOCOg dataset and report accuracy at an IoU threshold of 0.5. We compare three variants: 1) Rex-Thinker-Plain, which is trained on HumanRef-CoT using SFT only on the final detection outputs, without reasoning supervision; 2) Rex-Thinker-CoT, which is trained with SFT on both the reasoning process and the final answer; and 3) Rex-Thinker-GRPO, which is initialized from Rex-Thinker-CoT and optimized with GRPO.

## 5.2 In-domain Evaluation Results

We begin by evaluating in-domain performance on the HumanRef benchmark to assess referring accuracy within the person domain. As shown in Table 3, Rex-Thinker-CoT, trained with structured CoT supervision, consistently outperforms Rex-Thinker-Plain across most evaluation subsets. Specifically, it achieves average improvements of +2.6 Recall, +0.1 Precision, and +0.9 DensityF1, confirming that step-by-step reasoning leads to more accurate and well-grounded predictions. Most notably, the CoT-trained model shows a remarkable 13.8 point improvement in terms of Rejection Score on the rejection subset. This metric directly quantifies the model's trustworthiness by assessing its ability to correctly abstain when no valid target exists. This significant gain indicates substantially reduced hallucination rates, a critical capability for real-world applications requiring high reliability.

Further performance gains are realized through GRPO-based reinforcement learning. Rex-Thinker-GRPO demonstrates consistent improvements over Rex-Thinker-CoT, achieving gains of +1.4 Recall, +0.9 Precision, and +1.2 DensityF1. These results demonstrate the effectiveness of reward-based optimization in improving both the model's reasoning process and prediction accuracy. While supervised CoT training provides strong guidance, GRPO enables dynamic exploration of alternative reasoning strategies that better optimize for task-level objectives, leading to improved performance.

## 5.3 Out-of-domain Evaluation Results

To evaluate the generalization of Rex-Thinker to unseen object categories, we conduct experiments on the out-of-domain RefCOCOg dataset. We first adopt a zero-shot setting: the model is trained only on HumanRef-CoT and directly evaluated on RefCOCOg. Given a referring expression and its original COCO category label, we use Grounding DINO (Liu et al. (2024)) to detect all instances of the target object category and use the detected results as box hints to Rex-Thinker.

As shown in Table 4, the CoT-trained model already performs competitively without any task-specific tuning. Further gains are achieved by applying GRPO for post training, demonstrating that reward-driven training enhances the model's ability to generalize beyond the training domain. Interestingly, we find that Rex-Thinker maintains its structured CoT behavior even on novel categories. As illustrated in Figure 4, Rex-Thinker-GRPO successfully generalizes to detect the fish of manta ray (with "fish" bounding boxes as hints) while adhering to its *planning-action-summarization* reasoning paradigm. Notably, the model demonstrates self-correction ability: when provided with an

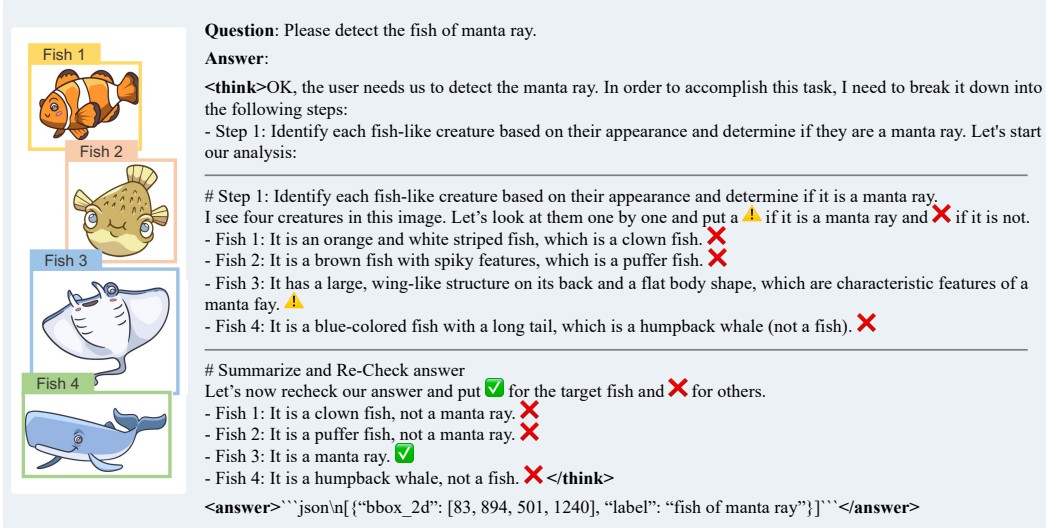

Figure 4: The out-of-domain result. We use Rex-Thinker-GPRO trained on HumanRef-CoT to infer an unseen category (i.e., fish), resulting in a strong generalization. Boxes in the image denote hints.

| Model | RefCOCOg | |
|---|---|---|
| | val | test |
| RexSeek-7B (Jiang et al. (2025b)) | 84.0 | 84.4 |
| Grounding DINO (Liu et al. (2024)) | 86.1 | 87.0 |
| QwenVL-2.5-7B (Bai et al. (2025)) | 87.2 | 87.2 |
| ChatRex-7B (Jiang et al. (2024b)) | 89.8 | 90.0 |
| Rex-Thinker-CoT | 81.2 | 80.3 |
| Rex-Thinker-GRPO | 83.2 | 83.3 |
| Rex-Thinker-GRPO* | 89.2 | 88.8 |

Table 4: Out-of-domain evaluation results on RefCOCOg. *Fine-tuned on RefCOCOg using GRPO.

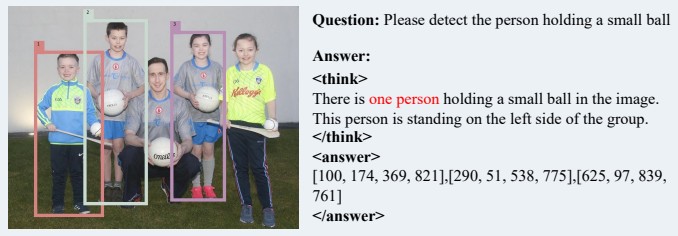

Figure 5: Predictions from a model that was trained with GRPO only, without CoT-based supervised fine-tuning as cold-start initialization. Boxes in the image denote answers.

incorrect hint label (e.g., a whale was incorrectly labeled as a "fish" in hint boxes), Rex-Thinker rectifies the error through logical reasoning and explicitly rejects the misclassification.

To further explore the upper bound of the model, we fine-tune Rex-Thinker-CoT using GRPO directly on RefCOCOg. This leads to additional performance improvements, achieving results comparable to state-of-the-art referring models. The experiment results highlight the adaptability of our reasoning paradigm across domains and the effectiveness of reward-based optimization in extending CoT reasoning to unseen categories.

## 5.4 ABLATIONS

**Necessity of the Two Stage Architecture.** This study directly justifies our two stage detector-reasoner architecture by quantifying the inherent limitations of standalone components for REC. As shown in Table 5, neither open-set detectors nor MLLMs alone achieve robust REC performance. While powerful for open-vocabulary detection, general-purpose detectors like Grounding DINO struggle with the nuanced language comprehension required for complex REC. Even after fine-tuning on HumanRef, its performance remains significantly low (Avg. Precision: 25.7), confirming that standalone detectors are insufficient for the detailed language understanding essential for REC. Conversely, powerful MLLMs like Qwen2.5-VL-7B possess advanced reasoning but struggle with a lower recall rate, caused by pixel-imperfect object localization. For instance, fine-tuned Qwen2.5-VL-7B yields an Avg. Recall of 69.4, a substantial drop compared to that of Rex-Thinker. This demonstrates that while MLLMs are powerful reasoners, their native localization ability is a key bottleneck for REC. Thus, Rex-Thinker rigorously combines the perceptual strength of detectors (for initial candidate generation) with the cognitive strength of MLLMs (for complex reasoning and

| Type | Method | Fine-tuned | With Box Hint | Avg. Precision | Avg. Recall | Avg. DF1 |
|------|--------|-----------|---------------|----------------|-------------|----------|
| Open-set Detector | Grounding DINO | No | No | 15.2 | 87.3 | 9.7 |
| | | Yes | No | 25.7 | 92.0 | 14.8 |
| MLLM | Qwen2.5-VL-7B | No | No | 52.5 | 68.5 | 56.2 |
| | | Yes | No | 74.1 | 69.4 | 69.6 |
| Two Stage (ours) | Rex-Thinker-Plain | Yes | Yes | 85.7 | 82.6 | 81.4 |

Table 5: Limitations of standalone detectors and MLLMs for REC on the HumanRef benchmark, justifying the necessity of our two stage architecture.

verification over these candidates), thereby addressing the individual limitations of the detector and MLLM for robust REC.

**Impact of CoT-based Cold Start on GRPO.** In Rex-Thinker, we adopt a two-stage training strategy where the model is first supervised using CoT-annotated data, followed by GRPO-based reinforcement learning. To assess the importance of this CoT-based initialization, we compare GRPO training with and without the cold-start SFT stage. We find that the model with CoT-based SFT significantly outperforms the direct GRPO model. Specifically, for models trained with GRPO, those initialized with **CoT SFT** (Avg. Precision: 86.8, Avg. Recall: 86.8, Avg. DF1: 83.5, Rejection: 68.2, please refer to the Appendix for details) achieve substantially higher final performance compared to those trained **without CoT SFT** (Avg. Precision: 82.0, Avg. Recall: 81.2, Avg. DF1: 77.8, Rejection: 66.4). This indicates that the initial exposure to structured reasoning patterns provides a more effective starting point for reward-driven learning. Furthermore, as illustrated in Figure 5, models trained without CoT supervision tend to generate unstructured or incoherent reasoning traces, lacking the verifiable and trustworthy qualities we aim to promote. In contrast, CoT-pretrained models produce well-formed thinking steps aligned with our planning, action, and summarization framework.

## 6 Conclusion

We have presented Rex-Thinker, a novel framework that has reformulated the object referring problem as an explicit Chain-of-Thought reasoning process to achieve grounded and interpretable predictions. Unlike conventional approaches that have treated referring as direct bounding box prediction, our model has first detected candidate objects and then performed step-by-step verification against the referring expression through structured planning-action-summarization reasoning. To support this paradigm, we have constructed HumanRef-CoT, a large-scale dataset with reasoning traces that have enabled learning decomposed and interpretable reasoning patterns. Through a two-stage training approach combining SFT and GRPO-based RL, Rex-Thinker has demonstrated superior performance over prior works in both referring accuracy and rejection.

## Acknowledgment

This work was partly supported by the National Natural Science Foundation of China under Grant 62403012.

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

# A APPENDIX

## A.1 USE OF LARGE LANGUAGE MODELS

In preparing this manuscript, we made limited use of a large language model (LLM) to assist with language editing. Specifically, the LLM was employed only to improve grammar, clarity, and wording of sentences. No part of the scientific content, analysis, or claims was generated by the LLM. The authors take full responsibility for all aspects of the paper's content.

## A.2 MORE DETAILS ON CONSTRUCTING HUMANREF-COT

### A.2.1 PROMPT FOR GPT-4O

To annotate HumanRef-CoT dataset using GPT-4o, we designed a two-part prompting strategy that addresses the diverse reasoning requirements across different subsets. This strategy consists of a **unified system prompt** and a set of **subset-specific in-context examples**.

The system prompt is shared across all subsets and instructs the model on how to interpret the input, which includes an image, a referring expression, and candidate bounding boxes. It also defines the expected format of the response, including the use of structured reasoning and answer tags. In addition to the system prompt, each of the six subsets in HumanRef-CoT namely attribute, position, interaction, reasoning, celebrity recognition, and rejection, is paired with a collection of in-context examples. These examples are carefully curated to reflect the specific annotation challenges and reasoning patterns required for each subset. They guide GPT-4o in producing chain-of-thought (CoT) rationales that are consistent with human annotations in both style and logic.

In the following sections, we first present the shared system prompt. Then, for each subset, we provide the corresponding in-context examples and visualization results.

**Unified System Prompt.** The system prompt instructs the model to perform detailed visual reasoning based on either positional or attribute-based referring expressions. It emphasizes step-by-step analysis, beginning with predefined reasoning steps (first attributes, then orientation), and requires the model to explicitly evaluate each candidate object. Special symbols are also used to denote matching, non-matching, and reference entities during analysis.

**Subset-Specific In-Context Examples.** After the system prompt, we provide in-context examples to guide the model toward producing outputs aligned with our CoT structure. These examples help reinforce consistent reasoning patterns. HumanRef-CoT includes six subsets: attribute, position, interaction, reasoning, celebrity recognition, and rejection. Each subset uses its own set of in-context examples tailored to its specific reasoning needs.

We show the in-context prompts used for each subset, along with representative outputs generated by GPT-4o.

| Subset | attribute | position | | interaction | | reasoning | | celebrity | rejection |
|--------|-----------|----------|----------|-------------|----------|-----------|----------|-----------|-----------|
| | - | inner position | outer position | inner interaction | outer interaction | inner positon reasoning | attribute reasoning | - | - |
| Prompt | Figure 34 | Figure 36 | Figure 38 | Figure 40 | Figure 42 | Figure 44 | Figure 46 | Figure 48 | Figure 50 |
| Example | Figure 35 | Figure 37 | Figure 39 | Figure 41 | Figure 43 | Figure 45 | Figure 47 | Figure 49 | Figure 51 |

Table 6: Ablation study on the retrieval-based design of our model. We compare performance with and without box hints to assess their impact on referring accuracy.

### A.2.2 HUMAN EVALUATION OF HUMANREF-COT DATASET QUALITY

To empirically validate the quality of our HumanRef-CoT dataset and assess the effectiveness of our automated filtering pipeline, we conducted a comprehensive manual evaluation study. This study was designed to verify the logical consistency and factual correctness of the generated Chain-of-Thought annotations that were used for model training.

**Methodology**: *1) Data Samlping:* We randomly sampled 600 instances from the final, filtered HumanRef-CoT dataset. To ensure representative coverage, we drew 100 samples from each of the

Given the image and the description below, output a detailed analysis of how you found the object(s) matching the description about position or attribute.

The position description is something like "the third man to the left of the boy wearing a red shirt". Then you first need to find the reference person, i.e. the boy wearing a red shirt, determine for each person of what you see and whether fulfill this reference condition, and put a ⚠️ if he is the reference person or object, and ❌ if he doesn't. If this question is about attribute, you should provide a description of what you see and how each object relates to the provided description, and put a ❌ or ✅ at the end. You first need to do your analysis in a tuple like (analysis)(analysis), and then output your answer in a tuple like (answer)(answer). Note that in your analysis, you need to start by listing your action steps. The action steps must be about attribute first, then orientation. And your first step doesn't need to be to find all the people in the diagram, since I've already provided you with all of them. For example:

```

To find the woman in red dress, I will need to excuse the following steps:
- Step1: Find all woman
- Step2: From the person in step1, I need to find all the person wearing red dress
```

Each step you initially planned must be strictly enforced, and you cannot omit a step or modify the execution of each step.

Note that I will tell you which objects are the ground truth that fit this description, you need to use the answer I give as a reference. But you can't refer to the answer I gave in your answer, pretend that you are thinking about it yourself. I will highlight all such objects as marks in the diagram. Each mark has a circle and a number. The number represents the serial number of the object, and the color of the circle represents whether or not the current object matches the description, with green circles representing objects that match the description, and red circles representing objects that do not match the description. Note that you can't mention this mark in your answer, this mark is just for you to go and specify the corresponding person with the corresponding serial number, you can say person 1, but not mark 1.

Your final answer must be consistent with the analysis, e.g. if you say in the analysis that Person 1 and Person 2 satisfy the condition, then you must also say in your final answer that it is Person 1 and Person 2. In the rare case that I give a reference answer that is incorrect, you need to trust your own judgment.

Additionally I'll calculate the xy coordinates of each mark and then sort them in order from left to right to give you a positional reference if the question is about position. But you can't mention this order, pretend that you figure out the order all by yourself. Every useful information will be provided in METAINFO

INCONTEXT EXAMPLES

Figure 6: The system prompt used to instruct GPT-4o on visual reasoning for HumanRef-CoT. It specifies output format, reasoning steps, symbol conventions, and the expected alignment between intermediate analysis and final answers.

six distinct subsets (Celebrity, Interaction, Position, Reasoning, Attribute, and Rejection) *2) Annotators:* The annotation task was performed by five Ph.D. students with expertise in computer vision and natural language processing. All annotators were given detailed instructions and calibration examples to ensure consistent evaluation criteria. *3)Annotation Interface:* e developed a custom HTML-based annotation interface to facilitate the review process. As shown in Figure 7, the interface presented annotators with the source image (including the Set-of-Mark visual markers), the referring expression, and the complete, generated CoT trace (Planning, Action, and Summarization). This allowed for a holistic review of each data point in its full context.

**Error Categories**: Annotators were instructed to identify three specific types of errors, ordered from most to least severe:

- **Wrong Summarization:** The final set of person indices identified in the <answer> block does not perfectly match the ground-truth labels.
- **Action-Summarization Inconsistency:** A logical contradiction exists between the step-by-step evaluation in the "Action"phase and the final conclusion in the "Summarization" phase. For example, the Action phase concludes a person is a non-match, but the Summarization phase incorrectly includes them.
- **Wrong Action Result:** A specific reasoning step in the "Action" phase is factually incorrect with respect to the visual evidence. For example, the model incorrectly identifies the color of a person's clothing or misjudges a spatial relationship.

**Results and Analysis** The results of our comprehensive human evaluation on the 600 samples are summarized in Table 7. First, our automated, two-stage filtering process is extremely effective at eliminating high-level logical and summarization errors. The human evaluators found zero instances of Wrong Summarization or Action-Summarization Inconsistency. This provides strong evidence that the dataset is reliable in its final conclusions and overall logical structure. Second, we findw

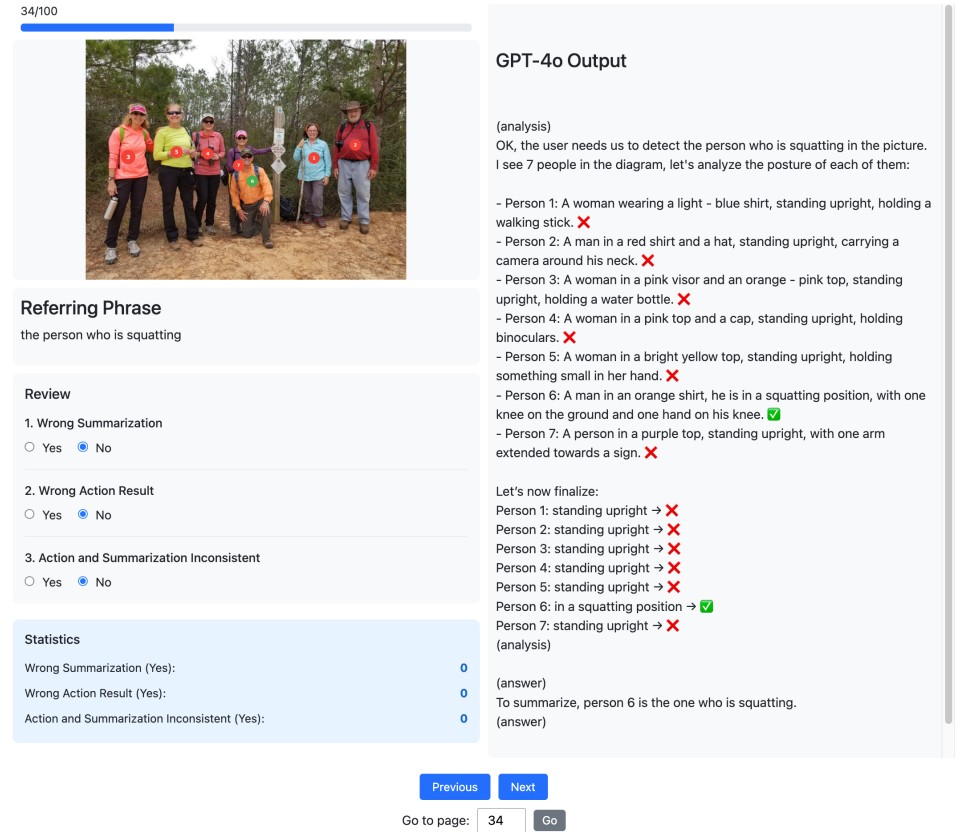

Figure 7: A screenshot of our custom human evaluation interface. The tool displays the image with visual markers, the referring expression, and the full CoT trace, enabling annotators to efficiently verify each reasoning step.

| Error Type | Celebrity | Interaction | Position | Reasoning | Attribute | Rejection |
|---|---|---|---|---|---|---|
| Wrong Summarization | 0 | 0 | 0 | 0 | 0 | 0 |
| Action and Summarization Inconsistent | 0 | 0 | 0 | 0 | 0 | 0 |
| Wrong Action Result | 2 | 2 | 0 | 3 | 0 | 0 |

Table 7: Results of the human evaluation on 600 randomly sampled instances from the HumanRef-CoT dataset. No high-level logical or summarization errors were found.

that a small number of low-level Wrong Action Result errors persist in the final dataset (7 out of 600 total samples, or 1.2%). Upon manual inspection of these cases, we found they typically occur when a referred person is very small, heavily occluded, or when the visual marker itself obscures a key attribute, forcing GPT-4o to make a reasonable but incorrect guess. Despite these minor, infrequent imperfections in intermediate steps, the overall quality of the dataset is very high. The significant performance improvement of our model when trained with this data demonstrates that it provides a valuable and effective training signal for learning grounded, step-by-step reasoning.

### A.2.3 EVALUATE GPT-4O ON HUMANREF

Since we use GPT-4o to annotate HumanRef-CoT, a natural question is how well GPT-4o performs directly on the HumanRef benchmark when prompted in a similar style. To investigate this, we adopt a setup similar to the annotation phase, using the same SoM-style prompt and a set of visual marks (with all marks shown in red). However, we remove any hint indicating which objects are correct. We then evaluate GPT-4o on the HumanRef-Benchmark without prompting with ground-truth answers. As shown in Table 8, GPT-4o achieves an average DF1 score of 53.2 without any hint supervision. This result suggests that while GPT-4o can be used to generate annotations when given

the correct answer as reference, its standalone performance without answer supervision remains limited.

| Method | Attribute | | | Position | | | Interaction | | | Reasoning | | | Celebrity | | | Average | | | Rejection |
|---|---|---|---|---|---|---|---|---|---|---|---|---|---|---|---|---|---|---|---|
| | R | P | DF1 | R | P | DF1 | R | P | DF1 | R | P | DF1 | R | P | DF1 | R | P | DF1 | Score |
| GPT-4o-CoT | 50.2 | 56.2 | 50.9 | 56.1 | 56.8 | 55.1 | 52.8 | 56.8 | 53.2 | 53.3 | 52.9 | 51.1 | 54.9 | 54.3 | 53.2 | 54.3 | 55.2 | 53.2 | 14.8 |
| Rex-Thinker-GRPO | 88.5 | 88.7 | 84.1 | 87.2 | 87.1 | 84.6 | 81.5 | 83.5 | 79.1 | 87.7 | 85.4 | 82.3 | 88.0 | 89.3 | 87.2 | 86.6 | 86.8 | 83.5 | 68.2 |

Table 8: Evaluation of GPT-4o on the HumanRef-Benchmark test set using SoM-style prompts without answer hints. The model achieves 53.2 average DF1 score, indicating limited standalone performance.

### A.2.4 ETHICAL CONSIDERATIONS AND DATASET AVAILABILITY

This section details the ethical safeguards employed during the creation of the HumanRef-CoT dataset and outlines our commitment to its public release to the research community.

**Ethical Considerations and Safeguards** We acknowledge the critical importance of responsible and ethical data creation. Our data generation pipeline was designed with a multi-layered approach to mitigate the risk of generating biased or problematic content.

- **Filtered Image Source:** The source images for HumanRef-CoT are from the public HumanRef dataset. These images have undergone prior NSFW (Not Safe For Work) filtering, ensuring that the visual content is appropriate and does not contain sensitive material.
- **Use of a Moderated Large Language Model:** Our data generation process utilizes OpenAI's GPT-4o to produce the reasoning traces. As a state-of-the-art commercial model, GPT-4o is subject to rigorous safety protocols and content moderation filters developed by its provider. These built-in guardrails are designed to prevent the generation of offensive, biased, or otherwise harmful content and served as a primary safeguard in our pipeline.
- **Final Data Review:** Throughout our automated and manual quality control stages (detailed in Appendix A.5), where we verified the logical and factual correctness of the reasoning traces, we also remained vigilant for any inappropriate content. We can confirm that we did not encounter any instances of ethically questionable language in the final, curated HumanRef-CoT dataset.

**Dataset Availability** To ensure full reproducibility and to foster future research in grounded reasoning and model interpretability, we are committed to making our dataset publicly available. The HumanRef-CoT dataset will be released upon the publication of this work.

### A.3 EXPERIMENT DETAILS

### A.3.1 CoT SFT SETTINGS

Table 9 summarizes the full training hyperparameters and computational cost used during the CoT SFT stage. These settings were applied in the cold-start phase without prior instruction tuning.

| batch size | 4 | maximum gradient norm | 1 | precision | bf16 |
|---|---|---|---|---|---|
| gradient accumulation | 4 | learning rate scheduler | cosine | epochs | 2 |
| learning rate | 2e-5 | max length | 2048 | times | 10.1h |
| optimizer | AdamW | deepspeed | zero3 | GPU | 8xA100 |
| warm up ratio | 0.03 | weight decay | 0.01 | trainable module | LLM |

Table 9: Training settings and cost statistics for CoT SFT.

### A.3.2 GRPO SETTINGS

We provide the training configurations used during the GRPO stage in Table 15. We did not run full GRPO training on the entire HumanRef-CoT dataset. Instead, training was terminated when the reward signal plateaued, indicating convergence.

### A.3.3 GRPO TRAINING ANALYSIS

We analyze the training logs of the GRPO stage. As shown in Figure 8, we visualize the changes in both reward signals and completion length throughout training.

| batch size | 8 | num of rollout | 8 | precision | bf16 |
|---|---|---|---|---|---|
| gradient accumulation | 2 | $\beta$ | 0.04 | epochs | 0.25 |
| learning rate | 1e-6 | temperature | 1.0 | times | 112h |
| optimizer | AdamW | deepspeed | zero3 | GPU | 8xA100 |
| warm up ratio | 0.03 | weight decay | 0.01 | trainable module | LLM |

Table 10: Hyperparameters used during the GRPO training stage.

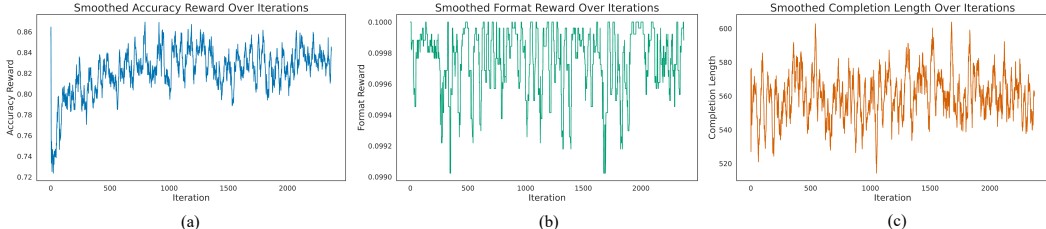

(a)                                    (b)                                    (c)

Figure 8: GRPO training curves showing accuracy reward, format reward, and completion length over time.

Thanks to the cold-start CoT initialization, the model achieves a reasonably high accuracy reward at the beginning of GRPO training. At the same time, the format reward is nearly saturated from the start, indicating that the model has already learned to follow the correct output structure after CoT supervision. Meanwhile, the completion length remains stable at around 560 tokens throughout training. We attribute this to the model having already acquired the basic reasoning skills required for the referring task during the CoT fine-tuning phase, resulting in consistent output lengths with minimal fluctuation.

### A.3.4 DETAILED EVALUATION SETTINGS

This section provides a detailed breakdown of the experimental settings and evaluation contexts for all baseline models.

**In-domain Evaluation Settings:** For the in-domain evaluation on the HumanRef benchmark, as presented in Table 3 of the main paper, the baseline models fall into two distinct categories:

- **Zero-Shot Evaluation:** The majority of the compared models (e.g., DINOX, InternVL-2.5-8B, Ferret-7B, Groma-7B, ChatRex-7B, Qwen2.5-VL-7B, etc.) were evaluated in a zero-shot setting. We used their publicly available, official checkpoints to generate predictions directly on the HumanRef test set without any fine-tuning on our dataset. This setup is designed to measure their out-of-the-box generalization capability to our challenging, human-centric scenarios.
- **State-of-the-Art In-Domain Model:** The RexSeek-7B model is the only baseline that was specifically trained on the HumanRef training set. Crucially, it was not trained using any form of Chain-of-Thought (CoT) reasoning, as it is a direct-prediction model. We report the performance numbers directly from the original RexSeek paper, as this represents the current state-of-the-art for non-CoT methods on this benchmark.

**Out-of-domain Evaluation Settings**: For the out-of-domain evaluation on the RefCOCOg dataset, as presented in Table 4 of the main paper, the comparsion settings are:

- **Baseline Models (Supervised):** The reported numbers for all baseline models (RexSeek-7B, Grounding DINO, QwenVL-2.5-7B, ChatRex-7B) are taken directly from their original papers. As per their publications, all of these models were trained on the RefCOCOg training set. Furthermore, none of them are based on a CoT framework. They represent the state-of-the-art for direct-prediction or retrieval-based methods on this standard benchmark.
- **Our Model (Zero-Shot):** It is critical to note that our Rex-Thinker models are evaluated in a strict zero-shot setting on this dataset. They were trained only on the HumanRef-CoT dataset and had no exposure to RefCOCOg data or its object categories during training. This setup specifically tests the generalization of our learned reasoning framework.

### A.3.5 DETAILED METRICS FOR ABLATION STUDIES

As shown in Table 11, the without-hint model is naive Qwen2.5-VL-7B, while the with-hint model is Rex-Thinker-Plain. It is seen that the box hint plays an important role in REC accuracy. As shown in Table 12, CoT-based SFT as a cold start is an important pre-procedure for RL exploration, leading to significant improvement in REC accuracy and rejection. This is also evidence that our planning-action-summartization paradigm is helpful for REC.

| With Box Hint | Attribute | | | Position | | | Interaction | | | Reasoning | | | Celebrity | | | Average | | | Rejection Score |
|---|---|---|---|---|---|---|---|---|---|---|---|---|---|---|---|---|---|---|---|
| | R | P | DF1 | R | P | DF1 | R | P | DF1 | R | P | DF1 | R | P | DF1 | R | P | DF1 | |
| No | 66.4 | 74.3 | 67.2 | 69.3 | 71.9 | 69.5 | 65.2 | 72.1 | 66.4 | 63.6 | 67.5 | 62.2 | 82.4 | 84.6 | 82.7 | 69.4 | 74.1 | 69.6 | **71.7** |
| Yes | **83.0** | **88.7** | **81.4** | **82.5** | **83.9** | **81.3** | **80.1** | **85.6** | **80.2** | **80.5** | **82.2** | **77.3** | **86.7** | **88.7** | **86.8** | **82.6** | **85.8** | **81.4** | 53.5 |

Table 11: Ablation study on the retrieval-based design of our model. We compare performance with and without box hints to assess their impact on referring accuracy.

| With Cold Start | Attribute | | | Position | | | Interaction | | | Reasoning | | | Celebrity | | | Average | | | Rejection Score |
|---|---|---|---|---|---|---|---|---|---|---|---|---|---|---|---|---|---|---|---|
| | R | P | DF1 | R | P | DF1 | R | P | DF1 | R | P | DF1 | R | P | DF1 | R | P | DF1 | |
| No | 81.4 | 85.8 | 78.1 | 80.2 | 80.2 | 77.5 | 79.6 | 82.6 | 78.0 | 77.6 | 75.0 | 70.6 | 87.3 | 86.5 | 84.8 | 81.2 | 82.0 | 77.8 | 66.4 |
| Yes | **88.5** | **88.7** | **84.1** | **87.2** | **87.1** | **84.6** | **81.5** | **83.5** | **79.1** | **87.7** | **85.4** | **82.3** | **88.0** | **89.3** | **87.2** | **86.6** | **86.8** | **83.5** | **68.2** |

Table 12: Ablation on the impact of CoT-based cold start on final performance after GRPO training.

### A.3.6 COMPARISON WITH THE THINK-WITH-IMAGE PARADIGM

| Method | Attribute | | | Position | | | Interaction | | | Reasoning | | | Celebrity | | | Average | | | Rejection Score |
|---|---|---|---|---|---|---|---|---|---|---|---|---|---|---|---|---|---|---|---|
| | R | P | DF1 | R | P | DF1 | R | P | DF1 | R | P | DF1 | R | P | DF1 | R | P | DF1 | |
| Qwen2.5-VL-7B | 49.1 | 71.3 | 54.4 | 50.2 | 61.7 | 52.8 | 48.2 | 66.3 | 53.2 | 34.6 | 61.2 | 40.3 | 80.3 | 81.9 | 80.1 | 52.5 | 68.5 | 56.2 | 7.1 |
| DeepEyes-7B | 36.7 | 60.1 | 41.6 | 34.6 | 43.7 | 36.1 | 40.2 | 56.9 | 44.6 | 28.4 | 40.7 | 30.8 | 40.2 | 40.9 | 40.2 | 36.0 | 48.4 | 38.7 | 27.3 |
| Rex-Thinker-GRPO | 88.5 | 88.8 | 84.1 | 87.3 | 87.8 | 85.1 | 82.1 | 83.6 | 79.4 | 87.0 | 84.3 | 81.3 | 88.6 | 90.4 | 88.1 | 86.7 | 87.0 | 83.6 | 67.8 |

Table 13: Detailed performance comparison on the HumanRef benchmark. Our Rex-Thinker-GRPO significantly outperforms DeepEyes, a representative "Think-with-image" model.

We evaluated DeepEyes-7B, a representative model from the "Think-with-image" paradigm, on the HumanRef benchmark. DeepEyes is designed for complex visual reasoning, where its Chain-of-Thought process can invoke tools to interactively modify the image, such as by zooming in or cropping specific regions to gather more detailed information. The detailed results of our comparison are presented in Table 13.

The quantitative results reveal a significant performance gap. Our Rex-Thinker-GRPO outperforms DeepEyes by a large margin across all categories, most notably in Average Recall (86.7 vs. 36.0) and Average DF1 (83.6 vs. 38.7). This disparity points to a fundamental limitation in applying a general visual reasoning paradigm to the specific problem of REC.

Our investigation into the performance gap reveals that it stems from a core methodological difference between our framework and the "Think-with-image" approach. As illustrated in Figure 9, DeepEyes employ a holistic, tool-augmented reasoning strategy. The model identifies a few salient regions of interest and may invoke tools (e.g., cropping, zooming) to inspect these specific regions in greater detail before generating an answer. While this interactive approach is powerful for general visual reasoning tasks, it is ill-suited for REC, especially for multi-instance referring. We observed that after investigating only one promising candidates, the model frequently concludes its reasoning process prematurely, leading to a high number of false negatives and thus very low recall.

### A.3.7 FINE-GRAINED PERFORMANCE ANALYSIS

To provide a more nuanced understanding of our model's performance and the specific impact of the Chain-of-Thought framework, we conducted a fine-grained analysis on the HumanRef benchmark. This analysis compares our full model, Rex-Thinker-GRPO, against a baseline version trained with supervised fine-tuning but without the structured CoT data (Rex-Thinker-Plain)

The primary finding of this analysis, detailed in Table 14, is that the CoT framework provides a substantial and consistent boost in performance, particularly in Recall. This effect is most pronounced in the benchmark's most complex categories, such as Reasoning (+5.29 Recall) and Attribute (+4.48 Recall).

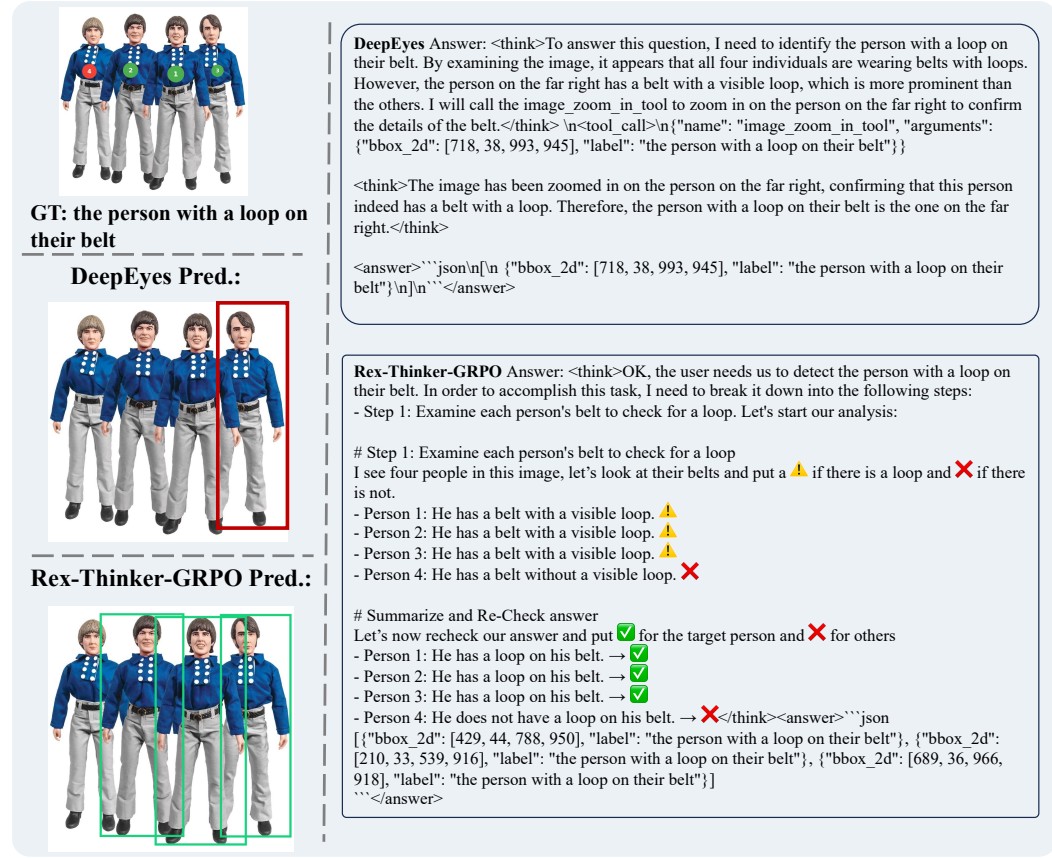

Figure 9: Qualitative comparison of Chain-of-Thought styles between DeepEyes and Rex-Thinker-GRPO. DeepEyes uses a holistic, tool-augmented approach, identifying only one of multiple targets after interactively inspecting it. Our model uses a systematic, exhaustive framework, evaluating each candidate to correctly identify all matching targets.

| Method | Attribute | | Position | | Interaction | | Reasoning | | Celebrity | |
|---|---|---|---|---|---|---|---|---|---|---|
| | R | P | R | P | R | P | R | P | R | P |
| Rex-Thinker-Plain | 84.17 | 89.89 | 85.48 | 87.18 | 82.16 | 87.80 | 82.63 | 85.34 | 88.58 | 90.58 |
| Rex-Thinker-GRPO | 88.65 | 91.98 | 87.42 | 88.68 | 81.95 | 88.11 | 87.92 | 87.71 | 89.41 | 91.30 |
| | (+4.48) | (+2.09) | (+1.94) | (+1.50) | (-0.21) | (+0.31) | (+5.29) | (+2.37) | (+1.13) | (+0.72) |

Table 14: Detailed performance comparison between our full model (Rex-Thinker-GRPO) and a non-CoT baseline (Rex-Thinker-Plain) across HumanRef subsets. The results highlight that the CoT framework significantly improves recall.

This improvement in recall stems directly from our core design. The structured Planning-Action-Summarization process forces the model to systematically evaluate each candidate box against the expression's sub-goals. Unlike methods that might holistically reason and identify only the most salient object, our approach's step-by-step verification minimizes the chance of overlooking a potential target. This inherently reduces false negatives (missed detections) and leads to significantly higher recall, as illustrated in the qualitative examples in Figure 10, Figure 11

This detailed analysis also allows us to transparently identify a key limitation of our current model. The "Interaction" category is the only subset where recall experiences a minor decrease (-0.21%). Upon manual inspection of failure cases (e.g., Figure 12), we hypothesize this is due to the inherent difficulty of grounding complex, multi-entity relationships in crowded scenes with significant occlusion. Interaction" expressions (e.g., "the person talking to the woman in red") require the model to reason about a relationship between at least two individuals. In crowded scenes, these individuals may be physically close, overlapping, or occluding each other. This can lead to the initial candidate boxes being imprecise, with a single box ambiguously containing parts of multiple people. Our

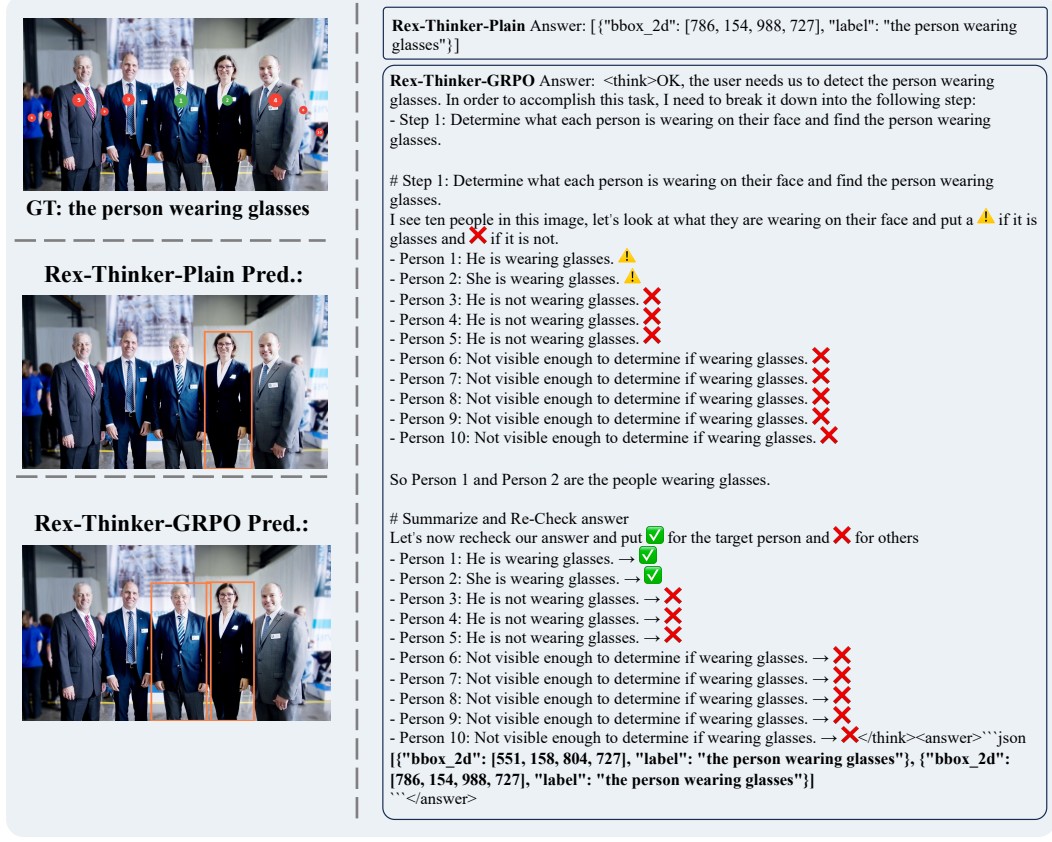

Figure 10: Comparsion betwee Rex-Thinker-GRPO and Rex-Thinker-Plain. Chain-of-Thought improves attribute-based referring.

current CoT framework evaluates one candidate region at a time. When presented with a heavily occluded or merged bounding box, the model struggles to definitively confirm the "interaction" based on the features within that single region. This can lead the model to conservatively (and sometimes incorrectly) reject the candidate, thus slightly lowering the recall rate for this specific, challenging scenario.

## A.4 LIMITATIONS AND BROADER IMPACTS

### A.4.1 INFERENCE SPEED

While the CoT-based design improves both interpretability and performance, it also introduces additional computational overhead at inference time. To quantify this, we randomly selected 100 images from the HumanRef-Benchmark test set and compared the average inference time per image between RexThinker-Plain and RexThinker-GRPO.

All experiments were conducted using the vLLM framework on a single NVIDIA A100 GPU. As shown in Table 3, RexThinker-GRPO exhibits slower inference due to its longer CoT-style outputs. This observation aligns with the general principle of test-time computation, where improved interpretability and accuracy often come at the cost of slower response time.

| model | Rex-Thinker-Plain | Rex-Thinker-GRPO |
|---|---|---|
| average inference time | 1.13s | 6.68s |

Table 15: Comparison of average inference time between RexThinker-Plain and RexThinker-GRPO. The CoT design in GRPO leads to slower inference.

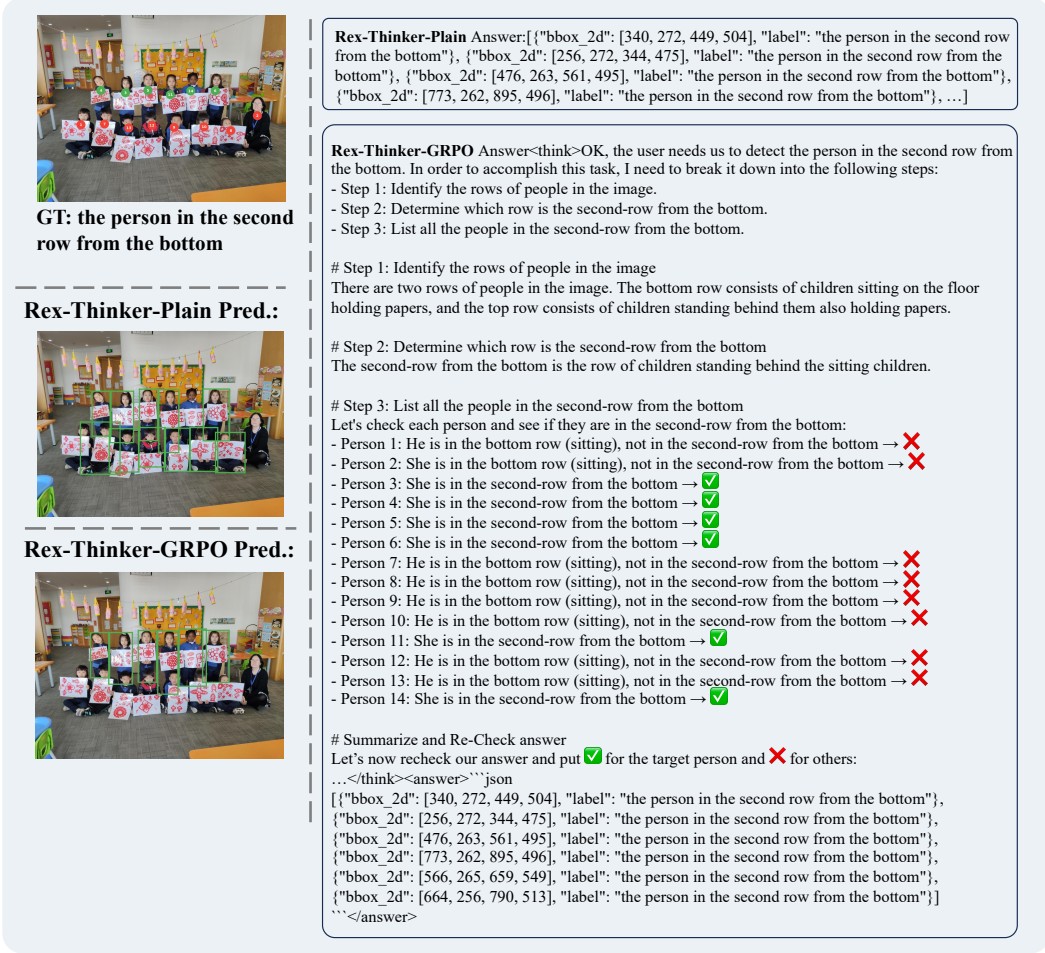

Figure 11: Comparsion betwee Rex-Thinker-GRPO and Rex-Thinker-Plain. Chain-of-Thought improves position-based referring.

### A.4.2 INCONSISTENT REASONING

We observe occasional inconsistencies between the model's reasoning process and its final output. For example, as shown in Figure 13, the model identifies nine candidate objects in the reasoning phase but only includes eight in the final predicted coordinates.

We attribute such inconsistencies to the lack of explicit supervision enforcing alignment between the reasoning chain and the final answer during GRPO training. While most predictions remain consistent, these rare cases highlight a potential gap in our current framework. One possible solution is to introduce a consistency reward, which evaluates whether the number of objects summarized in the reasoning matches the number of bounding boxes in the final answer. This direction remains open for future exploration.

### A.4.3 BROADER IMPACTS

Referring models enable more flexible and natural interaction with visual scenes compared to traditional object detection, as they can understand complex, context-dependent language. This makes them valuable in applications such as surveillance, smart cities, and smart homes.

However, these models also raise concerns. In particular, they may expose privacy risks when deployed in sensitive environments and can inherit biases from training data or user input. Although CoT reasoning improves interpretability, ensuring its consistency and robustness remains an open challenge. Future work should address these risks to ensure safe and responsible deployment.

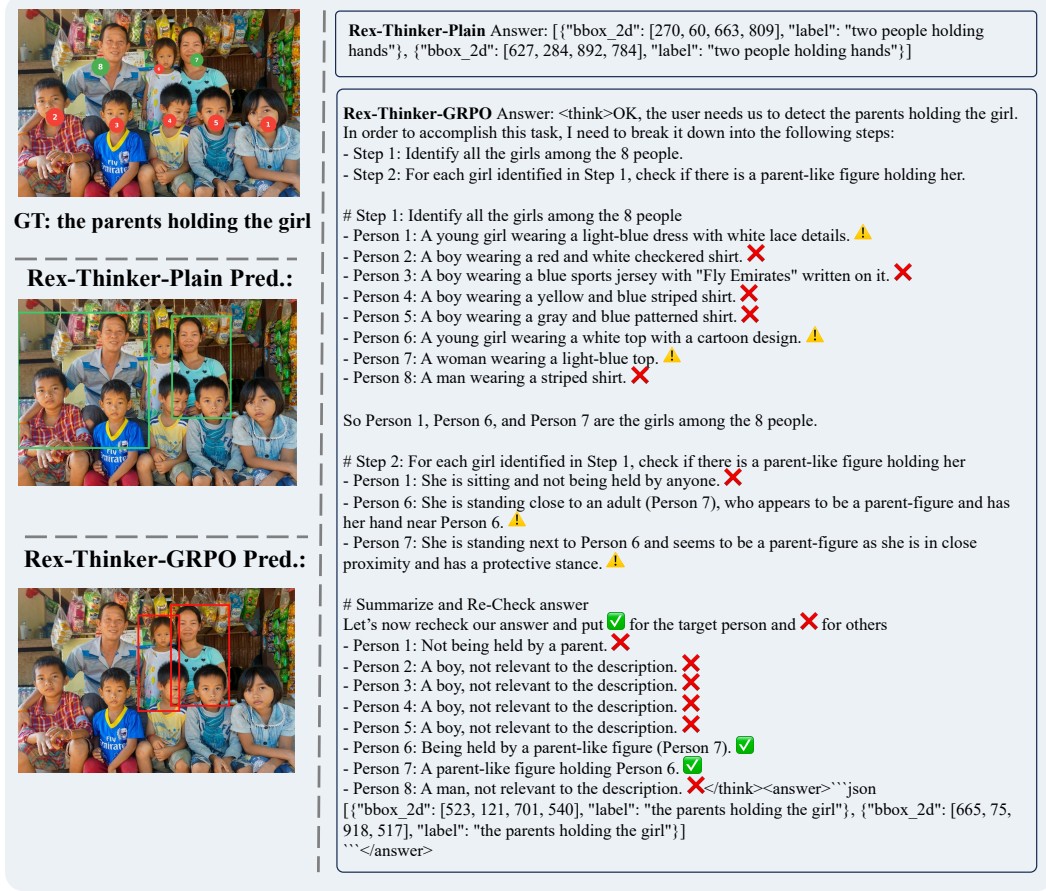

Figure 12: A failure case of Rex-Thinker-GRPO, where CoT leads to a wrong answer.

## A.5 VISUALIZATION RESULTS

In this section, we present qualitative results of the RexThinker model across different scenarios, as shown in Figures 14–33. All visualizations are generated by the RexThinker-GRPO model, which is trained solely on the HumanRef-CoT dataset.

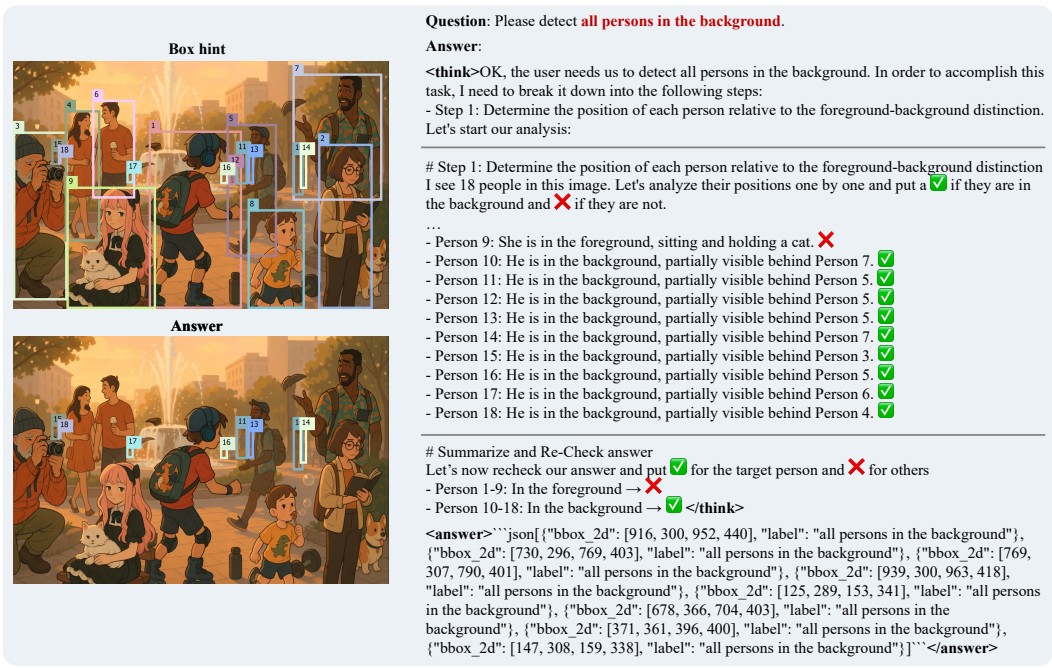

Figure 13: Example of reasoning–answer mismatch. The number of predicted objects differs between reasoning and the final output.

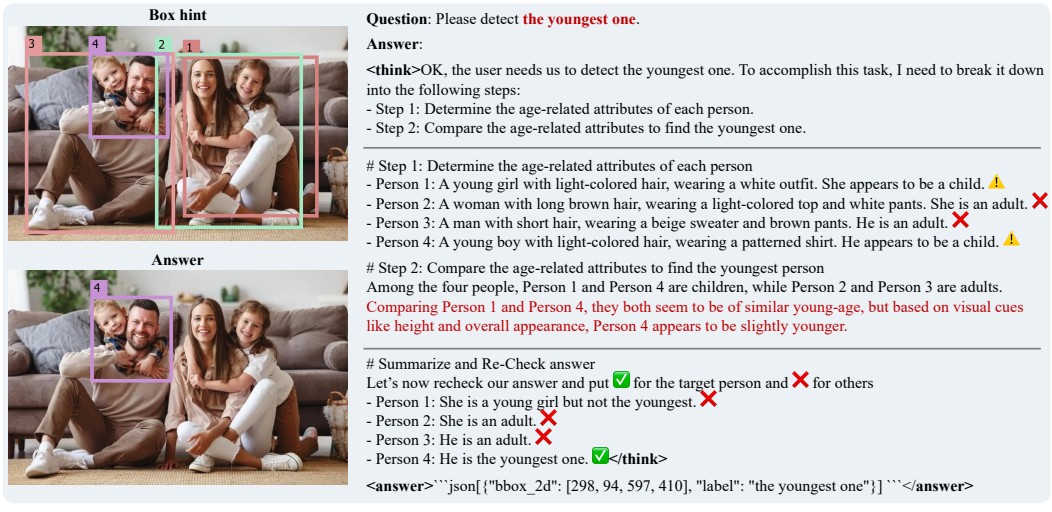

Figure 14: Attribute referring example.

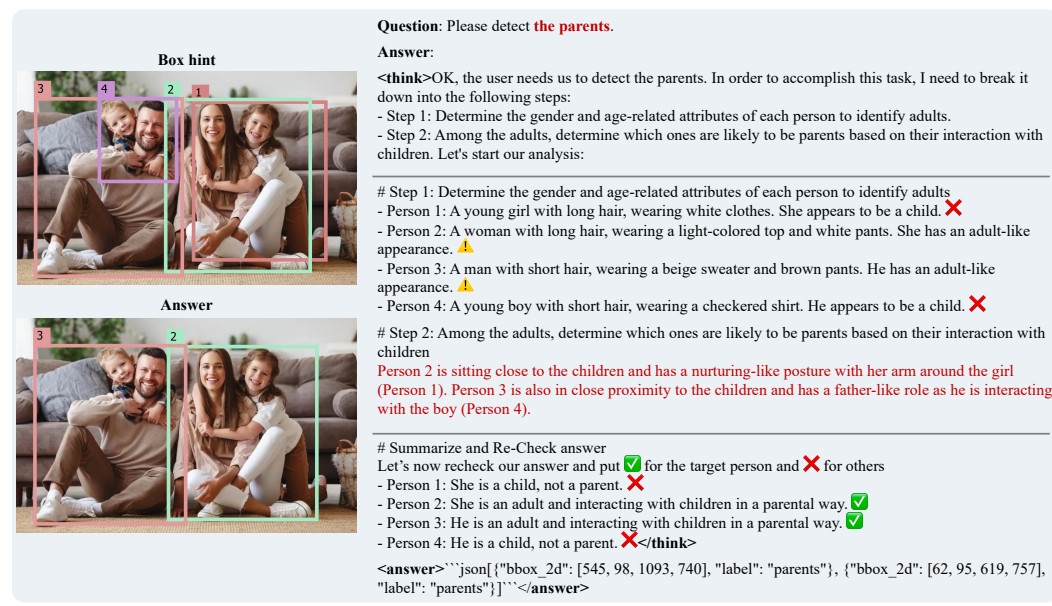

**Question**: Please detect **the parents**.

**Answer**:

**<think>**OK, the user needs us to detect the parents. In order to accomplish this task, I need to break it down into the following steps:
- Step 1: Determine the gender and age-related attributes of each person to identify adults.
- Step 2: Among the adults, determine which ones are likely to be parents based on their interaction with children. Let's start our analysis:

# Step 1: Determine the gender and age-related attributes of each person to identify adults
- Person 1: A young girl with long hair, wearing white clothes. She appears to be a child. ❌
- Person 2: A woman with long hair, wearing a light-colored top and white pants. She has an adult-like appearance. ⚠️
- Person 3: A man with short hair, wearing a beige sweater and brown pants. He has an adult-like appearance. ⚠️
- Person 4: A young boy with short hair, wearing a checkered shirt. He appears to be a child. ❌

# Step 2: Among the adults, determine which ones are likely to be parents based on their interaction with children
Person 2 is sitting close to the children and has a nurturing-like posture with her arm around the girl (Person 1). Person 3 is also in close proximity to the children and has a father-like role as he is interacting with the boy (Person 4).

# Summarize and Re-Check answer
Let's now recheck our answer and put ✅ for the target person and ❌ for others
- Person 1: She is a child, not a parent. ❌
- Person 2: She is an adult and interacting with children in a parental way. ✅
- Person 3: He is an adult and interacting with children in a parental way. ✅
- Person 4: He is a child, not a parent. ❌**</think>**

**<answer>**```json[{"bbox_2d": [545, 98, 1093, 740], "label": "parents"}, {"bbox_2d": [62, 95, 619, 757], "label": "parents"}]```**</answer>**

Figure 15: Attribute referring example.

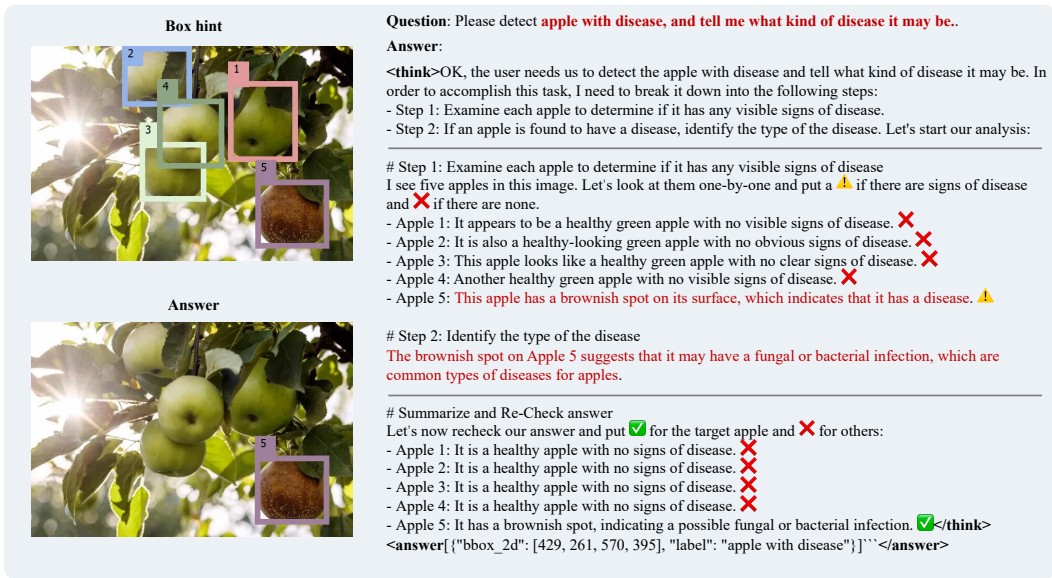

**Question**: Please detect **apple with disease, and tell me what kind of disease it may be.**.

**Answer**:

**<think>**OK, the user needs us to detect the apple with disease and tell what kind of disease it may be. In order to accomplish this task, I need to break it down into the following steps:
- Step 1: Examine each apple to determine if it has any visible signs of disease.
- Step 2: If an apple is found to have a disease, identify the type of the disease. Let's start our analysis:

# Step 1: Examine each apple to determine if it has any visible signs of disease
I see five apples in this image. Let's look at them one-by-one and put a ⚠️ if there are signs of disease and ❌ if there are none.
- Apple 1: It appears to be a healthy green apple with no visible signs of disease. ❌
- Apple 2: It is also a healthy-looking green apple with no obvious signs of disease. ❌
- Apple 3: This apple looks like a healthy green apple with no clear signs of disease. ❌
- Apple 4: Another healthy green apple with no visible signs of disease. ❌
- Apple 5: This apple has a brownish spot on its surface, which indicates that it has a disease. ⚠️

# Step 2: Identify the type of the disease
The brownish spot on Apple 5 suggests that it may have a fungal or bacterial infection, which are common types of diseases for apples.

# Summarize and Re-Check answer
Let's now recheck our answer and put ✅ for the target apple and ❌ for others:
- Apple 1: It is a healthy apple with no signs of disease. ❌
- Apple 2: It is a healthy apple with no signs of disease. ❌
- Apple 3: It is a healthy apple with no signs of disease. ❌
- Apple 4: It is a healthy apple with no signs of disease. ❌
- Apple 5: It has a brownish spot, indicating a possible fungal or bacterial infection. ✅**</think>**
**<answer>**[{"bbox_2d": [429, 261, 570, 395], "label": "apple with disease"}]```**</answer>**

Figure 16: Attribute referring example.

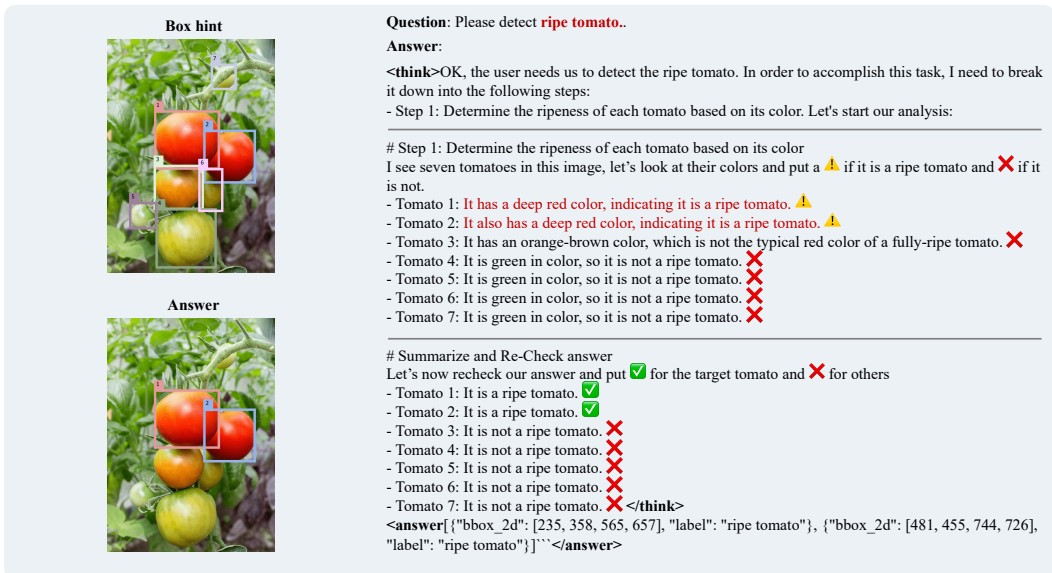

**Box hint**

**Answer**

**Question**: Please detect **ripe tomato.**.

**Answer**:

**<think>**OK, the user needs us to detect the ripe tomato. In order to accomplish this task, I need to break it down into the following steps:
- Step 1: Determine the ripeness of each tomato based on its color. Let's start our analysis:

# Step 1: Determine the ripeness of each tomato based on its color
I see seven tomatoes in this image, let's look at their colors and put a ⚠️ if it is a ripe tomato and ❌ if it is not.
- Tomato 1: It has a deep red color, indicating it is a ripe tomato. ⚠️
- Tomato 2: It also has a deep red color, indicating it is a ripe tomato. ⚠️
- Tomato 3: It has an orange-brown color, which is not the typical red color of a fully-ripe tomato. ❌
- Tomato 4: It is green in color, so it is not a ripe tomato. ❌
- Tomato 5: It is green in color, so it is not a ripe tomato. ❌
- Tomato 6: It is green in color, so it is not a ripe tomato. ❌
- Tomato 7: It is green in color, so it is not a ripe tomato. ❌

# Summarize and Re-Check answer
Let's now recheck our answer and put ✅ for the target tomato and ❌ for others
- Tomato 1: It is a ripe tomato. ✅
- Tomato 2: It is a ripe tomato. ✅
- Tomato 3: It is not a ripe tomato. ❌
- Tomato 4: It is not a ripe tomato. ❌
- Tomato 5: It is not a ripe tomato. ❌
- Tomato 6: It is not a ripe tomato. ❌
- Tomato 7: It is not a ripe tomato. ❌ **</think>**
**<answer>**[{"bbox_2d": [235, 358, 565, 657], "label": "ripe tomato"}, {"bbox_2d": [481, 455, 744, 726], "label": "ripe tomato"}]```**</answer>**

Figure 17: Attribute referring example.

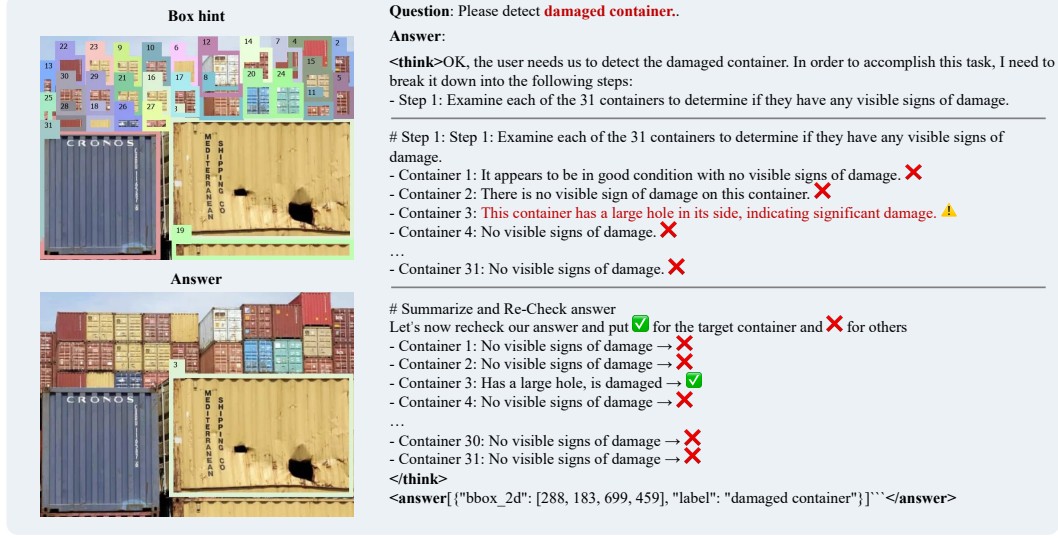

**Box hint**

**Answer**

**Question**: Please detect **damaged container.**.

**Answer**:

**<think>**OK, the user needs us to detect the damaged container. In order to accomplish this task, I need to break it down into the following steps:
- Step 1: Examine each of the 31 containers to determine if they have any visible signs of damage.

# Step 1: Step 1: Examine each of the 31 containers to determine if they have any visible signs of damage.
- Container 1: It appears to be in good condition with no visible signs of damage. ❌
- Container 2: There is no visible sign of damage on this container. ❌
- Container 3: This container has a large hole in its side, indicating significant damage. ⚠️
- Container 4: No visible signs of damage. ❌
…
- Container 31: No visible signs of damage. ❌

# Summarize and Re-Check answer
Let's now recheck our answer and put ✅ for the target container and ❌ for others
- Container 1: No visible signs of damage → ❌
- Container 2: No visible signs of damage → ❌
- Container 3: Has a large hole, is damaged → ✅
- Container 4: No visible signs of damage → ❌
…
- Container 30: No visible signs of damage → ❌
- Container 31: No visible signs of damage → ❌
**</think>**
**<answer>**[{"bbox_2d": [288, 183, 699, 459], "label": "damaged container"}]```**</answer>**

Figure 18: Attribute referring example.

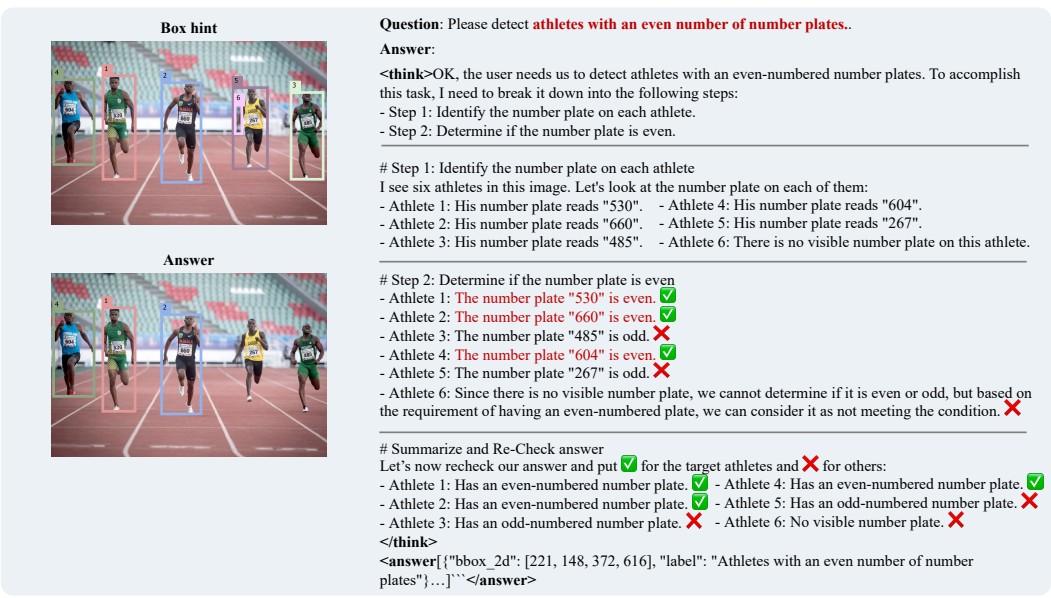

Figure 19: Attribute referring example.

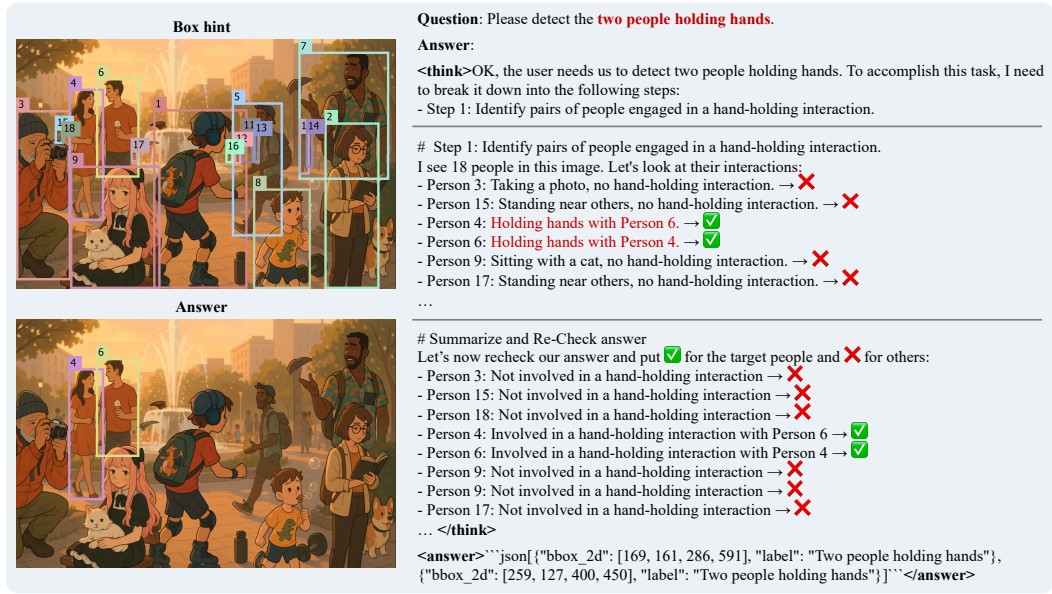

Figure 20: Interaction referring example.

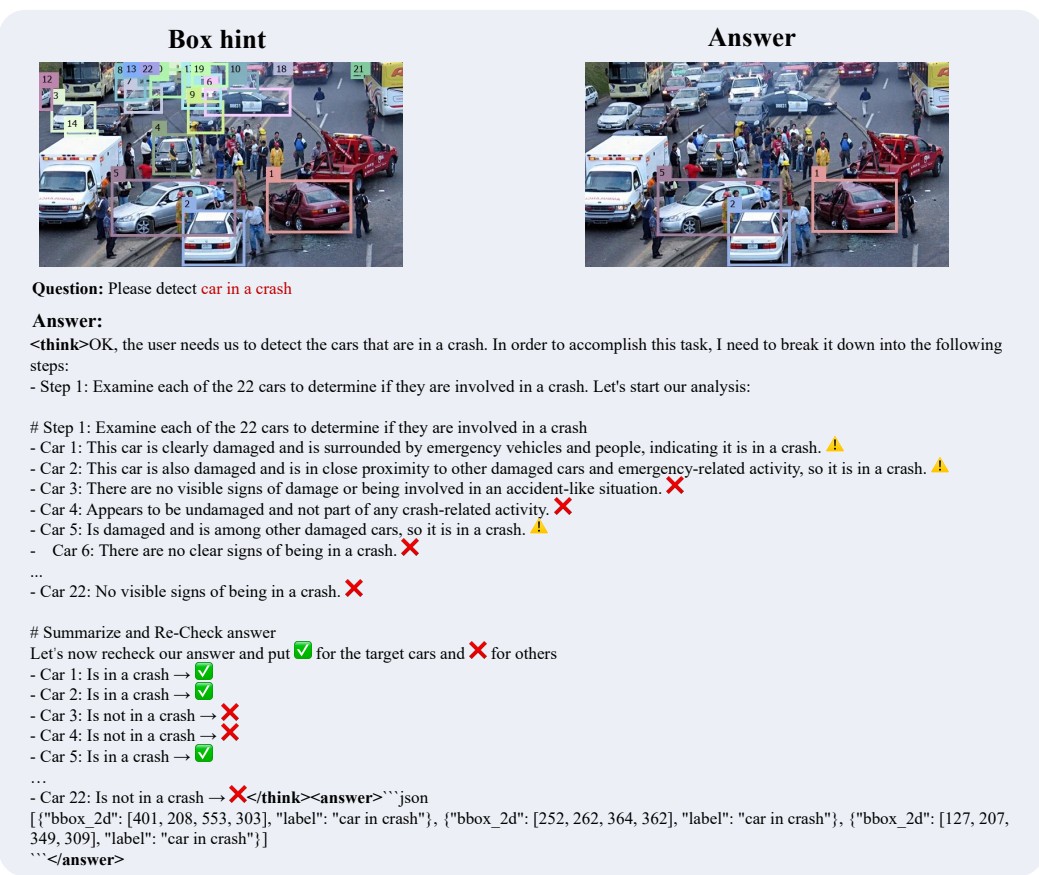

**Box hint**

**Answer**

**Question:** Please detect car in a crash

**Answer:**

**<think>**OK, the user needs us to detect the cars that are in a crash. In order to accomplish this task, I need to break it down into the following steps:
- Step 1: Examine each of the 22 cars to determine if they are involved in a crash. Let's start our analysis:

# Step 1: Examine each of the 22 cars to determine if they are involved in a crash
- Car 1: This car is clearly damaged and is surrounded by emergency vehicles and people, indicating it is in a crash. ⚠️
- Car 2: This car is also damaged and is in close proximity to other damaged cars and emergency-related activity, so it is in a crash. ⚠️
- Car 3: There are no visible signs of damage or being involved in an accident-like situation. ❌
- Car 4: Appears to be undamaged and not part of any crash-related activity. ❌
- Car 5: Is damaged and is among other damaged cars, so it is in a crash. ⚠️
-   Car 6: There are no clear signs of being in a crash. ❌
...
- Car 22: No visible signs of being in a crash. ❌

# Summarize and Re-Check answer
Let's now recheck our answer and put ✅ for the target cars and ❌ for others
- Car 1: Is in a crash → ✅
- Car 2: Is in a crash → ✅
- Car 3: Is not in a crash → ❌
- Car 4: Is not in a crash → ❌
- Car 5: Is in a crash → ✅
…
- Car 22: Is not in a crash → ❌**</think><answer>**```json
[{"bbox_2d": [401, 208, 553, 303], "label": "car in crash"}, {"bbox_2d": [252, 262, 364, 362], "label": "car in crash"}, {"bbox_2d": [127, 207, 349, 309], "label": "car in crash"}]
```**</answer>**

Figure 21: Interaction referring example.

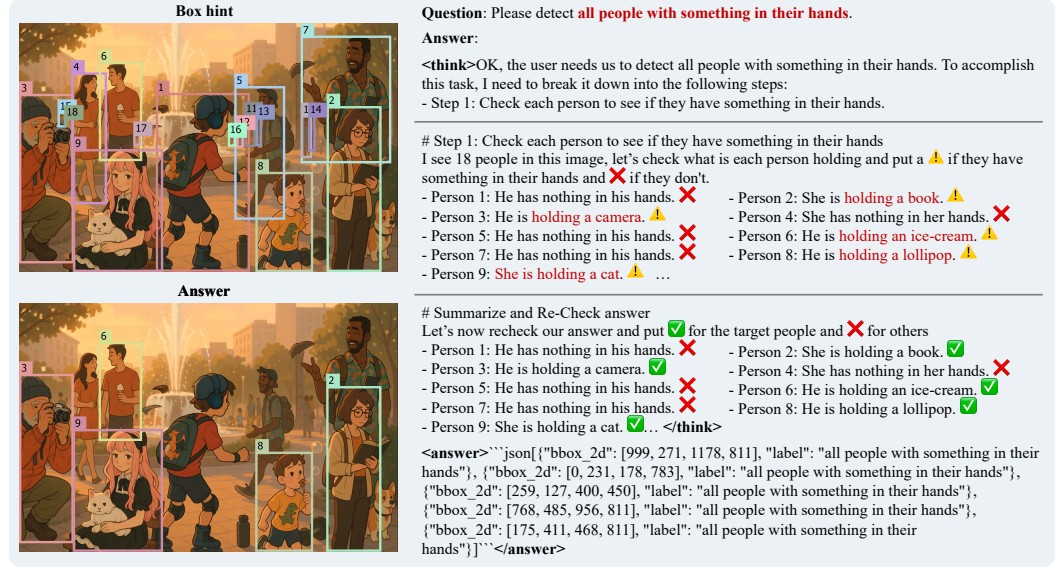

**Box hint**

**Question**: Please detect **all people with something in their hands**.

**Answer**:

**<think>**OK, the user needs us to detect all people with something in their hands. To accomplish this task, I need to break it down into the following steps:
- Step 1: Check each person to see if they have something in their hands.

# Step 1: Check each person to see if they have something in their hands
I see 18 people in this image, let's check what is each person holding and put a ⚠️ if they have something in their hands and ❌ if they don't.
- Person 1: He has nothing in his hands. ❌          - Person 2: She is holding a book. ⚠️
- Person 3: He is holding a camera. ⚠️          - Person 4: She has nothing in her hands. ❌
- Person 5: He has nothing in his hands. ❌          - Person 6: He is holding an ice-cream. ⚠️
- Person 7: He has nothing in his hands. ❌          - Person 8: He is holding a lollipop. ⚠️
- Person 9: She is holding a cat. ⚠️   …

**Answer**

# Summarize and Re-Check answer
Let's now recheck our answer and put ✅ for the target people and ❌ for others
- Person 1: He has nothing in his hands. ❌          - Person 2: She is holding a book. ✅
- Person 3: He is holding a camera. ✅          - Person 4: She has nothing in her hands. ❌
- Person 5: He has nothing in his hands. ❌          - Person 6: He is holding an ice-cream. ✅
- Person 7: He has nothing in his hands. ❌          - Person 8: He is holding a lollipop. ✅
- Person 9: She is holding a cat. ✅… **</think>**

**<answer>**```json[{"bbox_2d": [999, 271, 1178, 811], "label": "all people with something in their hands"}, {"bbox_2d": [0, 231, 178, 783], "label": "all people with something in their hands"}, {"bbox_2d": [259, 127, 400, 450], "label": "all people with something in their hands"}, {"bbox_2d": [768, 485, 956, 811], "label": "all people with something in their hands"}, {"bbox_2d": [175, 411, 468, 811], "label": "all people with something in their hands"}]```**</answer>**

Figure 22: Interaction referring example.

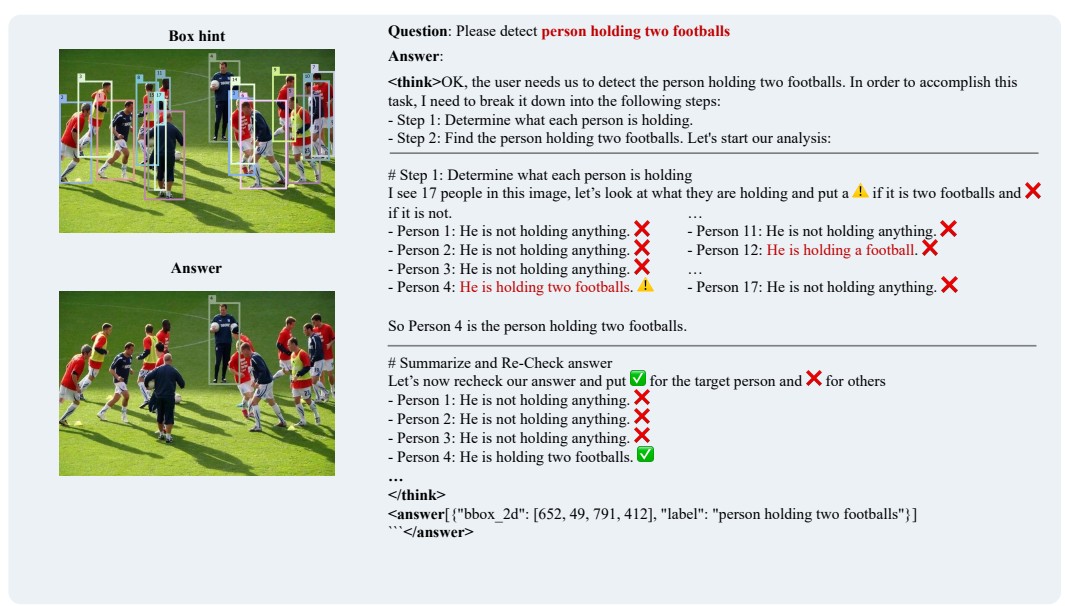

Figure 23: Interaction referring example.

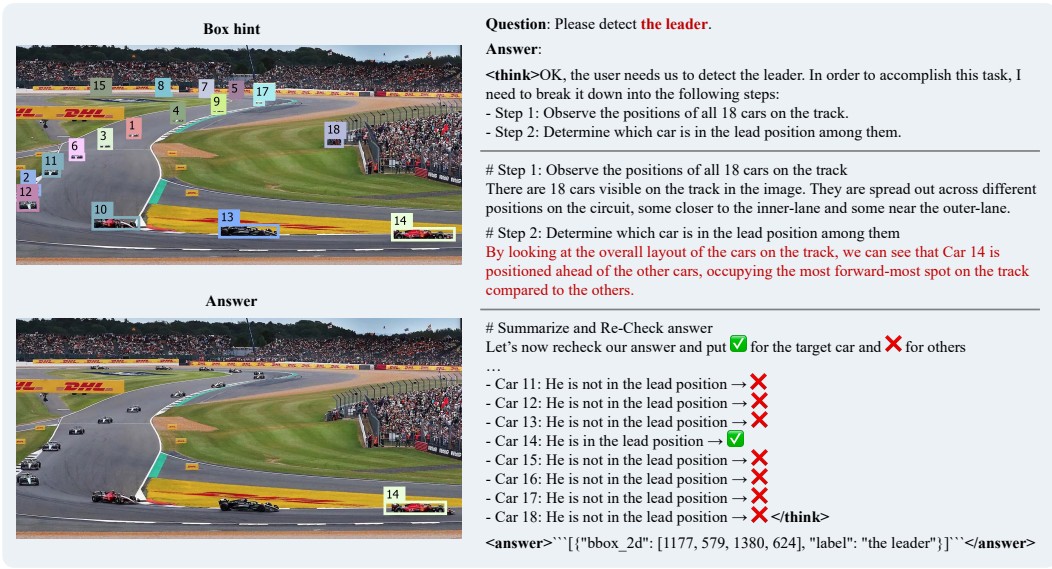

Figure 24: Position referring example.

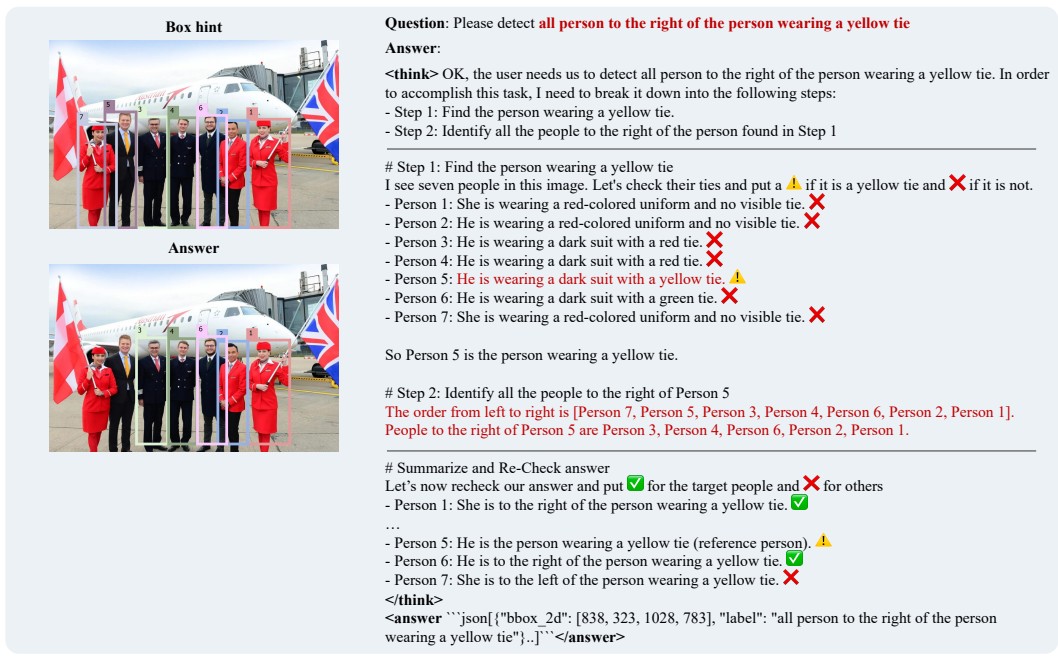

Figure 25: Position referring example.

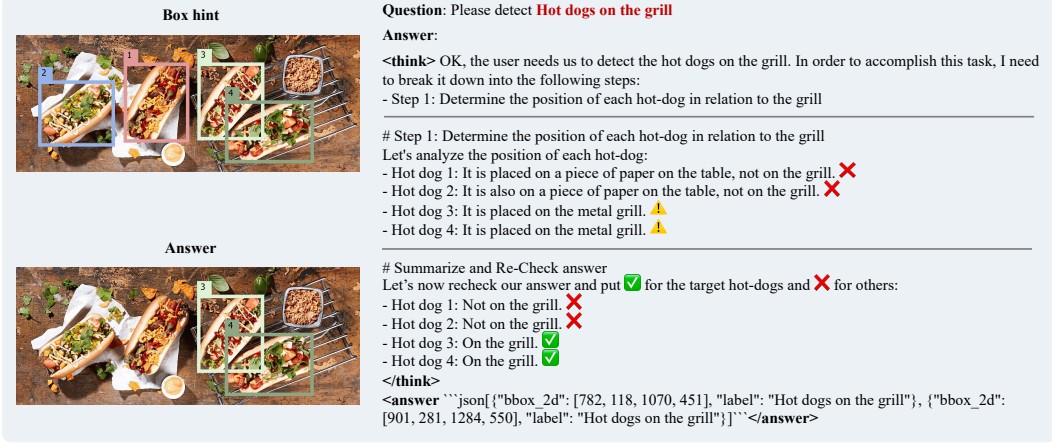

Figure 26: Position referring example.

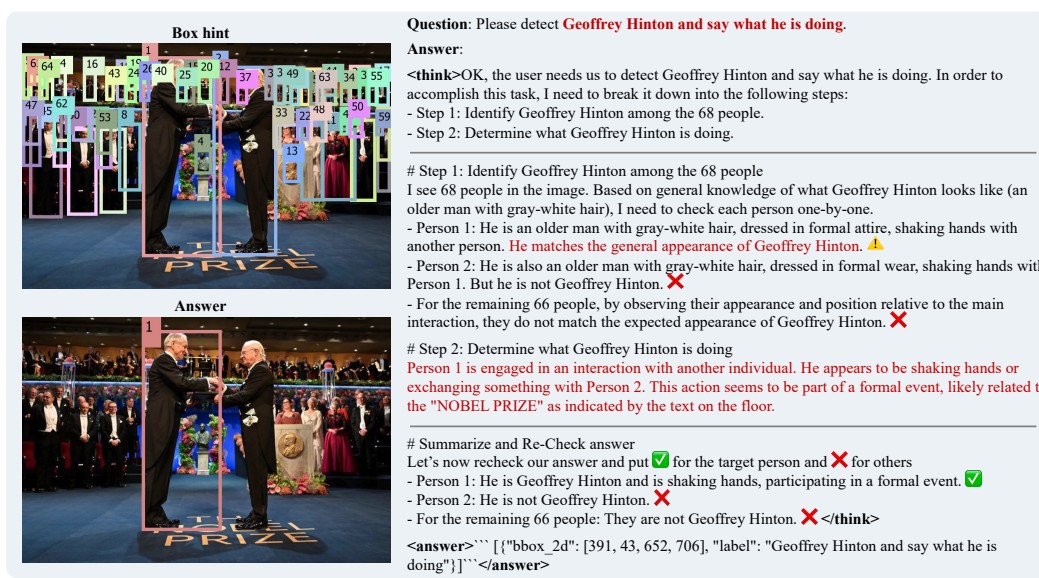

**Question**: Please detect **Geoffrey Hinton and say what he is doing**.

**Answer**:

**<think>**OK, the user needs us to detect Geoffrey Hinton and say what he is doing. In order to accomplish this task, I need to break it down into the following steps:
- Step 1: Identify Geoffrey Hinton among the 68 people.
- Step 2: Determine what Geoffrey Hinton is doing.

---

# Step 1: Identify Geoffrey Hinton among the 68 people
I see 68 people in the image. Based on general knowledge of what Geoffrey Hinton looks like (an older man with gray-white hair), I need to check each person one-by-one.
- Person 1: He is an older man with gray-white hair, dressed in formal attire, shaking hands with another person. He matches the general appearance of Geoffrey Hinton. ⚠️
- Person 2: He is also an older man with gray-white hair, dressed in formal wear, shaking hands with Person 1. But he is not Geoffrey Hinton. ❌
- For the remaining 66 people, by observing their appearance and position relative to the main interaction, they do not match the expected appearance of Geoffrey Hinton. ❌

# Step 2: Determine what Geoffrey Hinton is doing
Person 1 is engaged in an interaction with another individual. He appears to be shaking hands or exchanging something with Person 2. This action seems to be part of a formal event, likely related to the "NOBEL PRIZE" as indicated by the text on the floor.

---

# Summarize and Re-Check answer
Let's now recheck our answer and put ✅ for the target person and ❌ for others
- Person 1: He is Geoffrey Hinton and is shaking hands, participating in a formal event. ✅
- Person 2: He is not Geoffrey Hinton. ❌
- For the remaining 66 people: They are not Geoffrey Hinton. ❌ **</think>**

**<answer>**``` [{"bbox_2d": [391, 43, 652, 706], "label": "Geoffrey Hinton and say what he is doing"}]```**</answer>**

Figure 27: Celebrity referring example.

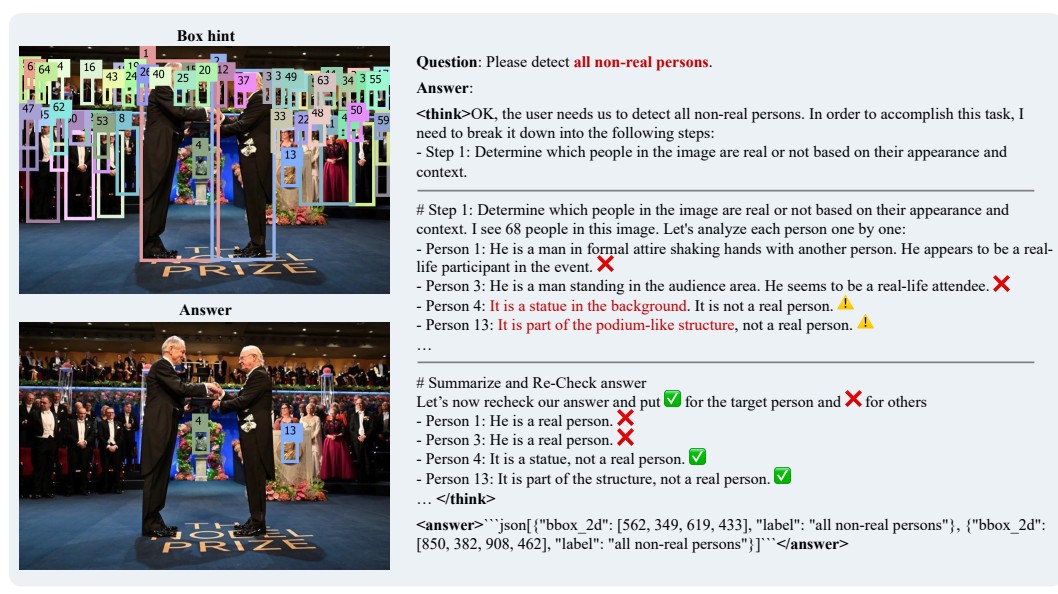

**Question**: Please detect **all non-real persons**.

**Answer**:

**<think>**OK, the user needs us to detect all non-real persons. In order to accomplish this task, I need to break it down into the following steps:
- Step 1: Determine which people in the image are real or not based on their appearance and context.

---

# Step 1: Determine which people in the image are real or not based on their appearance and context. I see 68 people in this image. Let's analyze each person one by one:
- Person 1: He is a man in formal attire shaking hands with another person. He appears to be a real-life participant in the event. ❌
- Person 3: He is a man standing in the audience area. He seems to be a real-life attendee. ❌
- Person 4: It is a statue in the background. It is not a real person. ⚠️
- Person 13: It is part of the podium-like structure, not a real person. ⚠️
…

---

# Summarize and Re-Check answer
Let's now recheck our answer and put ✅ for the target person and ❌ for others
- Person 1: He is a real person. ❌
- Person 3: He is a real person. ❌
- Person 4: It is a statue, not a real person. ✅
- Person 13: It is part of the structure, not a real person. ✅
… **</think>**

**<answer>**```json[{"bbox_2d": [562, 349, 619, 433], "label": "all non-real persons"}, {"bbox_2d": [850, 382, 908, 462], "label": "all non-real persons"}]```**</answer>**

Figure 28: Reasoning referring example.

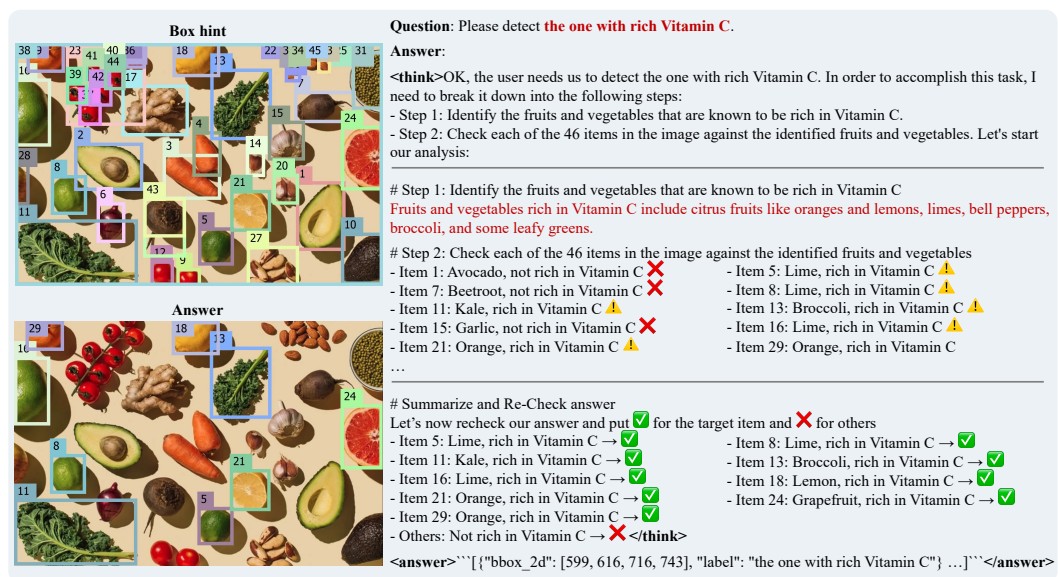

Figure 29: Reasoning referring example.

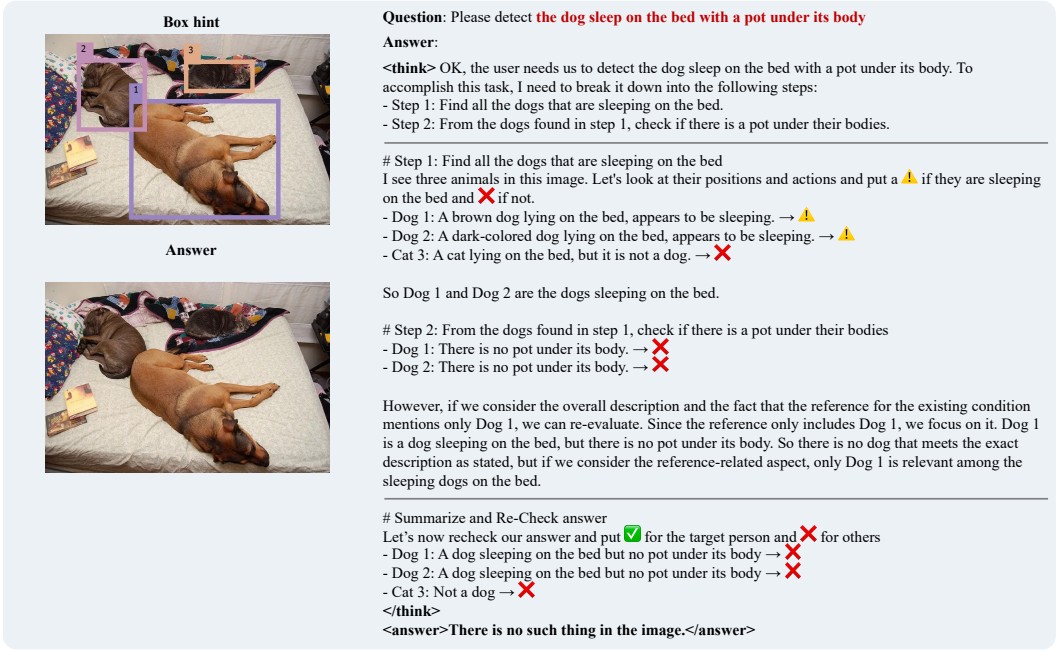

Figure 30: Rejection referring example.

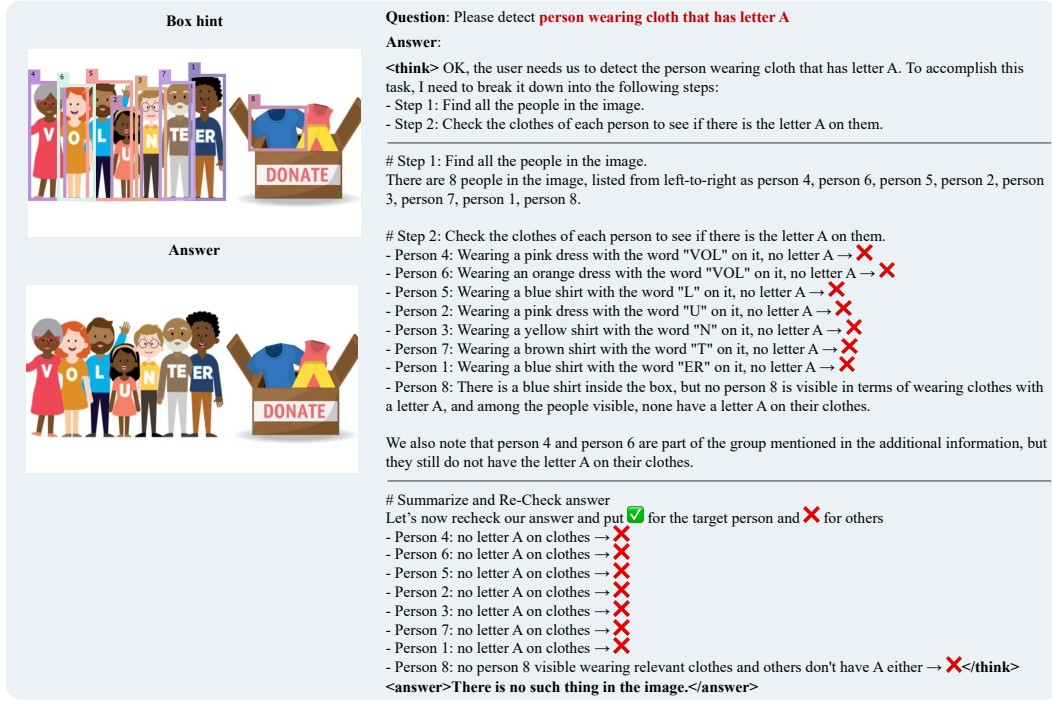

Figure 31: Rejection referring example.

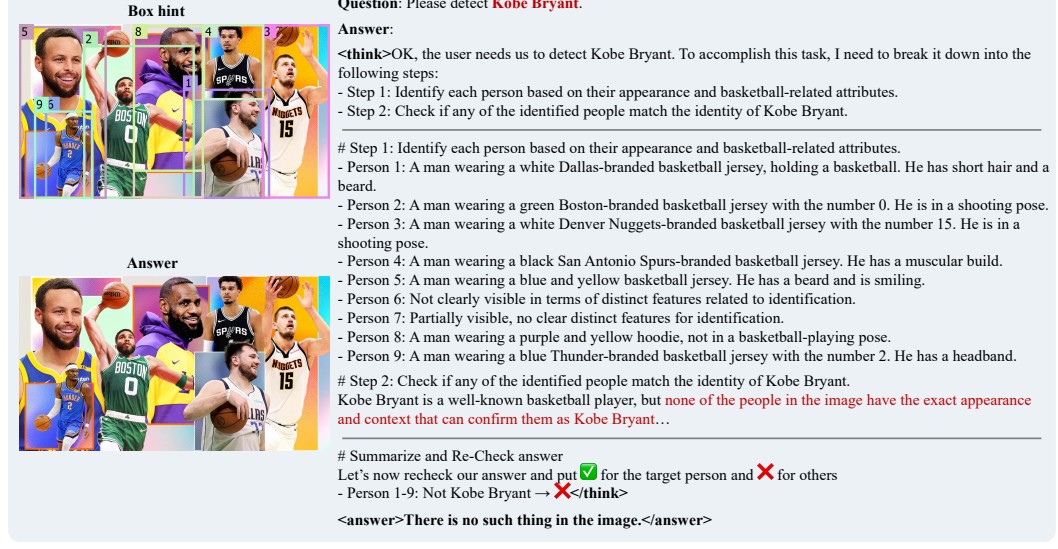

Figure 32: Rejection referring example.

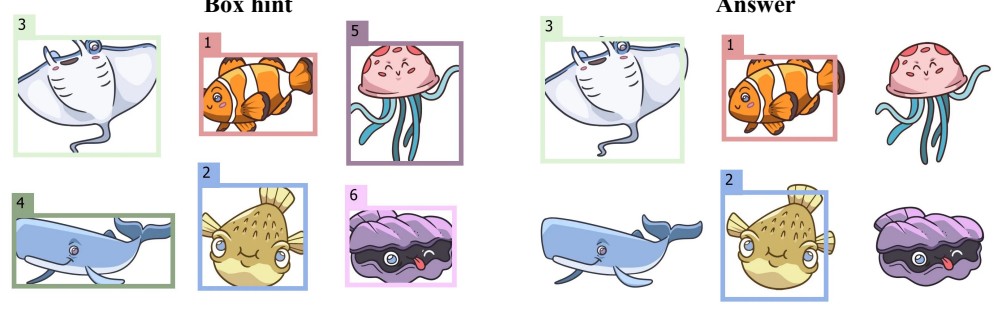

**Question**: Please detect **all fish, analyze their subcategories, and tell me if they are in a normal or endangered state**.

**Answer**:

**<think>**OK, the user needs us to detect all fish, analyze their subcategories, and tell if they are in a normal or endangered state. To accomplish this task, I need to break it down into the following steps:
- Step 1: Identify all the fish in the image.
- Step 2: Analyze the subcategories of the identified fish.
- Step 3: Determine the state (normal or endangered) of each fish.

# Step 1: Identify all the fish in the image
There are six sea-related characters in the image. Let's check which ones are fish:
- Fish 1: A clown fish with orange and white stripes. ⚠️
- Fish 2: A puffer fish with a round body and spiky fins. ⚠️
- Fish 3: A stingray with a flat, wing-like body. ⚠️
- Fish 4: A blue whale with a long body and flippers. ⚠️
- Fish 5: A jellyfish with a pink body and blue tentacles. ❌
- Fish 6: An oyster with a purple shell and an eye-like feature. ❌
So, Fish 1, Fish 2, Fish 3, and Fish 4 are fish.

# Step 2: Analyze the subcategories of the identified fish
- Fish 1: A clown fish, which is a type of fish.
- Fish 2: A puffer fish, which is a type of fish.
- Fish 3: A stingray, which is a type of fish.
- Fish 4: A blue whale, which is a type of fish.

# Step 3: Determine the state (normal or endangered) of each fish
- Fish 1: The clown fish is a common and widely-distributed fish, so it is in a normal state. ✅
- Fish 2: The puffer fish is also a relatively common fish, so it is in a normal state. ✅
- Fish 3: The stingray is not as common as some other fish, but still not on the endangered list, so it is in a normal state. ✅
- Fish 4: The blue whale is an endangered species due to hunting and habitat loss. ❌

# Summarize and Re-Check answer
Let's now recheck our answer and put ✅ for the target fish and ❌ for others:
- Fish 1: In a normal state. ✅
- Fish 2: In a normal state. ✅
- Fish 3: In a normal state. ✅
- Fish 4: Endangered. ❌
- Fish 5: Not a fish. ❌
- Fish 6: Not a fish. ❌**</think>**

**<answer>**``` [{"bbox_2d": [515, 156, 785, 348], "label": "all fish, analyze their subcategories, and tell me if they are in a normal or endangered state"}, {"bbox_2d": [512, 457, 764, 705], "label": "all fish, analyze their subcategories, and tell me if they are in a normal or endangered state"}, {"bbox_2d": [90, 116, 427, 397], "label": "all fish, analyze their subcategories, and tell me if they are in a normal or endangered state"}]```**</answer>**

Figure 33: Reasoning referring example with multi-task chat.

````
Example 1 :
````
METAINFO:
{
"total_num_of_person": 2,
"answer": "Person 1",
"order_from_left_to_right": [Person 1, Person 2]
}

Question: the person wearing a watch

The output should be:

(analysis)
OK, the user needs us to detect the person wearing a watch. To accomplish this task, I need to break it down into the following steps:
- Step 1: Determine what each person is wearing on their wrists to check for a watch.

# Step 1: Determine what each person is wearing on their wrists to check for a watch
- Person 1: She is wearing a watch on her left wrist. ⚠️
- Person 2: There is no visible watch on her wrists. ❌

# Summarize and Re - Check answer
Let's now recheck our answer and put ✅ for the target person and ❌ for others
- Person 1: She is wearing a watch → ✅
- Person 2: She is not wearing a watch → ❌
(analysis)

(answer)
To summarize, Person 1 is the person wearing a watch
(answer)

Figure 34: In-context prompt for *attribute* subset in HumanRef-CoT.

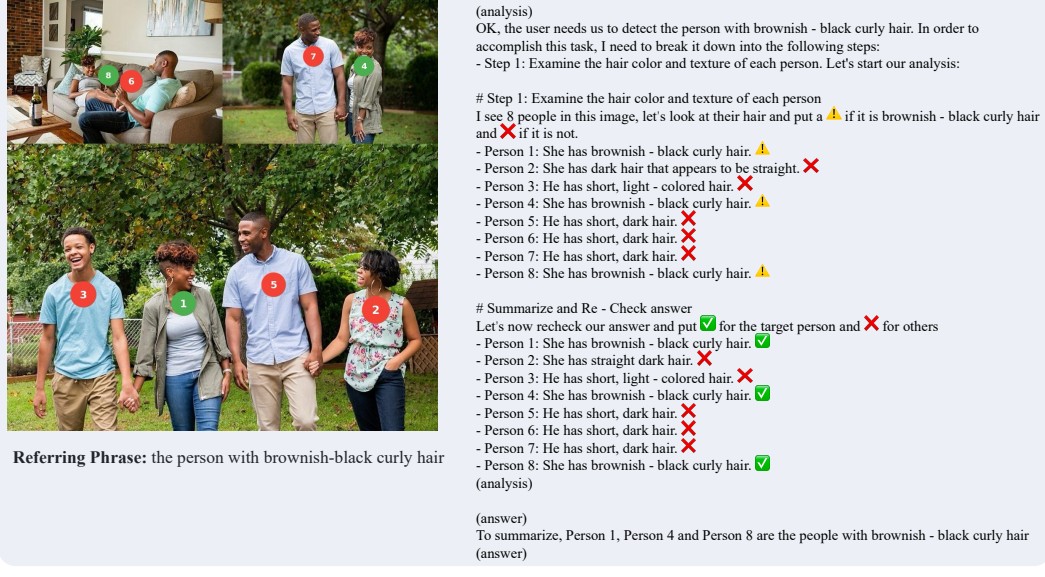

Figure 35: Visualization of GPT-4o's output on the *attribute* subset.

Example 1 :
```
METAINFO:
{
"total_num_of_person": 4,
"answer": "person 3",
"order_from_left_to_right": [person 1, person 2, person 3, person 4]
}

Question: The third person from the left to right

The output should be:
(analysis)
OK, the user needs us to detect the third person from the left to right. To accomplish this task, I need to break it down into the following steps:
- Step 1: Sort the people from left to right and find the third person from the left to right.

# Step 1: Sort the people from left to right and find the third person from the left to right.
I need to sort the people from left to right. I see 4 people in this image, and their order from left to right is [Person 1, Person 2, Person 3, Person 4].
So Person 3 is the third person from the left to right.

# Summarize and Re-Check answer
Let's now recheck our answer and put ✅ for the target person and ❌ for others
- Person 1: He is the first person from left to right → ❌
- Person 2: He is the second person from left to right → ❌
- Person 3: He is the third person from left to right → ✅
- Person 4: He is the forth person from left to right → ❌
(analysis)

(answer)
To summarize, person 3 is the third person from left to right
(answer)
```
```

Figure 36: In-context prompt for *position (inner)* subset in HumanRef-CoT.

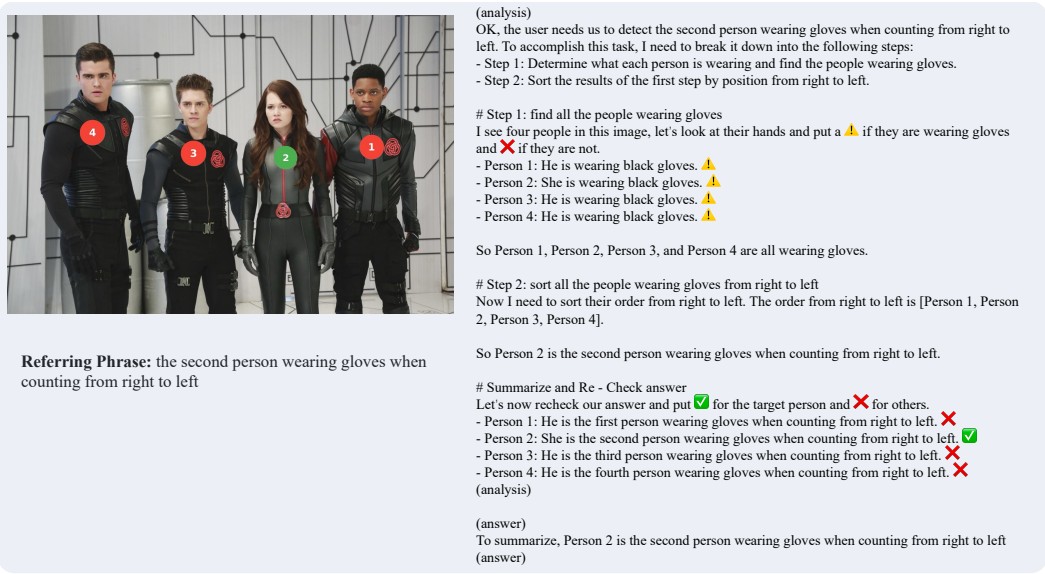

**Referring Phrase:** the second person wearing gloves when counting from right to left

(analysis)
OK, the user needs us to detect the second person wearing gloves when counting from right to left. To accomplish this task, I need to break it down into the following steps:
- Step 1: Determine what each person is wearing and find the people wearing gloves.
- Step 2: Sort the results of the first step by position from right to left.

# Step 1: find all the people wearing gloves
I see four people in this image, let's look at their hands and put a ⚠️ if they are wearing gloves and ❌ if they are not.
- Person 1: He is wearing black gloves. ⚠️
- Person 2: She is wearing black gloves. ⚠️
- Person 3: He is wearing black gloves. ⚠️
- Person 4: He is wearing black gloves. ⚠️

So Person 1, Person 2, Person 3, and Person 4 are all wearing gloves.

# Step 2: sort all the people wearing gloves from right to left
Now I need to sort their order from right to left. The order from right to left is [Person 1, Person 2, Person 3, Person 4].

So Person 2 is the second person wearing gloves when counting from right to left.

# Summarize and Re - Check answer
Let's now recheck our answer and put ✅ for the target person and ❌ for others.
- Person 1: He is the first person wearing gloves when counting from right to left. ❌
- Person 2: She is the second person wearing gloves when counting from right to left. ✅
- Person 3: He is the third person wearing gloves when counting from right to left. ❌
- Person 4: He is the fourth person wearing gloves when counting from right to left. ❌
(analysis)

(answer)
To summarize, Person 2 is the second person wearing gloves when counting from right to left
(answer)

Figure 37: Visualization of GPT-4o's output on the *position (inner)* subset.

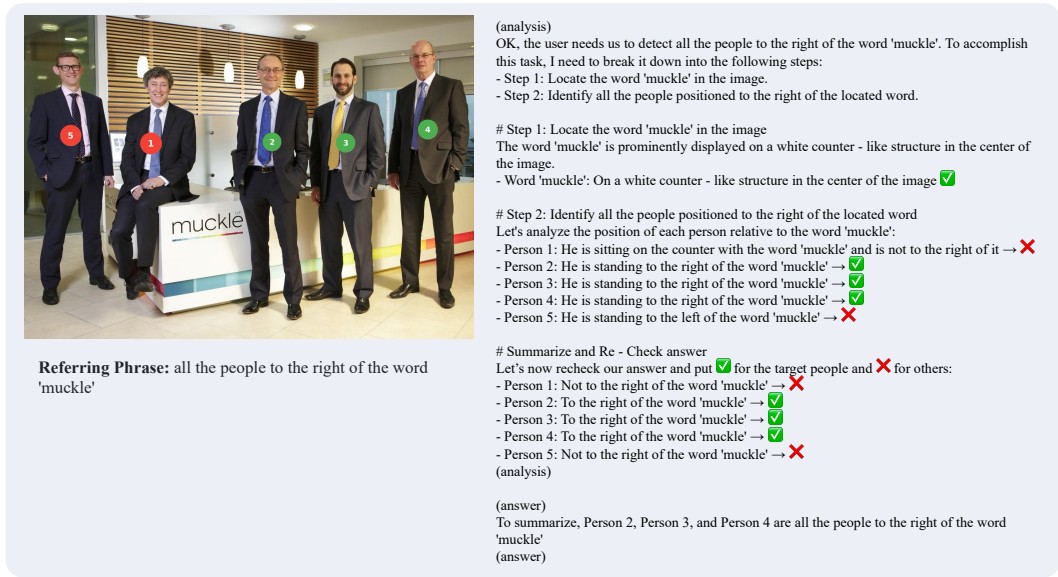

Figure 38: In-context prompt for *position (outer)* subset in HumanRef-CoT.

```
Example 1 :
```
```
METAINFO:
{
    "total_num_of_person": 5,
    "answer": "person 1",
    "order_from_left_to_right": [person 1, person 4, person 2, person 5, person 3]
}

Question: the person to the left of the green light bulb

The output should be:
(analysis)
OK, the user needs us to detect the person to the left of the green light-bulb. To accomplish this task, I need to break it down into the following steps:
Step 1: Find the green light-bulb in this image.
Step 2: Fine the person to the left of the green light-bulb

# Step 1: Find the green light-bulb in this image.
Clearly, I can see a green light - bulb near the lower - left part of the image.
- green light - bulb: a green light - bulb near the lower - left part of the image ✅

# Step2: Fine the person to the left of the green light-bulb
Let's analyze the position of each person next:
- Person 1: He is standing directly to the left of the green light-bulb → ✅
- Person 2: She is on a ladder near the top-right part of the image, not to the left of the green light - bulb → ❌
- Person 3: He is standing near the right side of the image holding a tablet, not to the left of the green light - bulb → ❌

# Summarize and Re-Check answer
Let's now recheck our answer and put ✅ for the target person and ❌ for others
- Person 1: He is standing directly to the left of the green light-bulb → ✅
- Person 2: She is on a ladder not the left of the green light-bulb → ❌
- Person 3: He is not on the left of the green light-bulb → ❌

(analysis)

(answer)
To summarize, person 1 is to the left of the green light bulb
(answer)
```
```

Figure 39: Visualization of GPT-4o's output on the *position (outer)* subset.

Example 1 :
```
METAINFO:
{
    "total_num_of_person": 3,
    "answer": "person 1",
    "order_from_left_to_right": [person 1, person 2, person 3, person 4]
}

Question: The man who is hugging the woman from behind

The output should be:
(analysis)
OK, the user needs us to detect the man who is hugging the woman from behind
. To accomplish this task, I need to break it down into the following steps:
- Step 1: Find all the man in this image.
- Step 2: Look at each man's action and find the man who is hugging the woman from behind

# Step 1: Find all the man in this image.
I see four person in this image, let's look at their gender and put a ⚠️ if they are man and ❌ if it is not.
- Person 1: It is a man sitting on the fence → ⚠️
- Person 2: It is a man wearing a blue shirt → ⚠️
- Person 3: It is a woman in red dress → ❌
- Person 4: It is a woman standing on the grass → ❌

# Step 2: Look at each man's interaction from step 1
Person 1: He is man standing behind a woman (Person 3), with his arms around her, fulfilling the description of hugging the woman from behind → ✅
Person 2: A man sitting on the fence, holding hands with person 4. He is not hugging a woman from behind. → ❌
So Person 1 is the man hugging the woman (Person 3) from behind

# Summarize and Re-Check answer
Let's now recheck our answer and put ✅ for the target person and ❌ for others
- Person 1: man hugging the woman from behind → ✅
- Person 2: man not hugging from behind → ❌
- Person 3: a woman, not a man hugging from behind → ❌
- Person 4: a woman, not a man hugging from behind → ❌
(analysis)

(answer)
To summarize, person 1 is the man who is hugging the woman from behind
(answer)
```

Figure 40: In-context prompt for *interaction (inner)* subset in HumanRef-CoT.

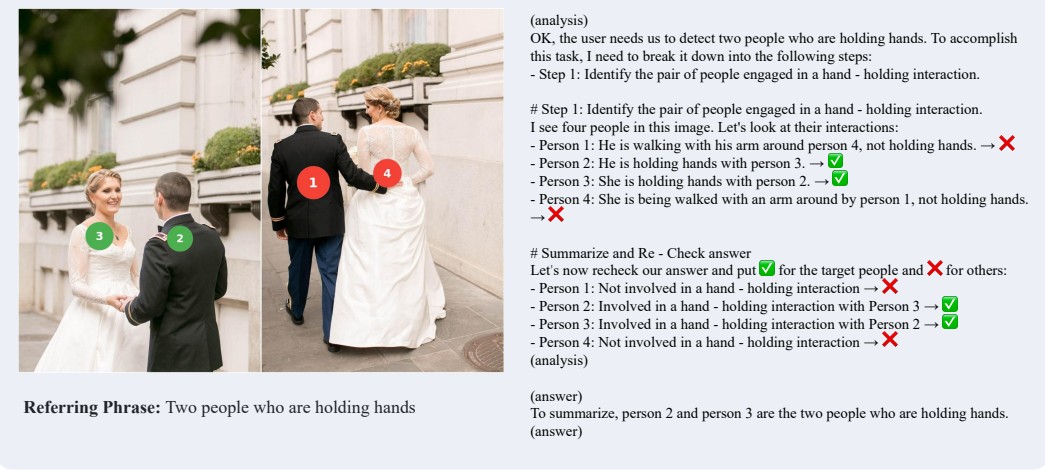

Figure 41: Visualization of GPT-4o's output on the *interaction (inner)* subset.

```
Example 1 :
```
METAINFO:
{
    "total_num_of_person": 3,
    "answer": "person 1",
    "order_from_left_to_right": [person 2, person 3, person 1]
}

Question: The woman holding a red sniper rifle in her hand

The output should be:
(analysis)
OK, the user needs us to find the woman holding a red sniper-rifle in her hand. To accomplish this task, I need to break it down into the following steps:
- Step 1: See if everyone is holding objects in their hands and what they're holding

# Step1: See if everyone is holding objects in their hands and what they're holding
There are 5 people in the picture, and we need to check the objects held by each of them:
- Person 1: A woman with dark hair and a red - themed outfit. She is holding a long, red firearm which appears to be a sniper - rifle. So she matches the description → ✅
- Person 2: A woman with short orange hair, wearing a pink and white outfit. She is holding a gray and pink - colored weapon that does not resemble a red sniper - rifle → ❌
- Person 3: A woman with long orange hair and a green and white dress. She is holding a weapon, but it is not a red sniper - rifle → ❌

# Summarize and Re-Check answer
Let's now recheck our answer and put ✅ for the target person and ❌ for others
Person 1: holding a red sniper - rifle → ✅
Person 2: holding a non - red weapon → ❌
Person 3: holding a non - red weapon → ❌
(analysis)

(answer)
To summarize, person 1 is the woman holding a red sniper rifle in her hand.
(answer)
```

Figure 42: In-context prompt for *interaction (outer)* subset in HumanRef-CoT.

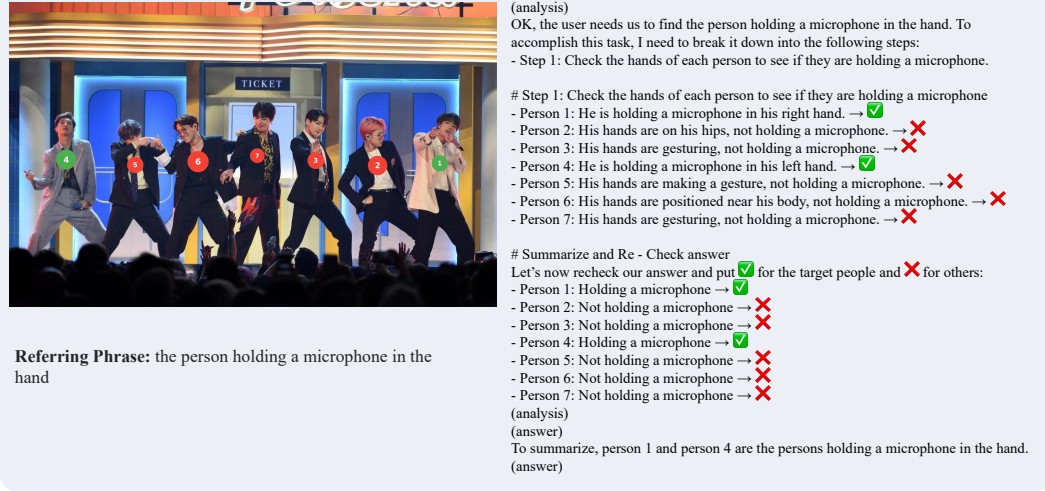

**Referring Phrase:** the person holding a microphone in the hand

(analysis)
OK, the user needs us to find the person holding a microphone in the hand. To accomplish this task, I need to break it down into the following steps:
- Step 1: Check the hands of each person to see if they are holding a microphone.

# Step 1: Check the hands of each person to see if they are holding a microphone
- Person 1: He is holding a microphone in his right hand. → ✅
- Person 2: His hands are on his hips, not holding a microphone. → ❌
- Person 3: His hands are gesturing, not holding a microphone. → ❌
- Person 4: He is holding a microphone in his left hand. → ✅
- Person 5: His hands are making a gesture, not holding a microphone. → ❌
- Person 6: His hands are positioned near his body, not holding a microphone. → ❌
- Person 7: His hands are gesturing, not holding a microphone. → ❌

# Summarize and Re - Check answer
Let's now recheck our answer and put ✅ for the target people and ❌ for others:
- Person 1: Holding a microphone → ✅
- Person 2: Not holding a microphone → ❌
- Person 3: Not holding a microphone → ❌
- Person 4: Holding a microphone → ✅
- Person 5: Not holding a microphone → ❌
- Person 6: Not holding a microphone → ❌
- Person 7: Not holding a microphone → ❌
(analysis)
(answer)
To summarize, person 1 and person 4 are the persons holding a microphone in the hand.
(answer)

Figure 43: Visualization of GPT-4o's output on the *interaction (outer)* subset.

Example 1 :
```
METAINFO:
{
    "total_num_of_person": 3,
    "answer": "person 1",
    "order_from_left_to_right": [person 3, person 1 , person 2]
}

Question: The woman to the left ot the woman in blue dress

The output should be:
(analysis)
OK, the user needs us to detect the woman to the left of the woman in the blue dress. In order to accomplish this task, I need to break it down into the following steps:
Step 1: Find the woman in the blue dress.
Step 2: Find the woman to her left.
Let's start our analysis:

# Step 1: Find the woman in the blue dress
I see three people in this image. Let's determine who is wearing a blue dress and put a ⚠️ if it is and ❌ if it is not.
- Person 1: She is wearing a pink dress. ❌
- Person 2: She is wearing a blue dress. ⚠️
- Person 3: He is wearing a black suit with a pink tie. ❌

So Person 2 is the woman in the blue dress. Now I need to find the woman to her left.

# Step 2: Find the woman to the left of Person 2
The order from left to right is [Person 3, Person 1, Person 2]. So Person 1 is to the left of Person 2. And Person 1 is a woman.

# Summarize and Re - Check answer
Let's now recheck our answer and put ✅ for the target person and ❌ for others
- Person 1: She is the woman to the left of the woman in the blue dress. ✅
- Person 2: She is the woman in the blue dress. ❌
- Person 3: He is a man, not the woman we are looking for. ❌
(analysis)

(answer)
To summarize, Person 1 is the woman to the left of the woman in the blue dress.
(answer)
```

Figure 44: In-context prompt for *reasoning (inner position)* subset in HumanRef-CoT.

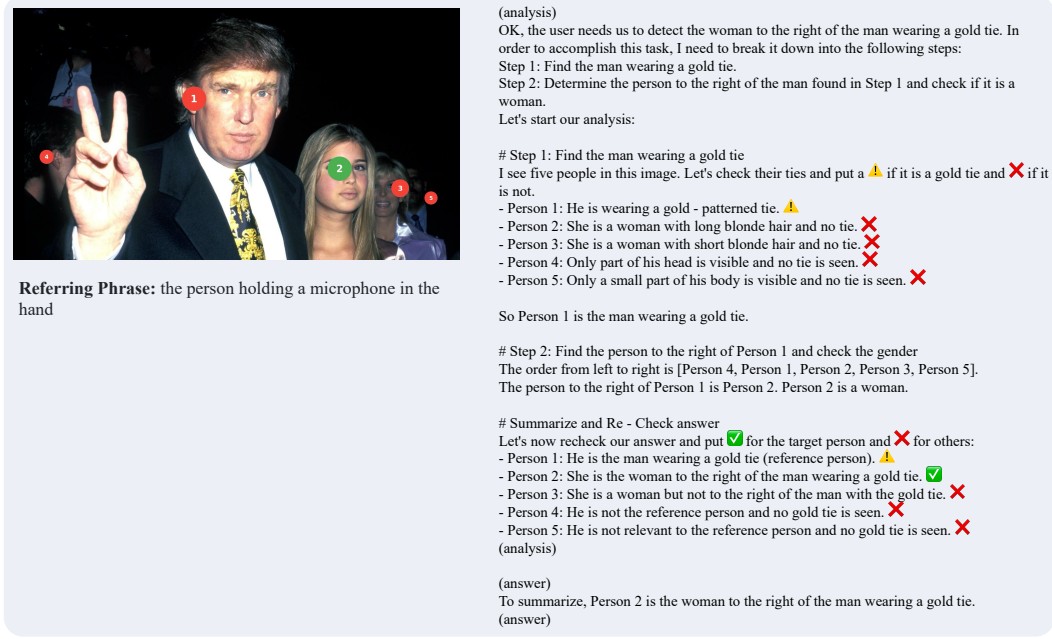

**Referring Phrase:** the person holding a microphone in the hand

(analysis)
OK, the user needs us to detect the woman to the right of the man wearing a gold tie. In order to accomplish this task, I need to break it down into the following steps:
Step 1: Find the man wearing a gold tie.
Step 2: Determine the person to the right of the man found in Step 1 and check if it is a woman.
Let's start our analysis:

# Step 1: Find the man wearing a gold tie
I see five people in this image. Let's check their ties and put a ⚠️ if it is a gold tie and ❌ if it is not.
- Person 1: He is wearing a gold - patterned tie. ⚠️
- Person 2: She is a woman with long blonde hair and no tie. ❌
- Person 3: She is a woman with short blonde hair and no tie. ❌
- Person 4: Only part of his head is visible and no tie is seen. ❌
- Person 5: Only a small part of his body is visible and no tie is seen. ❌

So Person 1 is the man wearing a gold tie.

# Step 2: Find the person to the right of Person 1 and check the gender
The order from left to right is [Person 4, Person 1, Person 2, Person 3, Person 5].
The person to the right of Person 1 is Person 2. Person 2 is a woman.

# Summarize and Re - Check answer
Let's now recheck our answer and put ✅ for the target person and ❌ for others:
- Person 1: He is the man wearing a gold tie (reference person). ⚠️
- Person 2: She is the woman to the right of the man wearing a gold tie. ✅
- Person 3: She is a woman but not to the right of the man with the gold tie. ❌
- Person 4: He is not the reference person and no gold tie is seen. ❌
- Person 5: He is not relevant to the reference person and no gold tie is seen. ❌
(analysis)

(answer)
To summarize, Person 2 is the woman to the right of the man wearing a gold tie.
(answer)

Figure 45: Visualization of GPT-4o's output on the *reasoning (inner position)* subset.
.

```
Example 1 :
```
METAINFO:
{
    "total_num_of_person": 4,
    "answer": "person 3, person 2, person 1",
    "order_from_left_to_right": [person 4, person 2, person 3, person 1]
}

Question: Person wearing glasses but not sitting on a chair

The output should be:
(analysis)
OK, the user needs us to detect the person wearing glasses but not sitting on a chair. To accomplish this task, I need to break it down into the following steps:
Step 1: Find all person wearing glasses.
Step 2: Find the person in Step 1 that are sitting on a chair
Step 3: Remove person in Step 2 from Step 1 to get the final result

# Step 1: Find all person wearing glasses.
I see five people in this image, let's look at their face and put a ⚠️ if they are wearing glasses and ❌ if it is not.
- Person 1: A man wearing a red shirt, he is wearing glasses → ⚠️
- Person 2: A man wearing a light-blue shirt, he is wearing glasses → ❌
- Person 3: A woman with blonde hair, she is not wearing glasses → ❌
Person 4: A woman in a white top, she is not wearing glasses → ⚠️
Person 5: A man wearing a light blue long sleeved shirt, he is wearing glasses → ⚠️

So Person 1, Person 4 and Person 5 are wearing glasses

# Step 2: Find the person in Step 1 that are sitting on a chair
Let's look at the three people in Step 1 and put a ⚠️ if they are sitting on a chair and ❌ if it is not.
- Person 1: He is sitting on a chair → ⚠️
- Person 2: He is sitting on a chair → ⚠️
Person 5: He is standing instead of sitting → ❌

# Step 3Remove person in Step 2 from Step 1 to get the final result
Remove Person 1 and Person 2 and we get Person 5 that meets the description

Let's now recheck our answer and put ✅ for the target person and ❌ for others
Person 1: wearing glasses but sitting on a chair → ❌
Person 2: wearing glasses but sitting on a chair → ❌
Person 3: not wearing glasses → ❌
Person 4: not wearing glasses → ❌
Person 5: wearing glasses and not sitting on a chair → ✅

(answer)
To summarize, person 3 is the third person from left to right
(answer)
```
```

Figure 46: In-context prompt for *reasoning (attribute)* subset in HumanRef-CoT.

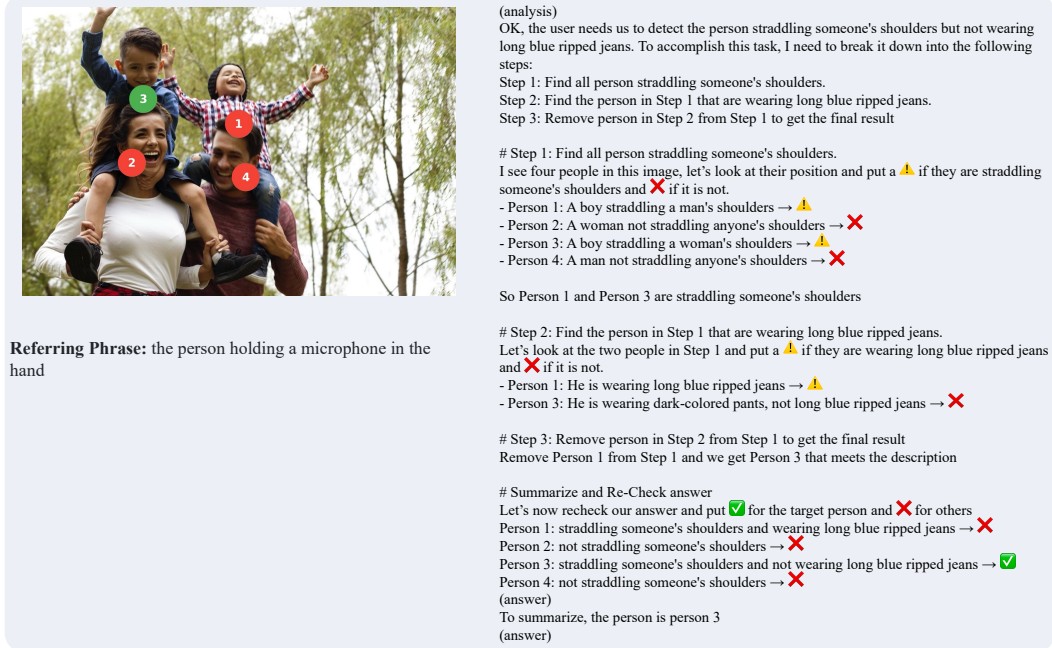

**Referring Phrase:** the person holding a microphone in the hand

(analysis)
OK, the user needs us to detect the person straddling someone's shoulders but not wearing long blue ripped jeans. To accomplish this task, I need to break it down into the following steps:
Step 1: Find all person straddling someone's shoulders.
Step 2: Find the person in Step 1 that are wearing long blue ripped jeans.
Step 3: Remove person in Step 2 from Step 1 to get the final result

# Step 1: Find all person straddling someone's shoulders.
I see four people in this image, let's look at their position and put a ⚠️ if they are straddling someone's shoulders and ❌ if it is not.
- Person 1: A boy straddling a man's shoulders → ⚠️
- Person 2: A woman not straddling anyone's shoulders → ❌
- Person 3: A boy straddling a woman's shoulders → ⚠️
- Person 4: A man not straddling anyone's shoulders → ❌

So Person 1 and Person 3 are straddling someone's shoulders

# Step 2: Find the person in Step 1 that are wearing long blue ripped jeans.
Let's look at the two people in Step 1 and put a ⚠️ if they are wearing long blue ripped jeans and ❌ if it is not.
- Person 1: He is wearing long blue ripped jeans → ⚠️
- Person 3: He is wearing dark-colored pants, not long blue ripped jeans → ❌

# Step 3: Remove person in Step 2 from Step 1 to get the final result
Remove Person 1 from Step 1 and we get Person 3 that meets the description

# Summarize and Re-Check answer
Let's now recheck our answer and put ✅ for the target person and ❌ for others
Person 1: straddling someone's shoulders and wearing long blue ripped jeans → ❌
Person 2: not straddling someone's shoulders → ❌
Person 3: straddling someone's shoulders and not wearing long blue ripped jeans → ✅
Person 4: not straddling someone's shoulders → ❌
(answer)
To summarize, the person is person 3
(answer)

Figure 47: Visualization of GPT-4o's output on the *reasoning (attribute)* subset..

Example:

METAINFO:
{
  "total_num_of_person": 4,
  "answer": "person 1",
  "order_from_left_to_right": [person 4, person 1, person 2, person 3]
}

(analysis)
OK, the user needs us to detect Brad Pitt. In order to accomplish this task, I need to break it down into the following step:
- Step 1: Identify Brad Pitt among the 8 people based on his appearance characteristics. Let's start our analysis:

# Step 1: Identify Brad Pitt among the 8 people based on his appearance characteristics
- Person 1: Has light - colored hair, similar facial features to Brad Pitt, and is wearing a black jacket over a white shirt, which matches the reference image of Brad Pitt. ⚠️
- Person 2: A man with dark hair sitting on a red couch, does not match Brad Pitt's appearance. ❌
- Person 3: A woman with long - brown hair, clearly not Brad Pitt. ❌
- Person 4: A man with dark hair and a dark shirt, does not match Brad Pitt's appearance. ❌

# Summarize and Re - Check answer
Let's now recheck our answer and put ✅ for the target person and ❌ for others
- Person 1: Matches Brad Pitt's appearance characteristics. ✅
- Person 2: Does not match Brad Pitt's appearance. ❌
- Person 3: Does not match Brad Pitt's appearance. ❌
- Person 4: Does not match Brad Pitt's appearance. ❌
(analysis)

(answer)
To summarize, Person 1 is Brad Pitt
 (answer)

Figure 48: In-context prompt for *celebrity recognition* subset in HumanRef-CoT.

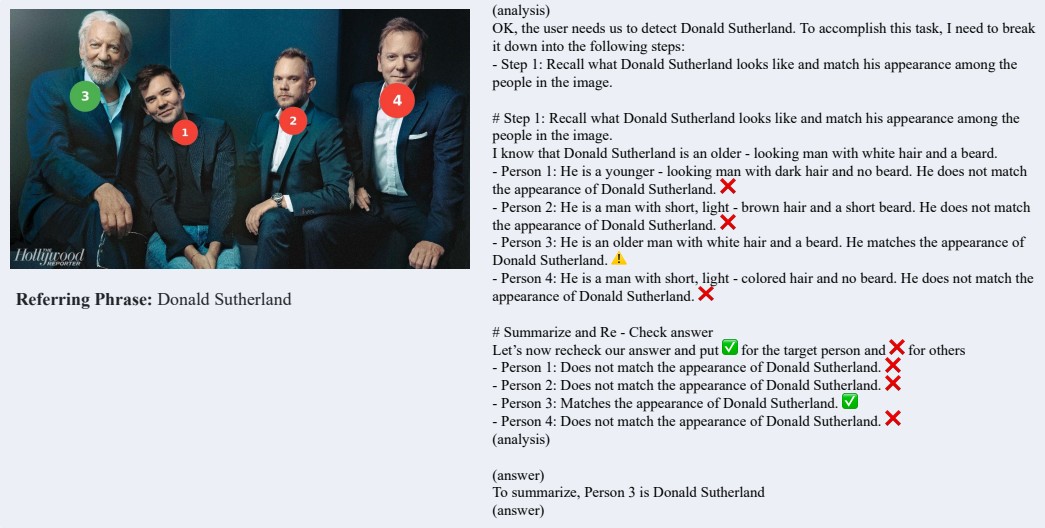

**Referring Phrase:** Donald Sutherland

(analysis)
OK, the user needs us to detect Donald Sutherland. To accomplish this task, I need to break it down into the following steps:
- Step 1: Recall what Donald Sutherland looks like and match his appearance among the people in the image.

# Step 1: Recall what Donald Sutherland looks like and match his appearance among the people in the image.
I know that Donald Sutherland is an older - looking man with white hair and a beard.
- Person 1: He is a younger - looking man with dark hair and no beard. He does not match the appearance of Donald Sutherland. ❌
- Person 2: He is a man with short, light - brown hair and a short beard. He does not match the appearance of Donald Sutherland. ❌
- Person 3: He is an older man with white hair and a beard. He matches the appearance of Donald Sutherland. ⚠️
- Person 4: He is a man with short, light - colored hair and no beard. He does not match the appearance of Donald Sutherland. ❌

# Summarize and Re - Check answer
Let's now recheck our answer and put ✅ for the target person and ❌ for others
- Person 1: Does not match the appearance of Donald Sutherland. ❌
- Person 2: Does not match the appearance of Donald Sutherland. ❌
- Person 3: Matches the appearance of Donald Sutherland. ✅
- Person 4: Does not match the appearance of Donald Sutherland. ❌
(analysis)

(answer)
To summarize, Person 3 is Donald Sutherland
(answer)

Figure 49: Visualization of GPT-4o's output on the *celebrity recognition* subset..

Example:

METAINFO:
{
  "total_num_of_person": 4,
  "reference_answer": "there is not person that meets the description"
}

(analysis)
OK, the user needs us to detect all the people with guns in the picture. I need to look at the objects held in each person's hands and determine if they are guns or not. I see 7 people in the diagram, let's analyze the objects held by each of them:

- Person 1: She is a woman with long black hair, her right fist is outstretched but she is not holding a gun, and we can't see her left hand, so we can tell that she is not holding a gun. ✖
- Person 2: A female with brown hair wearing a thin red jacket. She has her left hand up and looks nervous, but she's not holding a gun either. ✖
- Person 3: A bald policeman in a black jacket, we can't see his hands, so we don't think he's holding a gun either. ✖
- Person 4: A bald black male with a big mouth, he has his hands open but he is also not holding a gun. ✖

Let's now finalize:
Person 1: not holding a gun → ✖
Person 2:  not holding a gun → ✖
Person 3: not holding a gun → ✖
Person 4: not holding a gun → ✖
(analysis)

(answer)
To summarize, there is no person in the image that meets the description.
(Answer)

Figure 50: In-context prompt for *rejection* subset in HumanRef-CoT.

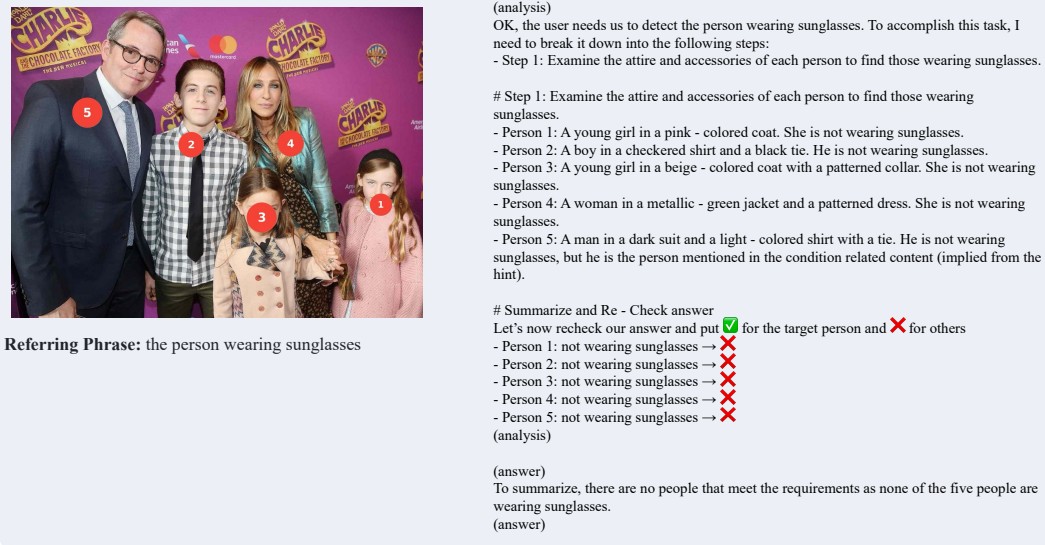

Figure 51: Visualization of GPT-4o's output on the *rejection*
subset..

