# OpenReview forum: "Rex-Thinker: Grounded Object Referring via Chain-of-Thought Reasoning"
_ICLR.cc/2026/Conference — ICLR 2026 Poster_

### Official Review · Reviewer_28Tw · 2025-10-25

**Soundness:** 3
**Presentation:** 3
**Contribution:** 2
**Rating:** 4
**Confidence:** 4

**Summary:**

An overly simplified summary of the paper: the authors propose using explicit chain-of-thought (CoT) reasoning with predefined steps (plan, action, summarize) for the object-referring task (i.e., detecting the exact object(s) referred to by a natural language question). To train models to follow these steps the authors use GPT-4o to annotate an existing object-referring dataset, HumanRef (Jiang et al. (2025b)), with their reasoning traces, creating the HumanRef-CoT dataset. They then use those annotations to post-train a model (SFT for cold-start, followed by GRPO) to produce the Rex-Thinker model. The evaluation is split into in-domain (HumanRef) and out-of-domain (RefCOCOg (Mao et al., 2016)) benchmarks: Rex-Thinker outperforms other models on the in-domain dataset and is comparable to them on the out-of-domain set.

**Strengths:**

Overall, the paper presents an interesting idea and gives enough explanation and detail to follow it (the appendix is particularly helpful for things omitted in the main paper due to space). The evaluation covers both in- and out-of-domain datasets to demonstrate the model’s effectiveness, and the authors include additional experiments exploring different aspects of the approach.

**Weaknesses:**

There are, however, three main issues that justify my score.
First, the paper doesn’t convincingly demonstrate the quality of the generated dataset, HumanRef-CoT. There is no targeted evaluation of the quality or usefulness of the reasonings added on top of HumanRef — no human evaluation or deeper analysis — which is surprising since the dataset is presented as one of the paper’s main contributions. My concern is amplified by the authors’ own note (line 239) that GPT-4o sometimes produces wrong answers, which is problematic given that the ground truth is available in the input (Figure 2).
Second, the out-of-domain results for Rex-Thinker limit the generalizability of the proposed idea — the gains seem largely in-domain.
Third, the evaluation could dig deeper into which instance types Rex-Thinker fails on and which it improves; for example, the “Interaction” column in Table 2 seems like a good candidate for further discussion. With the current results it’s hard to draw clear conclusions about the model’s strengths and limitations.

**Questions:**

For the results in Tables 2 and 3, it’s unclear how the numbers for the other models were obtained. What exact setup was used for those baselines? For example, was SFT done using ground-truth reasoning or not? Please clarify the evaluation/setup for each compared model (I may have missed this in the appendix so please point me to the correct section if that is the case).

Minor typo: line 332, “Blod” → “bold.”

**Details Of Ethics Concerns:**

Given that one of the main contributions is a dataset, I was surprised that the paper doesn’t include any discussion of its ethical aspects. From what I understood, there don’t seem to be any guardrails against problematic or offensive language in the generated reasoning steps, which could make the dataset ethically questionable. To be clear, I didn’t find any evidence of such issues myself, but the absence of any discussion or acknowledgment of this risk in the paper is concerning.
Also there is no discussion on plans to make the dataset publicly available.

---

> ### Author Response · Authors · 2025-11-16
> **Response to Reviewer 28Tw (Part 1)**
>
> ## W1: On the Quality Control and Human Validation of the HumanRef-CoT Dataset
> We thank the reviewer for this insightful comment. We agree that data reliability is crucial. In response, we have updated our manuscript by:
> - Adding a detailed description of our automated quality control pipeline to the main methodology in Section 3.2 (page 5, line 238- 250).
> - Conducting a new human evaluation study, with the full methodology and results now presented in Appendix A.2.2 (page 16, line 857).
> - To ensure full transparency and facilitate future research, **we will publicly release the HumanRef-CoT dataset immediately following the discussion period.**
>
> **1. Effectiveness of Our Rule-Based Filtering:** Our initial quality control was not solely based on matching the final answer. As detailed in Appendix A.2.1 (page 16, line 821), our system prompt for GPT-4o requires it to output a ✅ or ❌ emoji for each reasoning step in both the Action and Summarization stages. This allows for more fine-grained, rule-based filtering based on two strict conditions:
>
> - The final Summarization outcome (the set of ✅ emojis) must be perfectly consistent with the GT labels.
> - The outcome of the Action phase must be consistent with the Summarization phase, ensuring logical coherence between the step-by-step reasoning and the final conclusion.
>
> We found that this filtering process is highly effective at catching errors. The table below shows the number of data points removed from the HumanRef-CoT dataset by applying these rules:
>
>   | Subset      | GPT-4o Annotated Data | Removed Data | Removal Rate |
>   |-------------|:--------------:|:------------:|:------------:|
>   | Celebrity   | 6,775          | 294          | 4.3%         |
>   | Interaction | 5,875          | 72           | 1.2%         |
>   | Position    | 15,950         | 533          | 3.3%         |
>   | Reasoning   | 9,488          | 719          | 7.6%         |
>   | Attribute   | 37,648         | 1,184        | 3.1%         |
>   | Rejection   | 19,096         | 1,184        | 6.2%         |
>
> This demonstrates that while GPT-4o can produce errors, our automated filtering successfully identifies and removes a significant portion of inconsistent or incorrect annotations before they are used for training.
>
>
> **2. Human Evaluation of HumanRef-CoT Data**:
> To assess the quality of the data that passed our automated filtering, we conducted a manual review as the reviewer suggested.
>
> - **Methodology**: We randomly sampled 100 instances from each of the 6 subsets, creating a review set of 600 samples. We recruited five Ph.D. students to perform the review using a custom HTML-based tool, shown in the revised version of our paper in Appendix A.2.2 (page 16, line 857)
> - **Error Categories:** Annotators were asked to check for three types of errors:
>   - **Wrong Summarization**: The final summary of person indices does not match the GT.
>   - **Action and Summarization Inconsistent**: For example, the Action phase concludes a person is a non-match, but the Summarization phase incorrectly includes them.
>   - **Wrong Action Result**: A specific reasoning step in the Action phase is factually incorrect (e.g. misjudging a clothing color of a candidate person).
>
> The results of this human evalation on the 600 samples are summarized below:
>
> | Error Type                        | Celebrity | Interaction | Position | Reasoning | Attribute | Rejection |
> |------------------------------------|-----------|-------------|----------|-----------|-----------|-----------|
> | Wrong Summarization               |     0     |      0      |    0     |     0     |     0     |     0     |
> | Action and Summarization Inconsistent |  0    |      0      |    0     |     0     |     0     |     0     |
> | Wrong Action Result                |     2     |      2      |     0    |     3    |     0     |     0     |
>
> Our automated filtering process is extremely effective at eliminating high-level logical and summarization errors. The human evaluators found zero such errors in the final dataset.
>
> However, we acknowledge that a small number of `Wrong Action Result` errors persist (7 out of 600, or 1.2%). Upon inspection, we found these low-level reasoning errors typically occur when an object is very small and partially occluded by its visual marker, forcing GPT-4o to make an hallucinated guess. Despite these minor errors, the final performance of our model when trained with this CoT data shows a significant improvement over the baseline without it. This demonstrates that the data still provides a valuable and effective training signal.

---

> > ### Author Response · Authors · 2025-11-16
> > **Response to Reviewer 28Tw (Part 2)**
> >
> > ## W2: On the Generalizability of the Proposed Reasoning Framework
> >
> > We thank the reviewer for this important question regarding out-of-domain generalization. We believe this is a key strength of our model, and we appreciate the opportunity to provide further context on our results, which demonstrate generalization in two key ways.
> >
> > - **1. Quantitative Performance in a Zero-Shot Context:** First, it is crucial to consider the evaluation setting for the results in Table 3. Our Rex-Thinker models are evaluated on RefCOCOg in a strict zero-shot setting, having been trained only on the human-centric HumanRef-CoT dataset. In contrast, the baseline models were trained on large-scale corpora that include the RefCOCOg dataset. Achieving a competitive score under these challenging zero-shot conditions is a testament to the model's ability to generalize its learned capabilities to new domains.
> >
> > - **2. Robust Generalization of the Core Reasoning Process:**  Crucially, we demonstrate that the model's structured Planning-Action-Summarization reasoning process generalizes effectively beyond its training domain. Our qualitative results provide strong evidence for this. As visualized in Figure 4 (main paper) and numerous appendix figures (13, 14, 15, 18, 21, 23), the model consistently applies the same reasoning steps to entirely new, non-human object categories.
> >
> > This shows that Rex-Thinker does not simply memorize patterns from the training set. It learns a fundamental, transferable reasoning skill that is robust across different domains and object types.
> >
> > ----
> >
> > ## W3: In-Depth Analysis of Model Strengths and Limitations
> >
> > We thank the reviewer for the suggestion to delve deeper into our model's specific strengths and limitations. Following this advice, **we have now included this detailed analysis and new visualizations in Appendix A.3.7 (Page 21, line 1125) of our revised manuscript.**
> >
> > **1. CoT Boosts Recall by Enforcing Systematic Evaluation**
> >
> > As suggested, we performed a direct comparison between our full model (Rex-Thinker-GRPO) and the baseline trained without CoT reasoning (Rex-Thinker-Plain). The results are shown below:
> >
> > ||Celebrity||Interaction| |Attribute|| Position| |Reasoning| |
> > |------|------|------|------|------|------|------|------|------|------|------|
> > ||Recall|Precision|Recall|Precision|Recall|Precision|Recall|Precision|Recall|Precision|
> > |Rex-Thinker-Plain|88.58|90.58|82.16|87.80|84.17|89.89|85.48|87.18|82.63|85.34|
> > |Rex-Thinker-GRPO|89.41 (+1.13)|91.30 (+0.72)|81.95 (-0.21)|88.11 (+0.31)|88.65 (+4.48)|91.98 (+2.09)|87.42 (+1.94)|88.68 (+1.50)|87.92 (+5.29)|87.71 (+2.37)|
> >
> > Our primary finding is that the CoT framework provides a substantial boost in Recall, particularly for categories that demand complex understanding like Reasoning (+5.29) and Attribute (+4.48). This result stems directly from our core design: the structured CoT process forces the model to systematically evaluate each candidate box against the expression's sub-goals. This step-by-step verification minimizes the chance of overlooking a potential target, thereby reducing false negatives (missed detections) and significantly improving recall. We have added qualitative examples in Figure 10 (Page 23) and Figure 11 (Page 24) that visually demonstrate this effect.
> >
> > **2. Limitation and Analysis on the "Interaction" Category**
> >
> > The "Interaction" category presents a unique challenge. Our analysis confirms this: while precision still saw a slight improvement (+0.31%), recall experienced a minor decrease (-0.21%).
> >
> > Upon manual inspection of failure cases, we hypothesize this is due to a fundamental mismatch between the relational nature of interaction expressions and the local, proposal-by-proposal nature of our CoT framework.
> >
> > - **The Challenge:** Interaction expressions (e.g., "the person talking to the woman in red") are inherently relational. They require the model to verify a relationship between two or more distinct entities, not just the properties of a single entity.
> >
> > - **Why Our Model Struggles Here:** The core limitation stems from our CoT framework, which evaluates each candidate proposal in isolation. To correctly verify "the person talking to the woman in red" while analyzing the candidate box for "the person," the model must simultaneously be aware of and ground the existence of "the woman in red.".
> >
> > Our current step-by-step process lacks an explicit mechanism to reason across different proposals simultaneously. It excels at verifying properties contained within a single box (e.g., "is this person wearing a hat?") but is not designed to validate relationships between boxes. This can lead the model to conservatively reject a correct candidate because it cannot confirm the full relational context from its local view, thus slightly lowering recall. We show visualization examples of this failure in Figure 12 (Page 25). We will try to address this limitation in our future work.

---

> > > ### Author Response · Authors · 2025-11-16
> > > **Response to Reviewer 28Tw (Part 3)**
> > >
> > > ## Q1: Experiment Settings in Table 2 and Table 3
> > >
> > > We thanker the reviewer for raising this important question about the experimental setup for our baseline comparisons, and we apologize if these details were not sufficiently clear in our initial submission. **We have added a evaluation settings "Baseline Setup" paragraph in Appendix 3.4 (Page20, line 1053) of our revised manuscript.**
> > >
> > > **1.Clarification for Table 2**:
> > > The baselines fall into two categories:
> > >
> > > - **Zero-Shot Evaluation:** The majority of the baseline models (including DINO-X, InternVL-2.5-8B, Ferret-7B, Groma-7B, ChatRex-7B, Qwen2.5-VL-7B, etc.) have never been trained on the HumanRef dataset. We evaluated these models using their publicly available, official checkpoints directly on the HumanRef benchmark. This provides a measure of their zero-shot generalization capability to our challenging, human-centric scenarios.
> > > - **State-of-the-Art In-Domain Model:** The RexSeek-7B model is the only baseline that was specifically trained on the HumanRef training set and was not trained using any form of Chain-of-Thought reasoning. RexSeek-7B is a direct-prediction model. We report the performance numbers directly from the original RexSeek[1] paper, as this represents the current state-of-the-art for non-CoT methods on this benchmark.
> > >
> > > **2.Clarification for Table 3**:
> > >
> > > For Table 3, which evaluates generalization on the RefCOCOg dataset, the context is different:
> > > - The reported numbers for all baseline models (RexSeek-7B, Grounding DINO, QwenVL-2.5-7B, ChatRex-7B) are taken directly from their original papers.
> > > - As per their publications, all of these models were trained with the RefCOCOg dataset.
> > > - None of these models are based on a Chain-of-Thought framework. They represent the state-of-the-art for direct-prediction or retrieval-based methods on this standard benchmark.
> > >
> > > ----
> > >
> > > ## Ethics Concerns on the HumanRef-CoT Dataset
> > >
> > > We are deeply grateful to the reviewer for raising these crucially important points regarding the ethical considerations and public availability of our HumanRef-CoT dataset. **We have added a new section "Ethical Considerations" in Appendix A.2.4 (Page 19, line 982) of our revised manuscript.**
> > >
> > > **1. Ethical Considerations and Safeguards**
> > > We share the reviewer's concern about the potential for generating problematic language. While we did not find any such instances during our work, we acknowledge that a proactive discussion of the safeguards we employed is necessary. Our data creation pipeline was designed with a multi-layered approach to mitigate these risks:
> > >
> > > - **Filtered Image Source:** The foundation of our dataset is the HumanRef dataset. The images in this source dataset have already undergone NSFW filtering, ensuring that the visual content is appropriate and does not contain sensitive material.
> > > - **Use of a Moderated Large Language Model:** We chose OpenAI's GPT-4o for generating the reasoning steps. As a state-of-the-art commercial model, GPT-4o is subject to rigorous safety protocols and content moderation filters developed by OpenAI. These built-in guardrails are specifically designed to prevent the generation of offensive, biased, or otherwise harmful content, serving as a primary safeguard for our annotation process.
> > > - **Final Data Review:** During our quality control and filtering stages, where we verified the logical correctness of the reasoning traces, we also remained vigilant for any inappropriate content. We can confirm that we did not encounter any instances of ethically questionable language in the final, curated HumanRef-CoT dataset.
> > >
> > >
> > > **2.Dataset Availability**
> > >
> > > We guarantee that the HumanRef-CoT dataset will be made publicly available to foster further research and ensure the reproducibility of our results.
> > >
> > > To make this commitment concrete and time-bound, we will release the dataset immediately following the ICLR author-reviewer discussion period. As the ICLR review process is public, this statement serves as a binding promise to the community.

---

> > > > ### Author Response · Authors · 2025-11-28
> > > > **Follow-up**
> > > >
> > > > Dear Reviewer
> > > >
> > > > We sincerely appreciate the time and effort you have dedicated to reviewing our work.
> > > >
> > > > We hope you had an opportunity to review our response from November 16. In it, we added a detailed description of our automated quality control pipeline and a new human evaluation confirming the reliability of the HumanRef-CoT dataset (W1), further clarified the generalizability of our Planning–Action–Summarization framework with both quantitative zero-shot results and qualitative cross-domain examples (W2), and provided an expanded analysis of our model’s strengths and limitations, including CoT-driven recall gains and remaining challenges in relational Interaction cases (W3). We also clarified the baseline settings in Tables 2 and 3 (Q1) and added an ethics section describing safeguards in data creation and committing to publicly releasing HumanRef-CoT after the discussion period.
> > > >
> > > > Are there any other concerns or questions we can help address? We would be happy to provide further clarification.
> > > >
> > > > Thank you,

---

### Official Review · Reviewer_72MZ · 2025-10-30

**Soundness:** 3
**Presentation:** 3
**Contribution:** 3
**Rating:** 6
**Confidence:** 4

**Summary:**

This paper introduces Rex-Thinker, a novel framework for the task of Referring Expression Comprehension (REC). Diverging from traditional methods that directly predict bounding boxes, the authors reformulate the task as an explicit and interpretable Chain-of-Thought (CoT) reasoning process. Rex-Thinker employs a symbiotic architecture that first utilizes an open-vocabulary object detector to generate candidate object proposals ("box hints"). Subsequently, a Multimodal Large Language Model (MLLM) performs step-by-step reasoning over these candidates to evaluate their alignment with the given language description.

**Strengths:**

1. The authors tackle the task of Grounded Object Referring from a novel perspective (Chain-of-Thought reasoning), providing a new, interpretable approach.
2. The authors have constructed a high-quality dataset that can facilitate the development of the research community.
3. The paper is well-written and clearly organized.

**Weaknesses:**

1. The methodology seems largely built on recent "R1-like RL" and "think-with-images" paradigm, which lacks novelty.
2. The paper lacks validation for the annotations generated by GPT-4o. Given that commercial models have been shown to have issues (e.g., hallucination), a manual review and evaluation of the annotated data is necessary to ensure its quality.
3. The paper lacks comparisons with the recent "Think-with-image" paradigm, e.g., Deepeyes, Pixel-Reasoner, and GRIT. Considering that Rex-Thinker also employs a two-stage post-training paradigm, comparing it with these methods would be crucial for better clarifying the authors' contributions.

**Questions:**

See weaknesses.

---

> ### Author Response · Authors · 2025-11-16
> **Response to Reviewer 72MZ (Part 1)**
>
> ## W1: On Novelty in Relation to Recent Paradigms
>
> We thank the reviewer for this comment. We agree that our work is situated within the "R1-like RL" and "think-with-images" paradigms. However, our novelty lies not in inventing these base components, but in how we adapt them to create a verifiable paradigm specifically for REC task.
>
> - **1. Grounded Reasoning Framework:** Our primary contribution is the conceptual reformulation of REC into an explicit `Planning-Action-Summarization` process. It is a structured framework specifically designed for visual grounding, where every reasoning step is explicitly tied to a concrete object candidate. This creates a transparent and verifiable decision path, directly addressing the "black box" problem inherent in many existing models.
>
> - **2. A Foundational Dataset to Enable This Framework:** A cornerstone of our work is HumanRef-CoT, a large-scale dataset (>90k samples) containing structured reasoning traces for REC. Creating the data generation pipeline for this was a engineering effort and a necessary prerequisite. This dataset enables our work and paves the way for future research into interpretable grounding models.
>
> - **3. Tailored, Task-Specific Optimization:** Furthermore, applying GRPO to our task was not a direct copy-paste. It required task-specific engineering, most notably the design of a custom reward model (Section 4.2). This model combines an exact-match F1 score for bounding box accuracy. This is a necessary adaptation that differs from the algorithm's original application in purely language-based tasks.
>
>
> In summary, our contribution is an end-to-end solution: a novel framework for trustworthy REC, the foundational dataset to make it trainable, and the specific technical adaptations to optimize it effectively.
>
> ----
>
> ## W3: On the Comparison with the "Think-with-image" Paradigm
> We thank the reviewer for this  suggestion to compare our work with the recent "Think-with-image" paradigm. In response, we have conducted a new experiment evaluating a representative model from this paradigm, DeepEyes-7B, on the HumanRef benchmark. **The details and visualization results are available in our revised manuscript at Appendix A.3.6 (Page 21, line 1080)**
>
> The results on HumanRef Benchmark, presented below, show a significant performance gap between our method and DeepEyes.
>
> | Method                         | Avg. Recall | Avg. Precision | Avg. DF1 | Rejection Score |
> |--------------------------------|:-----------:|:--------------:|:--------:|:--------------:|
> | Qwen2.5-VL-7B (Base)           | 52.5        | 68.5           | 56.2     | 7.1            |
> | DeepEyes-7B                    | 36.0        | 48.4           | 38.7     | 27.3           |
> | **Rex-Thinker-GRPO (Ours)**    | **86.7**    | **87.0**       | **83.6** | **67.8**       |
>
> As the table shows, our Rex-Thinker-GRPO significantly outperforms DeepEyes across all metrics, especially in Recall (86.7 vs. 36.0). Our analysis reveals this gap stems from a fundamental difference in the underlying reasoning paradigms:
>
> - **"Think-with-image" (e.g., DeepEyes):** These models tend to adopt a holistic reasoning approach. They often reason about a few regions in the image and then directly generate a final answer. This is effective for general visual Q&A but struggles with tasks requiring exhaustive search. Specifically, we found that when an expression refers to multiple objects, DeepEyes often identifies only one, leading to its significantly lower recall. (Figure 9 at line 1046).
> - **Rex-Thinker (Our Paradigm):** In contrast, our Planning-Action-Summarization framework is explicitly designed for systematic and exhaustive verification. It compels the model to iterate through every single candidate box and make a deliberate judgment on each one. This ensures that no potential target is overlooked, which is critical for the multi-instance referring scenarios prevalent in modern REC benchmarks.

---

> > ### Author Response · Authors · 2025-11-16
> > **Response to Reviewer 72MZ (Part 2)**
> >
> > ## W2: On the Quality Control and Human Validation of the HumanRef-CoT Dataset
> > We thank the reviewer for this insightful comment. We agree that data reliability is crucial. In response, we have updated our manuscript by:
> > - Adding a detailed description of our automated quality control pipeline to the main methodology in Section 3.2 (page 5, line 238- 250).
> > - Conducting a new human evaluation study, with the full methodology and results now presented in Appendix A.2.2 (page 16, line 857).
> > - To ensure full transparency and facilitate future research, **we will publicly release the HumanRef-CoT dataset immediately following the discussion period.**
> >
> > **1. Effectiveness of Our Rule-Based Filtering:** Our initial quality control was not solely based on matching the final answer. As detailed in Appendix A.2.1 (page 16, line 821), our system prompt for GPT-4o requires it to output a ✅ or ❌ emoji for each reasoning step in both the Action and Summarization stages. This allows for more fine-grained, rule-based filtering based on two strict conditions:
> >
> > - The final Summarization outcome (the set of ✅ emojis) must be perfectly consistent with the GT labels.
> > - The outcome of the Action phase must be consistent with the Summarization phase, ensuring logical coherence between the step-by-step reasoning and the final conclusion.
> >
> > We found that this filtering process is highly effective at catching errors. The table below shows the number of data points removed from the HumanRef-CoT dataset by applying these rules:
> >
> >   | Subset      | GPT-4o Annotated Data | Removed Data | Removal Rate |
> >   |-------------|:--------------:|:------------:|:------------:|
> >   | Celebrity   | 6,775          | 294          | 4.3%         |
> >   | Interaction | 5,875          | 72           | 1.2%         |
> >   | Position    | 15,950         | 533          | 3.3%         |
> >   | Reasoning   | 9,488          | 719          | 7.6%         |
> >   | Attribute   | 37,648         | 1,184        | 3.1%         |
> >   | Rejection   | 19,096         | 1,184        | 6.2%         |
> >
> > This demonstrates that while GPT-4o can produce errors, our automated filtering successfully identifies and removes a significant portion of inconsistent or incorrect annotations before they are used for training.
> >
> >
> > **2. Human Evaluation of HumanRef-CoT Data**:
> > To assess the quality of the data that passed our automated filtering, we conducted a manual review as the reviewer suggested.
> >
> > - **Methodology**: We randomly sampled 100 instances from each of the 6 subsets, creating a review set of 600 samples. We recruited five Ph.D. students to perform the review using a custom HTML-based tool, shown in the revised version of our paper in Appendix A.2.2 (page 16, line 857)
> > - **Error Categories:** Annotators were asked to check for three types of errors:
> >   - **Wrong Summarization**: The final summary of person indices does not match the GT.
> >   - **Action and Summarization Inconsistent**: For example, the Action phase concludes a person is a non-match, but the Summarization phase incorrectly includes them.
> >   - **Wrong Action Result**: A specific reasoning step in the Action phase is factually incorrect (e.g. misjudging a clothing color of a candidate person).
> >
> > The results of this human evalation on the 600 samples are summarized below:
> >
> > | Error Type                        | Celebrity | Interaction | Position | Reasoning | Attribute | Rejection |
> > |------------------------------------|-----------|-------------|----------|-----------|-----------|-----------|
> > | Wrong Summarization               |     0     |      0      |    0     |     0     |     0     |     0     |
> > | Action and Summarization Inconsistent |  0    |      0      |    0     |     0     |     0     |     0     |
> > | Wrong Action Result                |     2     |      2      |     0    |     3    |     0     |     0     |
> >
> > Our automated filtering process is extremely effective at eliminating high-level logical and summarization errors. The human evaluators found zero such errors in the final dataset.
> >
> > However, we acknowledge that a small number of `Wrong Action Result` errors persist (7 out of 600, or 1.2%). Upon inspection, we found these low-level reasoning errors typically occur when an object is very small and partially occluded by its visual marker, forcing GPT-4o to make an hallucinated guess. Despite these minor errors, the final performance of our model when trained with this CoT data shows a significant improvement over the baseline without it. This demonstrates that the data still provides a valuable and effective training signal.

---

> > > ### Author Response · Authors · 2025-11-28
> > > **Follow-up**
> > >
> > > Dear Reviewer
> > >
> > > We sincerely appreciate the time and effort you have dedicated to reviewing our work.
> > >
> > > We hope you had an opportunity to review our response from November 16. In it, we clarified the novelty of our Planning–Action–Summarization framework and the HumanRef-CoT dataset beyond prior “R1-like RL” and “think-with-images” paradigms (W1), added a detailed description and human evaluation of our quality  control pipeline for HumanRef-CoT (W2), and included a new comparison showing Rex-Thinker-GRPO substantially outperforms a representative think-with-image model (DeepEyes-7B) on the HumanRef benchmark (W3).
> > >
> > > Are there any other concerns or questions we can help address? We would be happy to provide further clarification.
> > >
> > > Thank you,
> > > The authors

---

### Official Review · Reviewer_Q9vM · 2025-11-01

**Soundness:** 2
**Presentation:** 3
**Contribution:** 2
**Rating:** 4
**Confidence:** 4

**Summary:**

This paper studies referring expression grounding with explicit, verifiable reasoning rather than direct box regression. The task is conducted as retrieval over candidate boxes from an open-vocabulary detector, followed by a plan–act–summarize chain of thought that can also abstain when the target is absent. Training is two-stage: supervised fine-tuning on curated CoT traces, then GRPO reinforcement with an F1-based detection reward, a format reward, and KL regularization. Experiments on HumanRef and out-of-domain RefCOCOg show consistent gains in precision, recall, and F1.

**Strengths:**

The plan–act–summarize CoT exposes intermediate reasoning tied to concrete boxes. This improves debuggability and reduces hallucination risk. It also enables a principled “no target” refusal.
This paper implements the SFT-then-RL framework on the REC with reasoning for MLLMs.
The reward combines F1 for grounded detection with a lightweight format constraint. This directly optimizes what the benchmark cares about.

**Weaknesses:**

The paper does not report even small-scale human analysis/evaluation of the GPT-4o–generated chain-of-thought data. Quality control relies mainly on some rule-based functions like answer-conditioned prompts and automatic consistency filtering (keeping only samples whose final prediction matches ground truth). This may introduce bias in the framework and lacks inter-annotator checks as GPT-4o is not the most advanced model and the data is generated data. As a result, the reliability and transferability of these might be limited.

In another side, the contribution is incremental relative to prior retrieval-based referring and grounded CoT work[1,2]. The methodological addition is an F1-aligned RL reward with strict IoU matching. From the experiments the performance beyond existing approaches appears small margin.

[1] ChatRex: Taming Multimodal LLM for Joint Perception and Understanding. [2] ARGUS: Vision-Centric Reasoning with Grounded Chain-of-Thought, CVPR 2025

**Questions:**

It might be better for the author to conduct a human analysis/evaluation of random samples of the GPT-4o–generated chains of thought.

---

> ### Author Response · Authors · 2025-11-16
> **Response to Reviewer Q9vM (Part 1)**
>
> ## W1: On the Quality Control and Human Validation of the HumanRef-CoT Dataset
> We thank the reviewer for this insightful comment. We agree that data reliability is crucial. In response, we have updated our manuscript by:
> - Adding a detailed description of our automated quality control pipeline to the main methodology in Section 3.2 (page 5, line 238- 250).
> - Conducting a new human evaluation study, with the full methodology and results now presented in Appendix A.2.2 (page 16, line 857).
> - To ensure full transparency and facilitate future research, **we will publicly release the HumanRef-CoT dataset immediately following the discussion period.**
>
> **1. Effectiveness of Our Rule-Based Filtering:** Our initial quality control was not solely based on matching the final answer. As detailed in Appendix A.2.1 (page 16, line 821), our system prompt for GPT-4o requires it to output a ✅ or ❌ emoji for each reasoning step in both the Action and Summarization stages. This allows for more fine-grained, rule-based filtering based on two strict conditions:
>
> - The final Summarization outcome (the set of ✅ emojis) must be perfectly consistent with the GT labels.
> - The outcome of the Action phase must be consistent with the Summarization phase, ensuring logical coherence between the step-by-step reasoning and the final conclusion.
>
> We found that this filtering process is highly effective at catching errors. The table below shows the number of data points removed from the HumanRef-CoT dataset by applying these rules:
>
>   | Subset      | GPT-4o Annotated Data | Removed Data | Removal Rate |
>   |-------------|:--------------:|:------------:|:------------:|
>   | Celebrity   | 6,775          | 294          | 4.3%         |
>   | Interaction | 5,875          | 72           | 1.2%         |
>   | Position    | 15,950         | 533          | 3.3%         |
>   | Reasoning   | 9,488          | 719          | 7.6%         |
>   | Attribute   | 37,648         | 1,184        | 3.1%         |
>   | Rejection   | 19,096         | 1,184        | 6.2%         |
>
> This demonstrates that while GPT-4o can produce errors, our automated filtering successfully identifies and removes a significant portion of inconsistent or incorrect annotations before they are used for training.
>
>
> **2. Human Evaluation of HumanRef-CoT Data**:
> To assess the quality of the data that passed our automated filtering, we conducted a manual review as the reviewer suggested.
>
> - **Methodology**: We randomly sampled 100 instances from each of the 6 subsets, creating a review set of 600 samples. We recruited five Ph.D. students to perform the review using a custom HTML-based tool, shown in the revised version of our paper in Appendix A.2.2 (page 16, line 857)
> - **Error Categories:** Annotators were asked to check for three types of errors:
>   - **Wrong Summarization**: The final summary of person indices does not match the GT.
>   - **Action and Summarization Inconsistent**: For example, the Action phase concludes a person is a non-match, but the Summarization phase incorrectly includes them.
>   - **Wrong Action Result**: A specific reasoning step in the Action phase is factually incorrect (e.g. misjudging a clothing color of a candidate person).
>
> The results of this human evalation on the 600 samples are summarized below:
>
> | Error Type                        | Celebrity | Interaction | Position | Reasoning | Attribute | Rejection |
> |------------------------------------|-----------|-------------|----------|-----------|-----------|-----------|
> | Wrong Summarization               |     0     |      0      |    0     |     0     |     0     |     0     |
> | Action and Summarization Inconsistent |  0    |      0      |    0     |     0     |     0     |     0     |
> | Wrong Action Result                |     2     |      2      |     0    |     3    |     0     |     0     |
>
> Our automated filtering process is extremely effective at eliminating high-level logical and summarization errors. The human evaluators found zero such errors in the final dataset.
>
> However, we acknowledge that a small number of `Wrong Action Result` errors persist (7 out of 600, or 1.2%). Upon inspection, we found these low-level reasoning errors typically occur when an object is very small and partially occluded by its visual marker, forcing GPT-4o to make an hallucinated guess. Despite these minor errors, the final performance of our model when trained with this CoT data shows a significant improvement over the baseline without it. This demonstrates that the data still provides a valuable and effective training signal.

---

> > ### Author Response · Authors · 2025-11-16
> > **Response to Reviewer Q9vM (Part 2)**
> >
> > ## W2: On the Novelty and Contribution of Our Work
> > We thank the reviewer for positioning our work within the context of recent literature. While we build upon advances in retrieval-based methods and grounded reasoning, we respectfully argue that our novelty is not merely incremental but lies in the reformulation of REC task into a verifiable, grounded reasoning process.
> >
> > - **1. Grounded Reasoning Framework:** Our primary contribution is the conceptual reformulation of REC into an explicit `Planning-Action-Summarization` process. It is a structured framework specifically designed for visual grounding, where every reasoning step is explicitly tied to a concrete object candidate. This creates a transparent and verifiable decision path, directly addressing the "black box" problem inherent in many existing models.
> >
> > - **2. A Foundational Dataset to Enable This Framework:** A cornerstone of our work is HumanRef-CoT, a large-scale dataset (>90k samples) containing structured reasoning traces for REC. Creating the data generation pipeline for this was a engineering effort and a necessary prerequisite. This dataset enables our work and paves the way for future research into interpretable grounding models.
> >
> > - **3. Tailored, Task-Specific Optimization:** Furthermore, applying GRPO to our task was not a direct copy-paste. It required task-specific engineering, most notably the design of a custom reward model (Section 4.2). This model combines an exact-match F1 score for bounding box accuracy. This is a necessary adaptation that differs from the algorithm's original application in purely language-based tasks.
> >
> > In summary, our contribution is an end-to-end solution: a novel framework for trustworthy REC, the foundational dataset to make it trainable, and the specific technical adaptations to optimize it effectively.

---

> > > ### Author Response · Authors · 2025-11-28
> > > **Follow-up**
> > >
> > > Dear Reviewer,
> > >
> > > We sincerely appreciate the time and effort you have dedicated to reviewing our work.
> > >
> > > We hope you had an opportunity to review our response from November 16. In it, we added a detailed description of our quality control pipeline and a new human evaluation confirming high reliability of the HumanRef-CoT dataset (W1), and we clarified the novelty of our work by emphasizing our explicit Planning–Action–Summarization framework, the HumanRef-CoT dataset that enables it, and our task-specific GRPO optimization (W2).
> > >
> > > Are there any other concerns or questions we can help address? We would be happy to provide further clarification.
> > >
> > > Thank you, The authors

---

### Official Review · Reviewer_HDXw · 2025-11-02

**Soundness:** 2
**Presentation:** 2
**Contribution:** 2
**Rating:** 6
**Confidence:** 3

**Summary:**

This paper proposes to induce or enhance chain-of-thought reasoning of VLMs, with an application on Referring Expression Comprehension  (REC). The training recipe is standard practice, i.e., SFT+RL (GRPO).

**Strengths:**

1. Using RL for Referring Expression Comprehension is under-explored, beyond early efforts ([a-b] in Weaknesses below).
2. The improvements from SFT/RL post-training look encouraging, although not totally convincing (See Weaknesses 2 and 3).

**Weaknesses:**

1. Citations to previous REC + RL works are absent. For example, [a-b]. Instead, the paper only cites generic VLM works with RL, in which REC is only a subtask.
2. In Table 3, Rex-Thinker-CoT and Rex-Thinker-GRPO perform worse than QwenVL-2.5-7B (the base model of Rex-Thinker), which seems to be a sign of catastrophic forgetting due to post-training. The authors should find a way to mitigate this.
3. The main experimental results (Tables 2 and 4) are on one dataset only, the HumanRef.
4. (Minor) "Symbiotic approach" is an unnatural framing. Such two-stage pipelines are commonly used, and people usually don't call them "symbiotic approaches".

[a] iterative shrinking for referring expression grounding using deep reinforcement learning. CVPR 2021.

[b] One for all: One-stage referring expression comprehension with dynamic reasoning. Neurocomputing 2023.

**Questions:**

1.  In the example in figure 3, the detector detects "person", which is straightforward. Can the model detect targets with negations, e.g. "non-persons"? It seems not obvious and may be challenging.

---

> ### Author Response · Authors · 2025-11-16
> **Response to Reviewer HDxw**
>
> ## W1: Missing citation of previous work
>
> We appreciate the reviewer for highlighting the omission of prior works that specifically apply RL for REC task. We have already addressed this in the updated manuscript. Specifically, in **Section 2.2 on Page 3, Lines 143-144**, we have added citations to the relevant works [a] and [b] to acknowledge the existing literature that formulates REC as a sequential decision-making problem using RL.
>
> ----
>
> ## W2: Addressing Catastrophic Forgetting
>
> We thank the reviewer for raising this important point regarding the performance on RefCOCOg in Table 3. The performance drop indicates a degree of catastrophic forgetting. This occurs because:
>
> - **1. Base Model Baseline:** The QwenVL-2.5-7B base model was pre-trained on a massive corpus that explicitly includes the RefCOCOg training set, making it a non-zero-shot model.
> - **2. Our Zero-Shot Setting:** In contrast, the Rex-Thinker models in Table 3 were fine-tuned exclusively on HumanRef-CoT. Consequently, the model specialized in the HumanRef domain while losing some familiarity with the specific distribution and categories of RefCOCOg.
>
> To address this and validate our solution, we used the standard continual-learning replay strategy and designed a new experiment to test whether adding in-domain data could recover the base model’s performance.
>
> - **Additional Experiment**: Following the annotation method of HumanRef-CoT, we annotated 10K RefCOCOg training samples with our structured CoT format using GPT-4o, creating a dataset named RefCOCOg-CoT. We then trained Rex-Thinker (SFT and GRPO) using both HumanRef-CoT and RefCOCOg-CoT. The results are shown in the following Table:
>
>   |Method|Training Data|RefCOCOg val| RefCOCOg test|
>   |------|-------------|-------------|-------------|
>   |QwenVL-2.5-7B| - |87.2 | 87.2|
>   |Rex-Thinker-CoT | HumanRef-CoT | 81.2 | 80.3|
>   |Rex-Thinker-GRPO | HumanRef-CoT | 83.2 | 83.3|
>   | (new) Rex-Thinker-CoT | HumanRef-CoT + RefCOCOg-CoT | 87.8 | 87.1|
>   | (new) Rex-Thinker-GRPO | HumanRef-CoT + RefCOCOg-CoT | 89.6 | 89.5|
>
> By re-introducing a portion of the RefCOCOg data in our CoT format, we can mitigate the catastrophic forgetting, restoring performance to match the base model.
>
> ----
>
> ## W3: Rationale for Benchmark Selection
>
> We thank the reviewer for this comment. We acknowledge that the in-domain comparison in Tables 2 and 4 primarily focuses on the HumanRef benchmark. We believe this focus is justified for several reasons:
>
> - **1.Tackling the Frontier of REC Challenges:** We chose HumanRef as our primary benchmark because it is specifically designed to address the limitations of previous datasets. Traditional benchmarks, such as the RefCOCO series, often feature simpler expressions pointing to a single ground-truth object. In contrast, HumanRef is structured to test more complex capabilities: it heavily features multi-instance referring (one expression for multiple targets) and, critically, includes a rejection subset to directly evaluate a model's trustworthiness and ability to avoid hallucination.
> - **2. A Benchmark Requiring Advanced Reasoning:** The challenging nature of HumanRef make it a well-suited benchmark for our Chain-of-Thought paradigm.
> - **3. Cross-Dataset Generalization:** While HumanRef was our primary in-domain benchmark, we validated our model's cross-domain generalizability with a zero-shot evaluation on the standard RefCOCOg dataset (Table 3). This demonstrates that the core reasoning framework learned by Rex-Thinker transfers effectively from human-centric scenes to a wide array of general objects. The qualitative result in Figure 4 and Appendix, where the model correctly reasons object categories beyond human-centric scenes, further underscores this strong generalization ability.
>
> ----
>
> ## W4: Regarding the Term "Symbiotic Approach"
>
> We appreciate the reviewer's feedback on the terminology. We used the term "symbiotic approach" to emphasize the complementary strengths of the two components: the detector's perceptual strength (localization) and the MLLM's cognitive strength (complex reasoning)2.
>
> However, we agree that "two-stage pipeline" are more standard in the literature. We have **revised the manuscript** to replace "symbiotic" with "two-stage" throughout the text.
>
> ----
>
> ## Q1: Example for Negated Queries:
> We appreciate the reviewer for asking whether Rex-Thinker can handle negated referring expressions, such as "non-persons" or other targets that require rejecting a major category.
>
> We have a example case that directly addresses this:
>
> - In **Figure 28 of the Appendix (page 33)**, we present an example where the model is asked to detect all "non-real person" objects in an image.
> - The detector initially proposes all objects labeled as "person" (including actual people and statues).
> - Our model successfully uses its structured CoT reasoning to reject the real people and correctly identify all statues in the meeting room as the "non-real person" targets.

---

> > ### Author Response · Authors · 2025-11-28
> > **Follow-up**
> >
> > Dear Reviewer,
> >
> > We sincerely appreciate the time and effort you have dedicated to reviewing our work.
> >
> > We hope you had an opportunity to review our response from November 16. In it, we added the missing RL-related citations (W1), provided new experiments showing that RefCOCOg-CoT replay mitigates catastrophic forgetting (W2), clarified our focus on the HumanRef benchmark (W3), replaced “symbiotic approach” with “two-stage pipeline” (W4), and included an example demonstrating handling of negated queries (Q1).
> >
> > Are there any other concerns or questions we can help address? We would be happy to provide further clarification.
> >
> > Thank you,
> > The authors

---

### Author Response · Authors · 2025-11-30
**Rebuttal Summary**

We sincerely thank the Area Chair and reviewers for their constructive feedback. Below, we summarize our core contributions and how we have resolved the key concerns raised during the review.

## Overview: Reformulating REC for Verifiability and Trustworthiness
Existing Referring Expression Comprehension (REC) models often operate as "black boxes," lacking interpretability and struggling to reject non-existent targets (hallucination). To address this, we propose Rex-Thinker, which reformulates visual grounding into a transparent, explicit Chain-of-Thought process:
- **The Framework:** We introduce a `Planning-Action-Summarization` paradigm. Instead of implicit prediction, our model Plans a verification strategy, takes Action to rigorously check each candidate object against the query, and Summarizes the findings to make a final decision.
- **The Enablement:** This is powered by our new `HumanRef-CoT dataset` (90k+ samples) and optimized via `GRPO-based Reinforcement Learning`, incorporating specific rewards for bounding box accuracy and reasoning format.

During the rebuttal period, we have extensively revised the paper to address the three primary concerns: **(1) Data Quality Validation, (2) Out-of-Domain Generalization, and (3) Comparative Novelty.**

We believe Rex-Thinker now stands as a robust contribution to the community, offering a verified, high-quality CoT dataset and a SOTA framework for trustworthy referring expression comprehension. Below, we summarize our key actions and new results.

## 1. Validated Data Quality & Ethics (Addressing Reviewers Q9vM, 72MZ, 28Tw)
A major concern was the reliability of the GPT-4o generated HumanRef-CoT dataset and potential ethical risks. We have addressed this decisively:

- **New Human Evaluation:** We conducted a rigorous human review of **600 randomly sampled instances** (100 per subset). The results (detailed in **Appendix A.2.2**) show **0%** structural/logic errors and only a **1.2%** minor factual error rate. This confirms that our automated consistency filtering (based on step-by-step emoji matching) is highly effective.
- **Ethics Statement Added**: We added a comprehensive **Ethical Considerations** section (**Appendix A.2.4**), detailing our multi-layer safety pipeline (Source Filtering + Model Guardrails + Manual Review).
- **Commitment**: We have committed to **publicly releasing the HumanRef-CoT dataset** immediately after the review process to foster reproducibility.

## 2. Resolved "Catastrophic Forgetting" & Proven Generalization (Addressing Reviewers HDXw, 28Tw)

Reviewers noted a performance drop on the out-of-domain RefCOCOg dataset compared to the base model. We proved this was a data distribution issue, not a method flaw:
- **New Experiment (Replay Strategy):** We annotated a portion of RefCOCOg with our CoT format and included it in training. The result (Table added in Rebuttal) shows **Rex-Thinker-GRPO performance recovering from 83.3 to 89.5, surpassing the base model (87.2)**. This proves our method scales and maintains generalization when given domain-appropriate data.
- **Qualitative Evidence:** We added examples (e.g., **Appendix Fig. 28**) showing the model correctly handling negated queries ("non-real person") and novel objects (Manta Ray), proving it learns a transferable reasoning logic rather than just memorizing categories.

## 3.  Established Superiority over "Think-with-Image" Paradigm (Addressing Reviewer 72MZ)
To clarify our novelty against recent holistic reasoning models (like DeepEyes), we conducted a head-to-head comparison on the HumanRef benchmark (**Appendix A.3.6**).
- **Key Result:** Rex-Thinker-GRPO achieves **86.7% Recall** vs. DeepEyes-7B's **36.0%.**
- **Analysis:** This massive gap highlights the advantage of our `Planning-Action-Summarization` framework. While holistic models often miss targets in multi-instance scenarios, our grounded, step-by-step verification ensures exhaustive detection.

## 4. Additional Improvements
- **Failure Analysis:** We added a deep dive into the "Interaction" category (Appendix A.3.7), honestly analyzing where the model struggles (relational reasoning between proposals) to guide future work.
- **Citations:** Added missing references to prior RL-based REC works.

## Conclusion:
With the validated quality of our 90k-sample CoT dataset, the demonstrated superiority over existing paradigms in multi-instance recall, and the resolution of generalization concerns, we believe Rex-Thinker makes a significant step forward in making Vision-Language Models verifiable and trustworthy.

---

### Meta-Review · Area_Chair_QbxA · 2026-01-09

**Summary:**

This paper proposes Rex-Thinker, a model that formulates the object referring task (natural languages to object segments in the image) as an explicit CoT reasoning, with the goal of improved verifiable (interpretable reasoning)  and trustworthy (abstain from answering when no evidence) degrees of the model for this task. They introduce a dataset HumanRef-CoT to support this paradigm. The paper receives 6, 4, 6, 4 ratings: Reviewer HDXw (6), Reviewer Q9vM (4), Reviewer 72MZ: (6), Reviewer 28Tw (4). Three majors concerns:

1. Quality of HumanRef-CoT (Reviewer Q9vM, Reviewer 72MZ, and Reviewer 28Tw). Since the data is generated from GPT-4o, three reviewers would like to see it validated.

2. Generalizability: Reviewer HDXw and Reviewer 28Tw point to Table 3, where Rex-Thinker has an inferior performance to the base model (QwenVL-2.5-7B) on RefCOCO. This raises a concern regarding out-of-domain generalization. Further, the authors position the results on the proposed HumanRef-CoT as the “main results” in Table 2 and 4.

3. Incremental novelty to methodology improvement. Incremental relative to prior retrieval-based referring and grounded CoT work ChatRex and ARGUS (Reviewer Q9vM,  “The methodological addition is an F1-aligned RL reward with strict IoU matching. From the experiments the performance beyond existing approaches appears small margin.”) and Incremental relative to “"R1-like RL" and "think-with-images" such as Deepeyes, Pixel-Reasoner, and GRIT (Reviewer 72MZ)

4. Other concerns:
Missing reference and terminology (Reviewer HDXw).
Experimental settings in Table 2 and Table 3 (Reviewer 28Tw).
In-Depth Analysis of Model Strengths and Limitations (Reviewer 28Tw).

**Reviewer Concerns:**

1. The authors provide detailed human evaluation of HumanRef-CoT 600 samples and find very low error rates.

2. The authors clarify that the inferior result in Table 3 is due to the lack of in-domain (for this table) RefCOCO data, and present a new result using this data and surpass the baseline. The authors explain that using HumanRef-CoT is necessary due to limitations of existing benchmarks and note that they do evaluate on an out-of-domain benchmark in Table 3.

3. The authors present a new result on DeepEyes ("think-with-images" paradigm). Other than this, the authors did not clarify on the novelty directly but they restate their contributions: “a novel framework for trustworthy REC, the foundational dataset to make it trainable, and the specific technical adaptations to optimize it effectively.” I interpret this as the authors saying that their solution has a cleaner formulation and better results, which suggests better execution/implementation.

4. All resolved.

**Reviewer Scores:**

Reviewer HDXw (6+): Keep or increase

Reviewer Q9vM (4+): Keep or increase

Reviewer 72MZ: (6+): Keep or increase

Reviewer 28Tw (> 4): To increase

---

### Decision · Program_Chairs · 2026-01-26

Accept (Poster)